# MIXED-CURVATURE DECISION TREES AND RANDOM FORESTS

## ABSTRACT

Decision trees (DTs) and their random forest (RF) extensions are workhorses of classification and regression in Euclidean spaces. However, algorithms for learning in non-Euclidean spaces are still limited. We extend DT and RF algorithms to product manifolds: Cartesian products of several hyperbolic, hyperspherical, or Euclidean components. Such manifolds handle heterogeneous curvature while still factorizing neatly into simpler components, making them compelling embedding spaces for complex datasets. Our novel angular reformulation of DTs respects the geometry of the product manifold, yielding splits that are geodesically convex, maximum-margin, and composable. In the special cases of single-component manifolds, our method simplifies to its Euclidean or hyperbolic counterparts, or introduces hyperspherical DT algorithms, depending on the curvature. We benchmark our method on various classification, regression, and link prediction tasks on synthetic data, graph embeddings, mixed-curvature variational autoencoder latent spaces, and empirical data. Compared to six other classifiers, product DTs and RFs ranked first on 21 of 22 single-manifold benchmarks and 18 of 35 product manifold benchmarks, and placed in the top 2 on 53 of 57 benchmarks overall. This highlights the value of product DTs and RFs as straightforward yet powerful new tools for data analysis in product manifolds.

## 1 INTRODUCTION

While much of machine learning focuses on Euclidean spaces, these can fail to capture the true structure of complex datasets. For example, hierarchical structures, which are common in taxonomy (e.g., phylogenetic trees) are better represented in hyperbolic space due to its exponential volume growth, which naturally mirrors tree-like data (Sonthalia & Gilbert, 2020). Similarly, cyclical structures, often encountered in time-series data with periodic patterns (e.g., seasonal trends, neuronal spiking dynamics), can benefit from spherical representations (Ding & Regev, 2021).

However, many real-world datasets don't conform to a single geometric structure. Any constant-curvature manifold——whether hyperbolic, spherical, or Euclidean——would struggle to represent all the nuances of such data simultaneously. Product manifolds, as proposed by Gu et al. (2018), offer a solution. By combining multiple constant-curvature component manifolds (spherical, Euclidean, and hyperbolic spaces) into a single product manifold, they can better capture the complexity of such mixed-structure data. This flexibility reduces distortion when modeling pairwise distances and enables a more accurate representation of the underlying data structure.

Despite their advantages, product manifolds have seen limited adoption in machine learning, particularly for inference tasks like classification and regression. Existing work has primarily focused on applications in biology (McNeela et al., 2024) and knowledge graphs (Wang et al., 2021). However, tools for leveraging product manifold representations in downstream tasks remain scarce.

In this paper, we introduce mixed-curvature decision trees (DTs) and random forests (RFs), expanding the toolkit for analyzing product manifold data. By enabling inference directly on product manifold coordinates, our approach is well-suited for datasets that combine hierarchical, cyclical, and other complex geometric patterns. This framework provides a principled way to learn from such structures, achieving more accurate results than competing models. These contributions offer new possibilities for applying product manifold representations in fields ranging from biological modeling to temporal-spatial analysis.

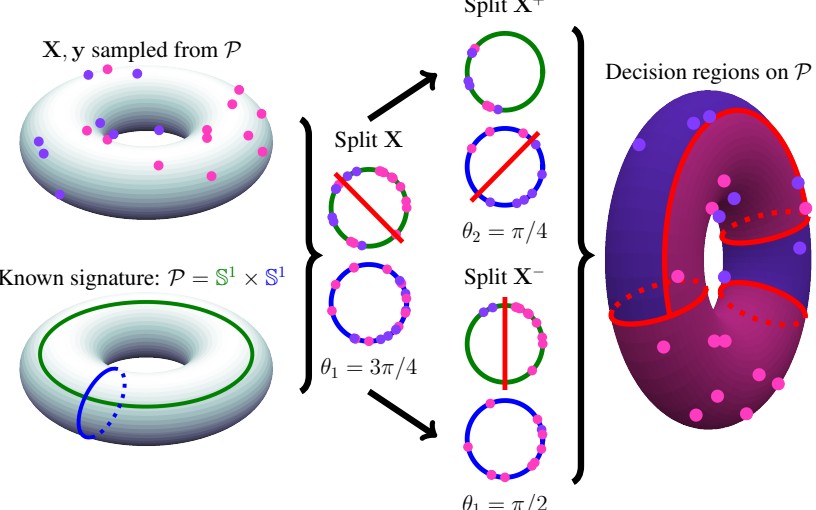

Figure 1: An illustration of the product manifold DT in action. We consider a sample of labeled points $(\mathbf{X}, \mathbf{y})$ from one of the simplest possible product manifolds: the torus $\mathcal{P} = \mathbb{S}^1 \times \mathbb{S}^1$. Since we know the signature for $\mathcal{P}$, we can factorize $\mathbf{X}$ into coordinates on two circles. Our DT **splits** these factorized coordinates to a maximum depth of 2, partitioning $\mathcal{P}$ into a total of $2^2 = 4$ disjoint decision areas (colored **positive** or **negative** to reflect the classes).

**Our contributions:**

1. We generalize DTs and RFs to *all* constant-curvature manifolds. Unlike existing methods, we represent data and splits as angles in two-dimensional subspaces. This guarantees splits are geodesically convex, maximum-margin, and composable. In the single-manifold case, this extends existing Euclidean and hyperbolic models or introduces *hyperspherical* DTs and RFs.
2. We introduce novel DT and RF algorithms for product manifolds.
3. We extend techniques for sampling distributions in non-Euclidean manifolds to describe mixtures of Gaussians in product manifolds.
4. We show how problems like link prediction in graphs and signal analysis can be recast as inference problems on product manifolds.
5. We demonstrate the effectiveness of our component- and product-manifold algorithms over competing algorithms on a suite of 57 diverse non-Euclidean benchmarks.

## 1.1 RELATED WORK

**Non-Euclidean representation learning.** Important background on manifolds in machine learning is given in Cayton (2005) and Bengio et al. (2014). Much of the work on product manifolds is indebted to early works on hyperbolic spaces, including Nickel & Kiela (2017); Chamberlain et al. (2017), and Ganea et al. (2018).

**Machine learning in product manifolds.** Tabaghi et al. (2021) describe linear classifiers, including perceptron and support vector machines; Tabaghi et al. (2024) adapt principal component analysis; and Cho et al. (2023) generalize Transformer architectures to product manifolds.

**Computationally tractable manifolds.** Besides product manifolds, other methods for representing data with heterogeneous curvature also exist: Borde & Kratsios (2023) is based on fractals, while Cruceru et al. (2020) is based on matrix manifolds.

**Product manifold-derived features.** Tsagkrasoulis & Montana (2017) train RF classifiers on distance matrices from arbitrary manifolds, e.g. product manifolds. Sun et al. (2021) and Borde et al. (2023b) use product manifolds to compute rich similarity measures as features for classification. Giovanni et al. (2022) introduce a heterogeneous variant of product manifolds; Borde et al. (2024) combine quasi-metrics and partial orders in a product manifold for graph representations.

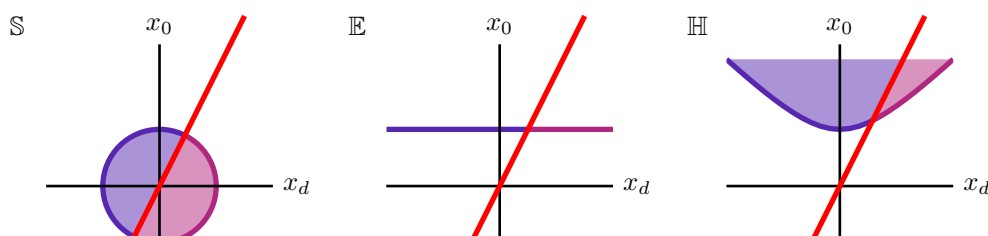

Figure 2: Decision boundaries in any constant-curvature manifold are found by 2-D projections into 2-dimensional subspaces. In this perspective, both data and splits are parameterized by an angle $\theta$. Each **split** divides the manifold into **positive-class regions** and **negative-class regions**.

**Hyperbolic random forests.** Our method is inspired by recent work by Doorenbos et al. (2023) and Chlenski et al. (2024) extending RFs to hyperbolic space. In particular, our angular-split perspective synthesizes the ideas in Chlenski et al. (2024) and Tabaghi et al. (2021).

**Applications of product manifolds.** Product manifolds are popular for embedding knowledge graphs Wang et al. (2021); Li et al. (2024); Nguyen-Van et al. (2023). In biology, they have been used to represent pathway graphs (McNeela et al., 2024), cryo-EM images (Zhang et al., 2021), and single-cell transcriptomic profiles (Tabaghi et al., 2021). Skopek et al. (2020) also embed image datasets into product manifolds.

## 2 PRELIMINARIES

We review relevant details of different Riemannian manifolds (Euclidean spaces, hyperspheres, hyperboloids, and product manifolds), along with key properties of the Euclidean and hyperbolic variants of DTs and RFs.

### 2.1 RIEMANNIAN MANIFOLDS

We will begin by reviewing key details of hyperspheres, hyperboloids, and Euclidean spaces. For more details, readers can consult Do Carmo (1992).

Each space described is a Riemannian manifold, meaning that it is locally isomorphic to Euclidean space and equipped with a distance metric. The shortest paths between two points $\mathbf{u}$ and $\mathbf{v}$ on a manifold are called geodesics. As all three spaces we consider have constant Gaussian curvature, we define simple closed forms for geodesic distances in each of the following subsections in lieu of a more general discussion of geodesic distances in arbitrary Riemannian manifolds.

Any constant-curvature manifold $\mathcal{M}$ is parameterized by a dimensionality $D$ and a curvature $K$. They can also all be considered embedded in an ambient space $\mathbb{R}^{D+1}$. Finally, for each point $\mathbf{x} \in \mathcal{M}$, the tangent plane at $\mathbf{x}$, $T_{\mathbf{x}}\mathcal{M}$, is the space of all tangent vectors at $\mathbf{x}$:

$$T_{\mathbf{x}}\mathcal{M} = \{\mathbf{x}' \in \mathcal{M} \; : \; \langle x', x \rangle_{\mathcal{M}} = 0\}. \tag{1}$$

#### 2.1.1 EUCLIDEAN SPACE

Euclidean spaces are naturally understood as $\mathbb{R}^D$, but we will use the notation $\mathbb{E}^D = \mathbb{R}^D$ when treating Euclidean spaces as manifolds. In contrast, we will continue to use $\mathbb{R}^D$ to refer to ambient spaces. Euclidean spaces use the familiar inner product (dot product), norm ($\ell_2$ norm), and distance function (Euclidean distance):

$$\langle \mathbf{u}, \mathbf{v} \rangle = u_0 v_0 + u_1 v_1 + \ldots + u_2 v_2, \tag{2}$$

$$\|\mathbf{u}\| = \sqrt{\langle \mathbf{u}, \mathbf{u} \rangle}, \tag{3}$$

$$\delta_{\mathbb{E}}(\mathbf{u}, \mathbf{v}) = \|\mathbf{u} - \mathbf{v}\|. \tag{4}$$

### 2.1.2 HYPERSPHERICAL SPACE

Hyperspheres can be viewed as surfaces *embedded* in a higher-dimensional, Euclidean ambient space. Hyperspherical space uses the same inner products as Euclidean space. The hypersphere is the set of points in the ambient space having a Euclidean norm equal to some radius inversely proportional to the curvature $K > 0$:

$$\mathbb{S}^{D,K} = \{\mathbf{x} \in \mathbb{R}^{D+1} \ : \ \|x\| = 1/K\}. \tag{5}$$

Because shortest paths between two points $\mathbf{u}$ and $\mathbf{v}$ in $\mathbb{S}^{D,K}$ through the ambient space leave the surface of the manifold, we must define the hyperspherical distance function for the shortest path entirely in $\mathbb{S}^{D,K}$ between $\mathbf{u}$ and $\mathbf{v}$:

$$\delta_{\mathbb{S}}(\mathbf{u}, \mathbf{v}) = \cos^{-1}(K^2 \langle \mathbf{u}, \mathbf{v} \rangle)/K. \tag{6}$$

### 2.1.3 HYPERBOLIC SPACE

Hyperbolic space is characterized by constant negative metric curvature. This has several consequences: for instance, the angles in any triangle sum to less than $\pi$, many lines through a point can be parallel to any given line, and neighborhoods grow exponentially with radius.

There are several equivalent models of hyperbolic space. For our purposes, we will describe the hyperbolic space from the perspective of the hyperboloid model. First, we must define the ambient Minkowski space. This is a vector space equipped with the Minkowski inner product:

$$\langle \mathbf{u}, \mathbf{v} \rangle_{\mathcal{L}} = -u_0 v_0 + u_1 v_1 + \ldots + u_n v_n. \tag{7}$$

Similar to the Euclidean case, we let $\|\mathbf{u}\|_{\mathcal{L}} = \langle \mathbf{u}, \mathbf{u} \rangle_{\mathcal{L}}$ (we do not wish to take the square root of a negative number). The hyperboloid of dimension $D$ and curvature $K < 0$, written $\mathbb{H}^{D,K}$, is a set of points with constant Minkowski norm:

$$\mathbb{H}^{D,K} = \{\mathbf{x} \in \mathbb{R}^D \ : \ \|\mathbf{x}\|_{\mathcal{L}} = -1/K^2, \ x_0 > 0\}, \tag{8}$$

Finally, the hyperbolic distance function for geodesic distances between $\mathbf{u}, \mathbf{v} \in \mathbb{H}^{D,K}$ is given by

$$\delta_{\mathbb{H}}(\mathbf{u}, \mathbf{v}) = -\cosh^{-1}(K^2 \langle \mathbf{u}, \mathbf{v} \rangle_{\mathcal{L}})/K. \tag{9}$$

### 2.1.4 MIXED-CURVATURE PRODUCT MANIFOLDS

We reiterate the definition of product manifolds from Gu et al. (2018). A product manifold $\mathcal{P}$ is the Cartesian product of one or more spherical, Euclidean, and hyperbolic manifolds:

$$\mathcal{P} = \mathbb{S}^{s_1,K_1} \times \mathbb{S}^{s_2,K_2} \times \cdots \times \mathbb{S}^{s_n,K_n} \times \mathbb{H}^{h_1,K_1} \times \cdots \times \mathbb{H}^{h_m,K_m} \times \mathbb{R}^d \tag{10}$$

The total number of dimensions is $\sum_i^n s_i + \sum_j^m h_j + d$. Each individual manifold is called a component manifold, and the decomposition of the product manifold into component manifolds is called the signature. Informally, the signature can be considered a list of dimensionalities and curvatures for each component manifold.

Distances in $\mathcal{P}$ decompose as the $\ell_2$ norm of the distances in each of the component manifolds:

$$\delta_{\mathcal{P}}(\mathbf{u}, \mathbf{v}) = \sqrt{\sum_{\mathcal{M} \in \mathcal{P}} \delta_{\mathcal{M}}(\mathbf{u}_{\mathcal{M}}, \mathbf{v}_{\mathcal{M}})^2}, \tag{11}$$

where $\mathbf{u}_{\mathcal{M}}$ and $\mathbf{v}_{\mathcal{M}}$ denotes the restriction of $\mathbf{u}$ and $\mathbf{v}$ to their components in $\mathcal{M}$ and $\delta_{\mathcal{M}}$ refers the distance function appropriate to $\mathcal{M}$.

For $\mathbf{x} \in \mathcal{P}$, the tangent plane at $\mathbf{x}$, $T_{\mathbf{x}}\mathcal{P}$, is the concatenation (denoted by the direct sum $\bigoplus$) of all component tangent planes:

$$T_{\mathbf{x}}\mathcal{P} = \bigoplus_{\mathcal{M} \in \mathcal{P}} T_{\mathbf{x}_{\mathcal{M}}}\mathcal{M}. \tag{12}$$

We additionally define the origin of $\mathcal{P}$, $\mu_0$, as the concatenation of the origins of each respective manifold. The origin is $(1/|K|, 0, \ldots)$ for $\mathbb{H}^{D,K}$ and $\mathbb{S}^{D,K}$, and $(0, 0, \ldots)$ for $\mathbb{E}^D$.

## 2.2 DECISION TREES AND RANDOM FORESTS

The Classification and Regression Trees (CART) (Breiman, 2017) algorithm fits a DT $\mathcal{T}$ to a set of labeled data $(\mathbf{X}, \mathbf{y})$. Specifically, it greedily selects a split at each set to partition the dataset in such a way as to maximize the information gain,

$$\text{IG}(\mathbf{y}) = C(\mathbf{y}) - \frac{|\mathbf{y}^+|}{|\mathbf{y}|} C(\mathbf{y}^+) - \frac{|\mathbf{y}^-|}{|\mathbf{y}|} C(\mathbf{y}^-). \tag{13}$$

In this case, $C(\cdot)$ is some sort of impurity function (we use Gini impurity for classification and variance for regression). Some splitting function $S(\cdot)$ is used to partition the *labels* $\mathbf{y}$ into two classes, $\mathbf{y}^+$ and $\mathbf{y}^-$; however, $S(\cdot)$ also partitions the *input space* (corresponding to some $\mathbf{X}$ that does not appear in Eq. 13) into decision regions. Classically, $S(\cdot)$ is a thresholding function and thus breaks the input space into high-dimensional boxes given some dimension $d$ and threshold $\theta$:

$$S(\mathbf{x}) = \mathbb{I}\{x_d > \theta\}. \tag{14}$$

This algorithm is applied recursively to each decision region until a stopping condition is met (e.g., maximum number of splits is reached). The result is a fitted DT, $\mathcal{T}$, which can be used for inference. During inference, an unseen point $\mathbf{x}$ is passed through $\mathcal{T}$ until it reaches a leaf node corresponding to some decision region. For classification, the point is then assigned the majority label inside that region; for regression, it is assigned the mean value inside that region.

Finally, a RF is an ensemble of DTs, typically trained on a bootstrapped subsample of the points and features in $\mathbf{X}$ (Breiman, 2001).

### 2.2.1 HYPERBOLIC DECISION TREE ALGORITHMS

The hyperplane perspective on DTs is helpful background for understanding our method: mathematically, thresholding $\mathbf{x}$ on a dimension is equivalent to taking its dot product with the normal vector of a separating hyperplane $\mathbb{P}$, even in hyperbolic space. Although this is easy to compute for classical thresholding boundaries, which are zero in all dimensions but $d$, this perspective principally admits *any hyperplane* $\mathbb{P}$ as a valid decision boundary.

Naturally, considerations around choosing an appropriate (and computationally efficient) $\mathbb{P}$ abound. To this end, Chlenski et al. (2024) impose homogeneity and sparsity constraints on the hyperplanes they consider for hyperbolic DTs. In hyperbolic space, homogenous hyperplanes— hyperplanes that contain the origin *of the ambient space*—intersect $\mathbb{H}^{D,K}$ at geodesic submanifolds: that is, $\mathbb{P} \cap \mathbb{H}^{D,K}$ is closed under shortest paths *according to* $\delta_{\mathbb{H}}$. The sparsity constraint enforces that the normal vectors of $\mathbb{P}$ must be nonzero only in two positions: the timelike coordinate $x_0$ and some other $x_d$, which ensures that only $\mathcal{O}(nd)$ candidate hyperplanes are considered per split, and each decision can be computed in $\mathcal{O}(1)$ time using sparse dot products.

## 3 MIXED-CURVATURE DECISION TREES

For any DT, we must transform the input $\mathbf{X}$ into a set of candidate hyperplanes. To this end, we reframe and generalize the hyperplane approach of hyperbolic DTs. First, we observe that homogenous hyperplanes are geodesically convex in *any constant-curvature manifold*; therefore, we can extend the hyperbolic DT approach to $\mathbb{E}$ and $\mathbb{S}$. Second, we observe that fitting sparse, homogenous DTs is equivalent to thresholding on angles under 2-dimensional projections.

We consider the set of all projections onto the basis $\{x_0, x_d\}$, which can be computed in $\mathcal{O}(1)$ time per projection by coordinate selection. First, we compute the angles in each projection:[1]

$$\theta(\mathbf{x}, d) = \tan^{-1}(x_0/x_d). \tag{15}$$

Next, we use a modified splitting criterion to account for the geometry of angular splits:

$$S(\mathbf{x}, d, \theta) = \mathbb{I}\{\theta(\mathbf{x}, d) \in [\theta, \theta + \pi]\}. \tag{16}$$

---

[1]Note that, in our implementation, we use the PyTorch `arctan2` function to ensure that we can recover the full range of angles in $[0, 2\pi)$. This is essential for properly specifying decision boundaries in $\mathbb{S}$.

Once the best angle is selected, we must compute angular midpoints to select $\mathbb{P}$ that intersects $\mathcal{M}$ at a point *geodesically equidistant* the two points to either side of it (Euclidean DTs do this by sampling averaging the threshold values). Angular midpoints for each component manifold are described in the following sections and summarized in Table 4 in the Appendix.

With the angular features and manifold-informed midpoint modifications in place, the rest of the algorithm follows Section 2.2 unmodified.

## 3.1 EUCLIDEAN DECISION TREES

While the intersections of homogenous hyperplanes in $\mathbb{R}^D$ with $\mathbb{E}^D$ are (trivially) convex, these lack the expressiveness of an ambient-space formulation. Thus, we embed $\mathbb{E}^D$ in $\mathbb{R}^{D+1}$ by a trivial lift:

$$\phi : \mathbb{E}^D \to \mathbb{R}^{D+1}, \ \phi(\mathbf{u}) = (1, \mathbf{u}). \tag{17}$$

For two points $\mathbf{u}, \mathbf{v} \in \mathbb{E}^D$, the midpoint angles in $\mathbb{E}^D$ can be described in terms of the coordinates of $\mathbf{u}$ and $\mathbf{v}$ or their respective projection angles $(\theta_u, \theta_{\mathbf{v}})$ as

$$m_{\mathbb{E}}(\mathbf{u}, \mathbf{v}) = \tan^{-1}(2/(u_d + v_d)) \tag{18}$$

$$= \tan^{-1}\left( \frac{\tan^{-1}(\theta_{\mathbf{u}}) \tan^{-1}(\theta_{\mathbf{v}})}{\tan^{-1}(\theta_{\mathbf{u}}) + \tan^{-1}(\theta_{\mathbf{v}})} \right). \tag{19}$$

While this presentation of Euclidean DTs is unconventional, it is completely equivalent to thresholding in the basis dimensions. See Appendix C for the proof.

## 3.2 HYPERBOLIC DECISION TREES

For two points $\mathbf{u}, \mathbf{v} \in \mathbb{H}^{D,K}$, we compute $\theta_{\mathbf{u}}$ and $\theta_{\mathbf{v}}$ according to Eq 15 and follow Chlenski et al. (2024) in computing the hyperbolic midpoint angle in $\mathbb{H}^{D,K}$ as:

$$V := \frac{\sin(2\theta_{\mathbf{u}} - 2\theta_{\mathbf{v}})}{\sin(\theta_{\mathbf{u}} + \theta_{\mathbf{v}}) \sin(\theta_{\mathbf{v}} - \theta_{\mathbf{u}})}, \tag{20}$$

$$m_{\mathbb{H}}(\mathbf{u}, \mathbf{v}) = \begin{cases} \cot^{-1}(V - \sqrt{V^2 - 1}) & \text{if } \theta_{\mathbf{u}} + \theta_{\mathbf{v}} < \pi \\ \cot^{-1}(V + \sqrt{V^2 - 1}) & \text{otherwise.} \end{cases} \tag{21}$$

## 3.3 HYPERSPHERICAL DECISION TREES

The hyperspherical case is quite simple, except that unlike hyperbolic space and the "lifted" Euclidean space after applying Eq 17, we lack a natural choice of $x_0$. We adopt the convention of fixing the first dimension of the embedding space as $x_0$, which intuitively corresponds to fixing a "north pole" at the origin $\mu_0 = (1/|K|, 0, \ldots)$.

Angular midpoints are particularly well-behaved in hyperspherical manifolds: given $\mathbf{u}, \mathbf{v} \in \mathbb{S}^{D,K}$, the hyperspherical midpoint angle by finding $\theta_{\mathbf{u}}$ and $\theta_{\mathbf{v}}$ using Eq 15 and taking their mean:

$$m_{\mathbb{S}}(\mathbf{u}, \mathbf{v}) = (\theta_{\mathbf{u}} + \theta_{\mathbf{v}})/2. \tag{22}$$

## 3.4 PRODUCT DECISION TREE ALGORITHM

Intuitively, the transition from DTs in a single component manifold to a product manifold is that we now iterate over all preprocessed angles together, using the angular midpoint formula appropriate to each component. The complete pseudocode for this algorithm is in Appendix B.

Allowing for a single DT to span all components—as opposed to, e.g., an ensemble of DTs, each operating in a single component,—allows the model to independently allocate its splits across components according to their relevance to the task at hand. Recasting DT learning in terms of angular comparisons has three major advantages over finding planar decision boundaries directly:

1. We can consider angles under *arbitrary* linear projections (not just projections onto basis dimensions) while maintaining $\mathcal{O}(1)$ decision complexity. For instance, we can easily search over all $\binom{D}{2}$ 2-dimensional projections if we wish.

2. As there is no longer any need to enforce the constraints in Equations 5 and 8 at inference time, it becomes possible to subsample the features (precomputed angles) in RFs.
3. Product manifolds can always represent additional features in a new Euclidean manifold. For instance, this can be useful for incorporating metadata into DT training.

## 4 BENCHMARKS

We carried out benchmarks to evaluate which model, given a labeled set of mixed-curvature embeddings, achieves the lowest validation error. While we produced embeddings using a range of datasets and embedding techniques, our results focus only on performance on downstream tasks. We describe our data generation/embedding methods in more detail in the Appendix.

We summarize our benchmark results, with references to specific figures and tables, in Table 1. Our full results can be found in Table 5 in the Appendix.

Table 1: Benchmarks summary. "#Top-$k$" columns count how often product DTs or RFs were among the top $k$ predictors for a given set of benchmarks.

| Manifold type | Task | Reference | #Top-1 | #Top-2 | Total |
|---|---|---|---|---|---|
| Single-curvature | Classification | Figure 3 | 10 (91%) | 11 (100%) | 11 |
| Single-curvature | Regression | Figure 4 | 11 (100%) | 11 (100%) | 11 |
| Product manifold | Classification | Table 2 | 11 (46%) | 22 (92%) | 24 |
| Product manifold | Regression | Table 3 | 7 (64%) | 9 (82%) | 11 |
| **Total** | | | 39 (68%) | 53 (93%) | 57 |

### 4.1 EXPERIMENT DETAILS

**Problem setup.** Given a dataset $\mathbf{X}$, a set of labels $\mathbf{y}$, and a product manifold $\mathcal{P}$, we evaluate a variety of classifiers on their ability to predict $\mathbf{y}$ from $\mathbf{X}$. We apply an identical 80:20 train-test split to all of our data, train our models on the training set, and evaluate performance on the test set.

**Results reporting.** We report 95% confidence intervals for micro-averaged $F_1$ scores for classification and root mean squared error (RMSE) for regression benchmarks. Pairwise statistical significance is determined by the Wilcoxon signed-rank test comparing all same-type classifiers (i.e. trees to trees and forests to forests). We also apply a Bonferroni correction: starting with a critical value of .05, we divide by the total number of comparisons carried out *for a given signature*: since we compare 5 different models, our critical value becomes $.05/10 = .005$.

### 4.2 DATASETS

**Synthetic data.** We develop a novel method to sample mixtures of Gaussians in $\mathcal{P}$ to generate classification and regression datasets. For classification, we generate 8 classes using 32 clusters. For regression, we generate a single scalar response variable using 32 clusters with randomly-generated intercepts. Our full method is described in Appendix Section A

**Graph embeddings.** For classification and regression on graph datasets, we generate embeddings that approximate shortest-path distances in the graph using the method described in Gu et al. (2018). We select the optimal signature from the candidate set $\{(\mathbb{H}^2)^2, \mathbb{H}^2\mathbb{E}^2, \mathbb{H}^2\mathbb{S}^2, \mathbb{S}^2\mathbb{E}^2, (\mathbb{S}^2)^2, \mathbb{H}^4, \mathbb{E}^4, \mathbb{S}^4\}$ by generating embeddings in each signature and selecting the signature with the lowest metric distortion. For link prediction, we embed all datasets in $\mathcal{P} = (\mathbb{S}^2\mathbb{E}^2\mathbb{H}^2)$, then create a binary classification dataset by associating each pair of nodes with a point in $\mathcal{P}^2\mathbb{E}^1$, where each pair of points is included and the last Euclidean dimension is the manifold distance $\delta_{\mathcal{P}}(\mathbf{x_i}, \mathbf{x_j})$; labels are simply whether there is an edge between nodes $i$ and $j$. Full details on graph embeddings are described in Appendix Section E.2.

**Mixed-curvature VAE latent space.** We follow Skopek et al. (2020) in training variational autoencoders (VAEs) whose latent space is $\mathcal{P}$. Once the VAE is trained, we use its encoder to generate

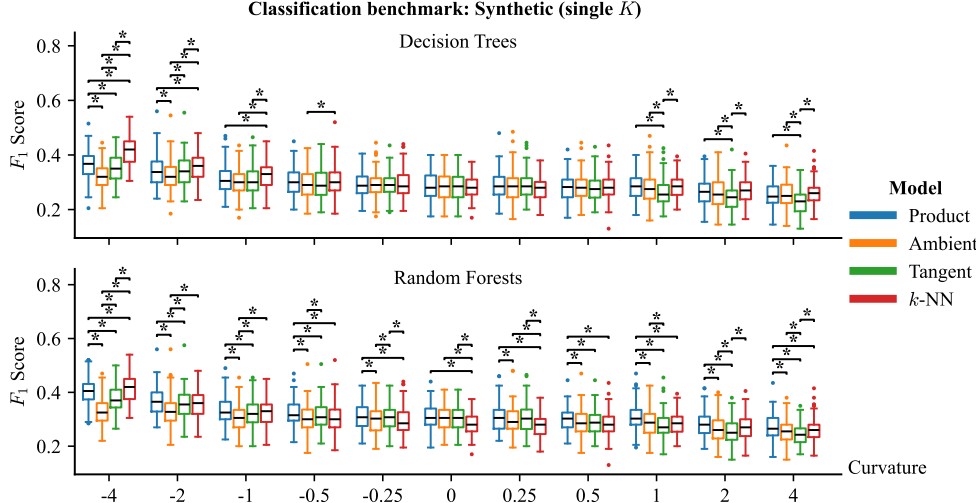

Figure 3: Classification benchmark comparison of DTs (top) and RFs (bottom). We report micro-averaged $F_1$ scores on a synthetic data classification task involving mixtures of 8 Gaussians in manifolds of varying constant curvatures $K$. We compare DTs and RFs in the **product manifold**, **the ambient space**, and **the tangent plane**, along with *k*-**nearest neighbors** on distances in $\mathcal{P}$. Statistical significance (Bonferroni-corrected $p < 0.05$) is marked with an asterisk (*). We omit product space perceptrons, which never achieved competitive results.

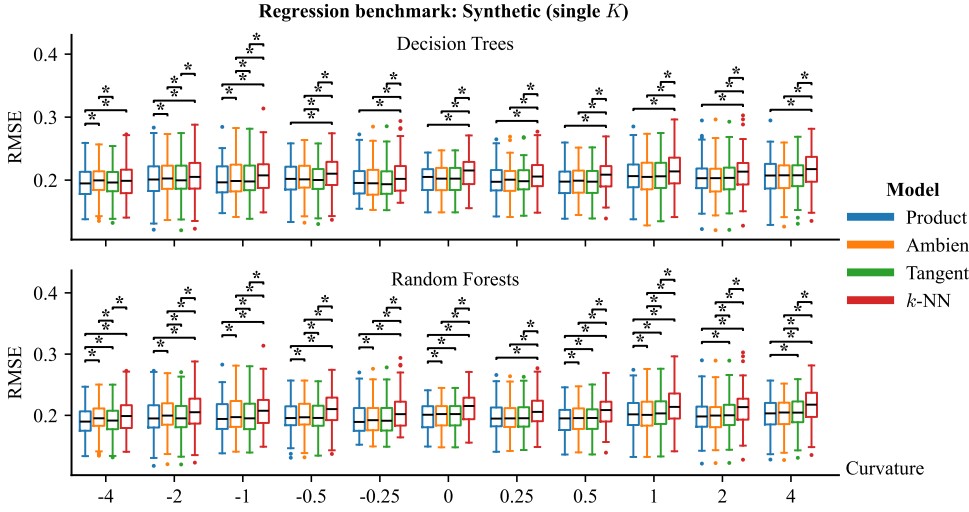

Figure 4: Regression benchmarks (RMSE) for single-curvature manifolds. We follow the conventions of Figure 3 and mark Bonferroni-corrected significance with an asterisk (*).

embeddings for our dataset and classify these embeddings. Full details on VAE training and downstream inference are described in Appendix Section E.3.

**Empirical datasets.** Some datasets can be represented in a non-Euclidean geometry without generating embeddings: for instance, geospatial data lives in $\mathbb{S}^2$, while cyclic time series embed in $\mathbb{S}^1$. We describe our approach to generating embeddings for these empirical datasets in Appendix Section E.4.

Table 2: $F_1$ scores for all classification and link prediction (LP) benchmarks. Best predictors are shown in **bold**, while second-best predictors are underlined. For brevity, we omit columns for low-performing methods and merge DT and RF columns (e.g. "Ambient" means $\max(\mathrm{mean(Ambient\ DT)}, \mathrm{mean(Ambient\ RF)})$.)

| | Dataset | Signature | $k$-NN | Ambient | Product |
|---|---|---|---|---|---|
| Synthetic (multi-$K$) | Gaussian | $(\mathbb{S}^2)^2$ | **35.7±.5** | 32.3±.5 | 33.1±.5 |
| | | $\mathbb{E}^4$ | **34.9±.5** | 30.0±.4 | 31.3±.4 |
| | | $\mathbb{H}^2\mathbb{E}^2$ | **40.3±.4** | 34.7±.5 | 36.1±.5 |
| | | $\mathbb{H}^2\mathbb{S}^2$ | **38.4±.5** | 33.3±.5 | 35.2±.5 |
| | | $\mathbb{H}^4$ | **47.5±.5** | 35.9±.4 | 40.0±.4 |
| | | $\mathbb{S}^2\mathbb{E}^2$ | **35.8±.5** | 32.7±.5 | 33.5±.4 |
| | | $\mathbb{S}^4$ | **33.1±.4** | 27.6±.4 | 28.0±.5 |
| | | $(\mathbb{H}^2)^2$ | **41.5±.5** | 34.5±.5 | 37.0±.5 |
| Graph embeddings | CiteSeer | $\mathbb{H}^2\mathbb{S}^2$ | 25.9±.5 | **26.1±.7** | 25.8±.6 |
| | Cora | $\mathbb{H}^4$ | 20.7±.4 | **28.9±.5** | **28.9±.4** |
| | PolBlogs | $(\mathbb{S}^2)^2$ | **93.5±.4** | 93.2±.5 | 92.9±.4 |
| | AdjNoun (LP) | $(\mathbb{S}^2\mathbb{E}^2\mathbb{H}^2)^2\mathbb{E}^1$ | 93.3±1.1 | **93.7±1.1** | **93.7±1.1** |
| | Dolphins (LP) | $(\mathbb{S}^2\mathbb{E}^2\mathbb{H}^2)^2\mathbb{E}^1$ | **96.6±.3** | 92.3±.9 | **96.6±.3** |
| | Football (LP) | $(\mathbb{S}^2\mathbb{E}^2\mathbb{H}^2)^2\mathbb{E}^1$ | 79.8±3.3 | **85.7±3.6** | **85.7±3.6** |
| | Karate Club (LP) | $(\mathbb{S}^2\mathbb{E}^2\mathbb{H}^2)^2\mathbb{E}^1$ | **95.1±1.5** | 88.6±2.2 | **95.1±1.5** |
| | Les Mis (LP) | $(\mathbb{S}^2\mathbb{E}^2\mathbb{H}^2)^2\mathbb{E}^1$ | **95.7±.7** | 92.7±.9 | 95.6±.8 |
| | PolBooks (LP) | $(\mathbb{S}^2\mathbb{E}^2\mathbb{H}^2)^2\mathbb{E}^1$ | **95.8±.4** | 92.9±.6 | **95.8±.4** |
| VAE | Blood | $\mathbb{S}^2\mathbb{E}^2(\mathbb{H}^2)^3$ | 17.4±.5 | 19.3±.5 | **20.1±.5** |
| | CIFAR-100 | $(\mathbb{S}^2)^4$ | 8.6±.4 | 11.5±.5 | **12.0±.3** |
| | Lymphoma | $(\mathbb{S}^2)^2$ | 77.8±1.4 | 81.7±1.2 | **83.7±1.2** |
| | MNIST | $\mathbb{S}^2\mathbb{E}^2\mathbb{H}^2$ | **41.9±3.7** | 35.7±2.8 | 39.4±2.3 |
| Other | Landmasses | $\mathbb{S}^2$ | **91.4±.2** | 83.5±.3 | 84.2±.3 |
| | Neuron 33 | $(\mathbb{S}^1)^5$ | 50.5±.5 | 76.2±.4 | **77.0±.4** |
| | Neuron 46 | $(\mathbb{S}^1)^5$ | 50.2±.2 | 61.1±.3 | **61.2±.3** |

Table 3: Regression results (RMSE) for all benchmarks. We follow the conventions of Table 2. CS PhDs is a graph embedding dataset, whereas Temperature and Traffic are empirical.

| Dataset | Signature | $k$-Neighbors | Ambient | Product |
|---|---|---|---|---|
| Synthetic (multi-$K$) | $(\mathbb{S}^2)^2$ | .196±.002 | **.191±.002** | **.191±.002** |
| | $\mathbb{E}^4$ | .194±.003 | .191±.002 | **.190±.002** |
| | $\mathbb{H}^2\mathbb{E}^2$ | .196±.003 | .194±.002 | **.193±.002** |
| | $\mathbb{H}^2\mathbb{S}^2$ | .197±.003 | .194±.002 | **.193±.002** |
| | $\mathbb{H}^4$ | **.175±.003** | .184±.003 | .178±.003 |
| | $\mathbb{S}^2\mathbb{E}^2$ | .199±.003 | **.194±.002** | **.194±.002** |
| | $\mathbb{S}^4$ | .194±.002 | **.188±.002** | .189±.002 |
| | $(\mathbb{H}^2)^2$ | .193±.003 | .193±.002 | **.191±.002** |
| CS PhDs | $\mathbb{H}^4$ | .053±.005 | .052±.005 | **.041±.004** |
| Temperature | $\mathbb{S}^2\mathbb{S}^1$ | 7.198±.212 | **4.531±.187** | 7.130±.123 |
| Traffic | $\mathbb{E}^1(\mathbb{S}^1)^4$ | .510±.003 | **.505±.003** | .534±.003 |

## 4.3 BASELINES

We use Scikit-Learn (Pedregosa et al., 2011) DTs and RFs in both the ambient space $\mathbb{R}^{D+1}$ and the tangent plane $\mathcal{T}_{\mu_0}\mathcal{P}$ as baselines. Ambient space models operate directly on ambient space coordinates. Tangent plane models project points from $\mathcal{P}$ to $\mathcal{T}_{\mu_0}\mathcal{P}$ by applying the logarithmic map at $\mu_0$ as a preprocessing step. We use Scikit-Learn $k$-nearest neighbor ($k$-NN) classifiers and regressors with precomputed pairwise distance matrices according to $\delta_{\mathcal{P}}$ (Eq. 11). Finally, we implemented the product space perceptron algorithm described in Tabaghi et al. (2021). For our own models, we set hyperparameters identically to Scikit-Learn DTs and RFs, except we consider all $\binom{D}{2}$ projections—for a total of 3 features per 2-dimensional component manifold, just like ambient space methods use. Full details for each model can be found in Appendix E.1.

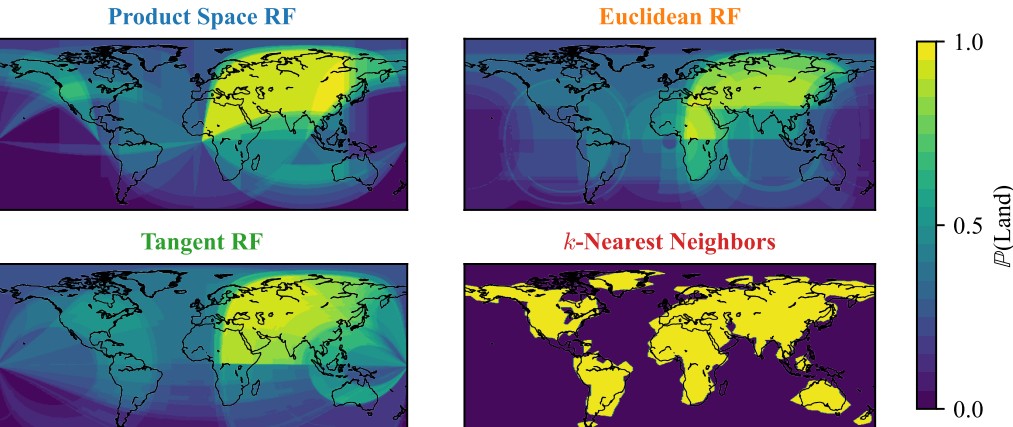

Figure 5: We color a world map with each model's predicted $\mathbb{P}(\text{Land})$ for the "Landmasses" dataset, a land vs. water classification benchmark in $\mathbb{S}^2$. Each RF consists of 12 DTs with a max depth of 3. Note the artifacts learned by **Euclidean**, **tangent** RFs, and **$k$-NN** models.

## 4.4 RESULTS

For single-curvature synthetic datasets, our method was the best classifier in 10 out of 11 signatures (Figure 3) and the best regressor (Figure 4 for all signatures. In Tables 2 and 3, we demonstrate consistently good performance across a diverse range of benchmarks.

Further experiments can be found in the Appendix: we provide ablations in F, detailed tables and latent space visualizations in G, comparisons to MLP and GNN models in I, runtime and computational complexity analysis in J, and interpretability experiments in K.

## 5 CONCLUSION

We present strong preliminary evidence favoring mixed-curvature DTs and RFs. In particular, we motivate and describe our entire algorithm and demonstrate its effectiveness across a highly diverse set of 57 benchmarks covering a variety of tasks and geometries.

Product manifold DTs and RFs offer a valuable balance of expressiveness and simplicity, positioned between extremely legible but underpowered linear classifiers and powerful but uninterpretable neural networks operating in product manifolds. We believe that these qualities, combined with their demonstrated performance across our benchmark datasets, are compelling evidence of our method's usefulness in a non-Euclidean data analysis toolkit.

**Limitations.** While we view our work as downstream of signature selection and embedding generation, its value heavily depends on the availability of good product manifold embeddings. There are challenges in selecting appropriate signatures (Borde et al., 2023a), and product manifolds are not able to represent all patterns in data (Borde & Kratsios, 2023). Furthermore, it is computationally intensive to generate of embeddings. There are also tradeoffs between DTs and RFs and other high-performing methods, especially graph neural networks when topologies are known. The complexity of working with non-Euclidean data could pose a potential barrier to adoption. Finally, the lack of a privileged basis (Elhage et al., 2023) in non-Euclidean embeddings makes the inductive bias of decision trees less well-motivated.

**Future work.** It may be possible to exploit non-privileged basis dimensions using approaches such as rotation forests (Bagnall et al., 2020), random 2-D subspace angles, or oblique decision trees. A continuous unification, such as the $\kappa$-stereographic model described in (Skopek et al., 2020), may be more robust and elegant. Extensions to simplex geometry (Mishra et al., 2020) are also worth considering.

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

# A  GAUSSIAN MIXTURE DETAILS

## A.1  OVERALL STRUCTURE

The structure of our sampling algorithm is as follows. Note that, rather than letting $\mathcal{M}$ be a manifold of arbitrary curvature, we force its curvature to be one of $\{-1, 0, 1\}$ for implementation reasons. This necessitates rescaling steps, which take place in Equations 29, 33, and 39. The result is equivalent to performing the equivalent steps, without rescaling, on a manifold of the proper curvature.

1. Generate $\mathbf{c}$, a vector that divides $m$ samples into $n$ clusters:

$$\mathbf{p_{raw}} = \langle p_0, p_1, \ldots, p_{n-1} \rangle \tag{23}$$

$$p_i \sim \text{Uniform}(0, 1) \tag{24}$$

$$\mathbf{p_{norm}} = \frac{\mathbf{p_{raw}}}{\sum_{i=0}^{n-1} p_i} \tag{25}$$

$$\mathbf{c} = \langle c_0, c_1, \ldots c_{m-1} \rangle \tag{26}$$

$$c_i \sim \text{Categorical}(n, \mathbf{p_{norm}}) \tag{27}$$

2. Sample $\mathbf{M_{euc}}$, an $n \times D$ matrix of $n$ class means:

$$\mathbf{M_{euc}} = \langle \mathbf{m_0}, \mathbf{m_1}, \ldots, \mathbf{m_{n-1}} \rangle^T \tag{28}$$

$$\mathbf{m_i} \sim \mathcal{N}(0, \sqrt{K}\mathbf{I}). \tag{29}$$

3. Move $\mathbf{M_{euc}}$ into $T_0\mathcal{M}$, the tangent plane at the origin of $\mathcal{M}$, by applying $\psi : \mathbf{x} \to (0, \mathbf{x})$ per-row to $\mathbf{M_{euc}}$:

$$\mathbf{M_{tan}} = \langle \psi(\mathbf{m_0}), \psi(\mathbf{m_1}), \ldots \psi(\mathbf{m_{n-1}}) \rangle^T, \tag{30}$$

$$\psi : \mathbb{R}^D \to \mathbb{R}^{D+1}, \ \mathbf{x} \to \langle 0, \mathbf{x} \rangle. \tag{31}$$

4. Project $\mathbf{M_{tan}}$ onto $\mathcal{M}$ using the exponential map from $T_0\mathcal{M}$ to $\mathbf{M_{tan}}$:

$$\mathbf{M} = \exp_0(\mathbf{M_{tan}}). \tag{32}$$

5. For $0 \leq i < n$, sample a corresponding covariance matrix. Here, $\sigma$ is a variance scale parameter that can be set:

$$\mathbf{\Sigma_i} \sim \text{Wishart}(\sigma\sqrt{K}\mathbf{I}, D) \tag{33}$$

6. For $0 \leq j < m$, sample $\mathbf{X_{euc}}$, a matrix of $m$ points according to their clusters' covariance matrices:

$$\mathbf{X_{euc}} = \langle \mathbf{x_0}, \mathbf{x_1}, \ldots \mathbf{x_{m-1}} \rangle^T \tag{34}$$

$$x_j \sim \mathcal{N}(0, \mathbf{\Sigma_{c_j}}). \tag{35}$$

7. Apply $\psi(\cdot)$ from Eq 31 to each $\mathbf{x_j}$ to move it into $T_0\mathcal{M}$:

$$\mathbf{X_{tan}} = \langle \psi(\mathbf{x_0}), \psi(\mathbf{x_1}), \ldots \psi(\mathbf{x_{m-1}}) \rangle^T. \tag{36}$$

8. For each row in $\mathbf{X_{tan}}$, apply parallel transport from $T_0\mathcal{M}$ to its class mean:

$$\mathbf{X_{PT}} = \langle \mathbf{x_{0,\mu}}, \mathbf{x_{1,\mu}}, \ldots, \mathbf{x_{m-1,\mu}} \rangle \tag{37}$$

$$\mathbf{x_{j,\mu}} = PT_{0 \to \mathbf{m_{c_j}}}(\mathbf{x_j}) \tag{38}$$

9. Use the exponential map at $T_\mu\mathcal{M}$ to move the points onto the manifold:

$$\mathbf{X_{\mathcal{M}}} = \langle \mathbf{x_{0,\mathcal{M}}}, \mathbf{x_{1,\mathcal{M}}}, \ldots, \mathbf{x_{m-1,\mathcal{M}}} \rangle \tag{39}$$

$$\mathbf{x_{j,\mathcal{M}}} = \frac{\exp_{\mathbf{m_{c_j}}}(\mathbf{x_{j,\mu}})}{\sqrt{K}} \tag{40}$$

10. Repeat steps 2–9 for as many manifolds as desired; produce a final embedding by concatenating all component embeddings column-wise:

$$\mathbf{X} = \langle \mathbf{X_{\mathcal{M}_0}}, \mathbf{X_{\mathcal{M}_1}}, \ldots \mathbf{X_{\mathcal{M}_p}} \rangle \tag{41}$$

## A.2 EQUATIONS FOR MANIFOLD OPERATIONS

First, we provide the forms of the parallel transport operation in hyperbolic, hyperspherical, and Euclidean spaces:

$$\text{PT}^{\mathbb{H}}_{\nu\to\mu}(\mathbf{v}) = \mathbf{v} + \frac{\langle \mu - \alpha\nu, \nu \rangle_{\mathcal{L}}}{\alpha + 1}(\nu + \mu) \tag{42}$$

$$\alpha = -\langle \nu, \mu \rangle_{\mathcal{L}} \tag{43}$$

$$\text{PT}^{\mathbb{S}}_{\nu\to\mu}(\mathbf{v}) = \mathbf{v}\cos(d) + \frac{\sin(d)}{d}(\mu - \cos(d)\nu) \tag{44}$$

$$d = \cos^{-1}(\nu \cdot \mu) \tag{45}$$

$$\text{PT}^{\mathbb{E}}_{\nu\to\mu}(\mathbf{v}) = \mathbf{v} + \mu - \nu. \tag{46}$$

The exponential map is defined as follows in each of the three spaces:

$$\exp^{\mathbb{H}}_{\mu}(\mathbf{u}) = \cosh(\|\mathbf{u}\|_{\mathcal{L}})\mu + \sinh(\|\mathbf{u}\|_{\mathcal{L}})\frac{\mathbf{u}}{\|\mathbf{u}\|_{\mathcal{L}}} \tag{47}$$

$$\exp^{\mathbb{S}}_{\mu}(\mathbf{u}) = \cos(\|\mathbf{u}\|)\mu + \sin(\|\mathbf{u}\|)\frac{\mathbf{u}}{\|\mathbf{u}\|} \tag{48}$$

$$\exp^{\mathbb{E}}_{\mu}(\mathbf{u}) = \mathbf{u}. \tag{49}$$

## A.3 GENERATING CLASSIFICATION TARGETS

To generate classification targets covering $p \leq n$ classes, all we need to do is map clusters to classes. To ensure that each class has at least one associated cluster, we arbitrarily assign the first $p$ clusters to the first $p$ classes. In the $p = n$ case, this is equal to the $p$-dimensional identity matrix, and we conclude. In the $p < n$ case, we assign the remaining $n - p$ by drawing assignments from a uniform categorical distribution over the $p$ classes.

## A.4 GENERATING REGRESSION TARGETS

To generate regression targets, we draw per-cluster slopes and intercepts:

$$\beta_{i,k} \sim Uniform(-1, 1) \tag{50}$$

$$\alpha_i \sim Uniform(-10, 10 \tag{51}$$

We then multiply each $x_j \in \mathbf{X_e uc}$ (i.e. the pre-transport samples from the normal distribution) by $\beta$ and add $\alpha$:

$$y_j = \mathbf{x}_j \beta + \alpha + \varepsilon \tag{52}$$

To make the regression task more constrained and, therefore, to make the RMSEs across samples more comparable, we further normalize the labels to the range $[0, 1]$ by subtracting the minimum $y$ value and dividing by the range.

## A.5 RELATIONSHIP TO OTHER WORK

Nagano et al. (2019) developed the overall technique used for a single cluster and a single manifold, i.e. steps 6–9. Chlenski et al. (2024) modified this method to work for mixtures of Gaussians in $\mathbb{H}^{d,1}$, and deployed it for $d \in 2, 4, 8, 16$. This corresponds to steps 1–5 of our procedure (although note that our covariance matrices are sampled differently in step 5). Thus, our contribution is simply to add step 10, modify step 5 to use the Wishart distribution, to add curvature-related scaling factors in Equations 29, 33, and 39, and to generate classification and regression targets as described in the preceding sections.

We apply this to *hyperspherical* manifolds, for which the von Mises-Fisher (VMF) distribution is typically preferred. This is an unconventional choice, but has been employed previously by Skopek et al. (2020) in their mixed-curvature VAE formulation. We do not argue for the superiority of our approach over the VMF distribution in general; however, we prefer to use ours for these benchmarks, as it allows us to draw simpler parallels between manifolds of different curvatures.

# B  PRODUCT SPACE DECISION TREE PSEUDOCODE

---

**Algorithm 1** Product Space Decision Tree

---

1: **Procedure** FIT:
2:     $\mathcal{P}$        (signature of) product manifold
3:     $\mathbf{X}$        data points
4:     $\mathbf{y}$        target labels
5: **Initialize:**
6:     $\mathcal{T}$        an empty tree
7: **return**  FITTREE($\mathbf{X}, \mathbf{y}, 0$)
8:
9: **Procedure** FITTREE:
10:     $\mathbf{X}$        data points
11:     $\mathbf{y}$        target labels
12:     $t$        current depth of the tree.
13: **Initialize:**
14:     $d_{\text{best}}$    dimension of best split,
15:     $\theta_{\text{best}}$    angle of best split,
16:     $IG_{\text{best}}$  information gain of best split.
17: **for** each $d \in \mathbf{D}'$ **do**
18:     $\mathcal{M} \leftarrow$ component manifold for dimension $d$
19:     $\Theta \leftarrow$ GETCANDIDATES($\mathcal{M}, \mathbf{X}, d$)
20:     **for** each candidate $\theta \in \Theta$ **do**
21:         Partition $\mathbf{X}, \mathbf{y}$ into $\mathbf{X}^+, \mathbf{X}^-, \mathbf{y}^+, \mathbf{y}^-$ via Eq. 16.
22:         Apply Eq. 13 on $\mathbf{y}^+, \mathbf{y}^-$ to compute $IG_{\text{current}}$
23:         **if** $IG_{\text{current}} > IG_{\text{best}}$ **then**
24:             $d_{\text{best}}, \theta_{\text{best}}, IG_{\text{best}} \leftarrow d, \theta, IG_{\text{current}}$
25:         **end if**
26:     **end for**
27: **end for**
28: **if** no valid split was found **then**
29:     **return**  $\mathcal{N}$, a new **leaf node** with $\mathbf{y}$ probabilities.
30: **else**
31:     Create $\mathcal{N}$, a decision node with $d_{\text{best}}$ and $\theta_{\text{best}}$
32:     $\mathcal{N}_L \leftarrow$ FITTREE($\mathbf{X}^-, \mathbf{y}^-, t+1$)
33:     $\mathcal{N}_R \leftarrow$ FITTREE($\mathbf{X}^+, \mathbf{y}^+, t+1$)
34:     **return**  $\mathcal{N}$ with left child $\mathcal{N}_L$ and right child $\mathcal{N}_R$
35: **end if**
36:
37: **Procedure** GETCANDIDATES:
38:     $\mathcal{M}$     A component manifold
39:     $\mathbf{X}$     A dataset of points in $\mathcal{M}$
40:     $d$     A dimension index
41: **if** $d$ is the special dimension **then**
42:     **return**  empty array []
43: **end if**
44: $\Theta \leftarrow$ Angles of $\mathbf{X}$ via Eq. 15
45: $\Theta \leftarrow$ sort and deduplicate $\Theta$
46: **return**  [$\theta_m$ for $\theta_i, \theta_{i+1} \in \Theta$ via Eq. 18, 20, or 22] (depending on curvature of $\mathcal{M}$).

---

## C PROOF OF EQUIVALENCE FOR EUCLIDEAN CASE

A classical CART tree splits data points according to whether their value in a given dimension is greater than or less than some threshold value $t$. Midpoints are simple arithmetic means. This can be written as:

$$S'(\mathbf{x}, d, t) = \begin{cases} 1 \text{ if } x_d > t, \\ 0 \text{ otherwise.} \end{cases} \tag{53}$$

$$m_{DT}(\mathbf{u}, \mathbf{v}) = \frac{u_d + v_d}{2}. \tag{54}$$

In our transformed DT, we lift the data points by applying $\phi : \mathbf{x} \to (1, \mathbf{x})$ and then check which side of an axis-inclined hyperplane they fall on. The splitting function is based on the angle $\theta$ of inclination with respect to the $(0, d)$ plane, i.e., $\langle 1, x_d \rangle$. Our midpoints are computed to ensure equidistance in the original manifold:

$$S(\mathbf{x}, d, \theta) = \text{sign}(\sin(\theta)x_d - \cos(\theta)x_0) \tag{55}$$

$$m_{\mathbb{E}}(\mathbf{u}, \mathbf{v}) = \tan^{-1}\left(\frac{2}{u_d + v_d}\right) \tag{56}$$

To demonstrate the equivalence of the classical DT formulation to our transformed algorithm in $\mathbb{E}$, we will show that Equation 53 is equivalent to Equation 55 and Equation 54 is equivalent to Equation 56 under

$$\theta = \cot^{-1}(t). \tag{57}$$

### C.1 EQUIVALENCE OF SPLITS

First, we show that Equations 53 and 55 are equivalent, assuming $t \neq 0$:

$$S(\mathbf{x}, d, \theta) = \text{sign}(\sin(\theta)x_d - \cos(\theta)x_0) = 1 \tag{58}$$

$$\iff \sin(\theta)x_d - \cos(\theta) > 0 \tag{59}$$

$$\iff \frac{\sin(\theta)}{\cos(\theta)}x_d = \tan(\theta)x_d > 1 \tag{60}$$

$$\iff x_d/t > 1 \tag{61}$$

$$\iff x_d > t \tag{62}$$

$$\iff S'(\mathbf{x}, d, t) = 1 \tag{63}$$

### C.2 EQUIVALENCE OF MIDPOINTS

Now, we show that Equations 54 and 56 are equivalent:

$$\cot^{-1}(m_{DT}(\mathbf{u}, \mathbf{v})) = \cot^{-1}\left(\frac{u_d + v_d}{2}\right) \tag{64}$$

$$= \tan^{-1}\left(\frac{2}{u_d - v_d}\right) \tag{65}$$

$$= m_{\mathbb{E}}(\mathbf{u}, \mathbf{v}) \tag{66}$$

# D SUMMARY OF ANGULAR MIDPOINT FORMULAS

Table 4: Distance functions and midpoint angle formulas for each component manifold type.

| Manifold $\mathcal{M}$ | Distance $\delta_{\mathcal{M}}(\mathbf{u}, \mathbf{v})$ | Midpoint angle $\theta_{\mathcal{M}}(\mathbf{u}, \mathbf{v})$ |
|---|---|---|
| $\mathbb{S}^{D,K}$ | $\cos^{-1}\left(\dfrac{K^2\langle\mathbf{u}, \mathbf{v}\rangle}{K}\right)$ | $\dfrac{\theta_{\mathbf{u}} + \theta_{\mathbf{v}}}{2}$ |
| $\mathbb{E}^{D,0}$ | $\sqrt{\langle\mathbf{u}, \mathbf{v}\rangle}$ | $\tan^{-1}\left(\dfrac{2}{u_d + v_d}\right)$ |
| $\mathbb{H}^{D,K}$ | $\dfrac{-\cosh^{-1}(K^2\langle\mathbf{u}, \mathbf{v}\rangle_{\mathcal{L}})}{K}$ | $\cot^{-1}(V - \sqrt{V^2 - 1})$ if $\theta_{\mathbf{u}} + \theta_{\mathbf{v}} < \pi,$ $\cot^{-1}(V + \sqrt{V^2 - 1})$ otherwise. $V := \dfrac{\sin(2\theta_{\mathbf{u}} - 2\theta_{\mathbf{v}})}{2\sin(\theta_{\mathbf{u}} + \theta_{\mathbf{v}})\sin(\theta_{\mathbf{v}} - \theta_{\mathbf{u}})}$ |

# E FULL EXPERIMENTAL HYPERPARAMETERS

## E.1 SCIKIT-LEARN HYPERPARAMETERS

**Random forests and decision trees.** For fairness, we set all DT and RF hyperparameters identically. Specifically, we set the following hyperparameters for both DTs and RFs:

- max_depth = 5
- min_samples_split = 2
- min_samples_leaf = 1
- min_impurity_decrease = 0.0

For RFs, we also set the following hyperparameters:

- n_estimators = 12
- max_features = "sqrt"
- bootstrap = True (subsamples the training data)
- max_samples = None (draws $n$ samples from a set of $n$ points)

Because the scikit-learn implementation differs substantially from ours, subsamples vary even when the random seed is set. Nevertheless, we also employ the same random seed for all RF models.

$k$**-nearest neighbor models.** For $k$-nearest neighbors, we use default hyperparameters.

**Product space perceptrons and SVMs.** Product space perceptrons only have one hyperparameter, which is the relative weight assigned to each component manifold. We elect to give each component manifold equal weight.

Neither the SVM code provided by Tabaghi et al. (2021) nor our own reimplementation would run on our datasets. In particular, we had issues satisfying the convexity constraints described in their paper, causing the solve to crash. Correcting this mistake and augmenting our benchmarks with SVM evaluations is a direction for future research.

**Product space decision trees and random forests.** For our models, we set the n_features = "n_choose_2" parameter. This means that we consider all $\binom{n}{2}$ linear projections. We do this because we restrict ourselves to 2-dimensional component manifolds, and therefore we only observe $\binom{3}{2} = 3$ total angles, equal to the number of features used by ambient space Euclidean methods.

## E.2 GRAPH EMBEDDINGS

**Learning embeddings.** We reimplement the method in Gu et al. (2018) to learn graph embeddings. In particular, we use the NetworkX package (Hagberg et al., 2008) to load the graph, extract

the largest connected component, and compute pairwise distances between nodes using the Floyd-Warshall algorithm. For embedding purposes, we treat all graphs as undirected. Pairwise distances were normalized into the range $[0, 1]$ by dividing by the maximum distance.

**Embedding hyperparameters.** Embeddings were learned using Riemannian Adam (Becigneul & Ganea, 2018) implemented in Geoopt (Kochurov et al., 2020). For each signature, we train 10 randomly-initialized embeddings for 3,000 epochs each. We treat the first 300 epochs as a burn-in period, during which the learning rate is .01 and the curvature of each manifold is fixed. For the remaining epochs, we train embedding coordinates with a learning rate of 0.1 and scale factors with a learning rate of 0.01. These hyperparameters were chosen based on their stability and convergence in exploratory experiments.

**Train-test split.** Because embeddings must be learned per-node, it was not possible to perform a train-test split prior to the embedding step; however, we performed the train-test split at the node level for all tasks including link prediction. This means that we discarded all edges between test and training nodes from our dataset. While we acknowledge the embeddings step could be a source of leakage, we have no reason to believe this would bias evaluations in favor of any particular model. Future work should focus on developing methods to learn node embeddings in phases or to use masked gradients to minimize leakage at the embedding step.

**Evaluations.** Since it was not clear *a priori* which signature would embed each graph the best, we learned 10 embeddings for each candidate signature and took the one with the best $D_{avg}$ to be the benchmark signature. Our reasoning is that the lowest-distortion embedding of the graph is the most appropriate benchmark for evaluating the geometrical appropriateness of a classifier. Thus, scores for the lowest-distortion signature appear in Tables 2 and 3, whereas scores for all signatures can be found in Table 5.

**Link prediction.** To generate link prediction datasets, we trained 100 randomly initialized sets of node embeddings in $\mathbb{S}^2 \times \mathbb{E}^2 \times \mathbb{H}^2$. If we let $\mathbf{X}$ be our original node embeddings and $\mathcal{E}$ be the ground-truth edges of the graph, we then generated the following dataset:

$$\mathbf{X}_{LP} = \{\langle \mathbf{x}_i, \mathbf{x}_j \rangle \text{ for } (\mathbf{x}_i, \mathbf{x}_j, \delta_{\mathcal{P}}(\mathbf{x}_i, \mathbf{x}_j) \in \mathbf{X}\} \tag{67}$$

$$\mathbf{y}_{LP} = \{\mathbb{I}\{(\mathbf{x}_i, \mathbf{x}_j) \in \mathcal{E}\} \text{ for } (\mathbf{x}_i, \mathbf{x}_j) \in \mathbf{X}\} \tag{68}$$

The corresponding signature is $(\mathcal{P})^2 \times \mathbb{E}^1$; in the case of our embeddings, that is $(\mathbb{S}^2 \times \mathbb{E}^2 \times \mathbb{H}^2)^2 \times \mathbb{E}^1$.

### E.3  VAE TRAINING

**Encoder/decoder architectures.** Following Tabaghi et al. (2021), we use the following encoder/decoder architectures:

- Lymphoma dataset: Two 200-dimensional hidden layers, 500 epochs
- Blood cell dataset: Three 400-dimensional hidden layers, 200 epochs
- Omniglot and MNIST: 400-dimensional latent
- CIFAR-100: $4 \times 4$ convolutional kernels with stride 2 and padding 1. Encoder: 3 CNN layers of 64, 128, and 512 channels. Decoder: 2048-dimensional dense layer, followed by 2 CNN layers of 256, 64, and 3 channels.

**Training hyperparameters.** Our VAEs were trained using the Adam optimizer (Kingma & Ba, 2017) with default parameters (learning rate .001, $\beta_1 = 0.9$, $\beta_2 = 0.999$. In all models, each layer except the last is followed by a ReLU activation function. Curvatures were trained identically, except using a learning rate of .0001, after 100 burn-in epochs. Because some training details were omitted from the original papers, we additionally chose the following hyperparameters:

- Batch size: 4,096
- Number of samples per point: 64
- $\beta$ (weight for KL-divergence in VAE loss): 1

**Train-test split.** To minimize the risk of data leakage, we trained our VAEs on only the training data, then used the trained VAEs to generate embeddings for the training and test data. Embeddings were generated by running points through the VAE encoder and taking the returned mean parameter.

**Evaluations.** To conserve memory, we randomly subsampled 1,000 points from the training and test sets for each evaluation. We ran 10 trials per dataset in total.

### E.4 Empirical Datasets

**Landmasses.** We generated a geospatial classification dataset for land versus water prediction by sampling 1,000 points from an evenly sampled grid of 10,000 longitudes and latitudes, transforming them to 3-dimensional coordinates, and assigning a "land" or "water" label to each point using the Basemap library in Matplotlib (Hunter, 2007). For classification, we associate the 3-dimensional coordinates with the signature $\mathbb{S}^2$.

**Neural spiking prediction.** We use patch-clamp electrophysiology datasets downloaded from the Allen Mouse Brain Atlas (Jones et al., 2009). We arbitrarily pick Neurons 33 and 46 for their non-trivial spiking dynamics. To represent signals in product spaces, we apply a Fast Fourier Transform and take the top 5 Fourier coefficients by magnitude. We then take their corresponding frequencies $f_i$ and represent each time point in $\mathbb{S}^1$ via the following transformation:

$$\phi : \mathbb{R}^1 \to (\mathbb{S}^1)^5, \ \phi(t) = \left( \cos\left(2\pi \frac{t}{f_i}\right), \ \sin\left(2\pi \frac{t}{f_i}\right) \right)\Big|_{i=1}^5 \tag{69}$$

This yields a product space representation in $(\mathbb{S}^1)^5$. We plot both signals, along with their reconstruction using their top 5 Fourier components, in Figure 6.

**Global temperature by month.** We downloaded a list of global average monthly temperatures for the 400 largest cities in the world from Wikipedia (Wikipedia, 2024). We transform longitude and latitude into 3-D coordinates to represent our data in $\mathbb{S}^2$. To convert months to $\mathbb{S}^1$ valued coordinates, we transform ordinal representations of months $t \in [0, 11]$ via the following transformation:

$$\phi : \mathbb{R}^1 \to \mathbb{S}^1, \ \phi(t) = \left( \cos\left(2\pi \frac{t}{12}\right), \ \sin\left(2\pi \frac{t}{12}\right) \right) \tag{70}$$

This yields a product space representation of the data in $\mathbb{S}^2 \times \mathbb{S}^1$.

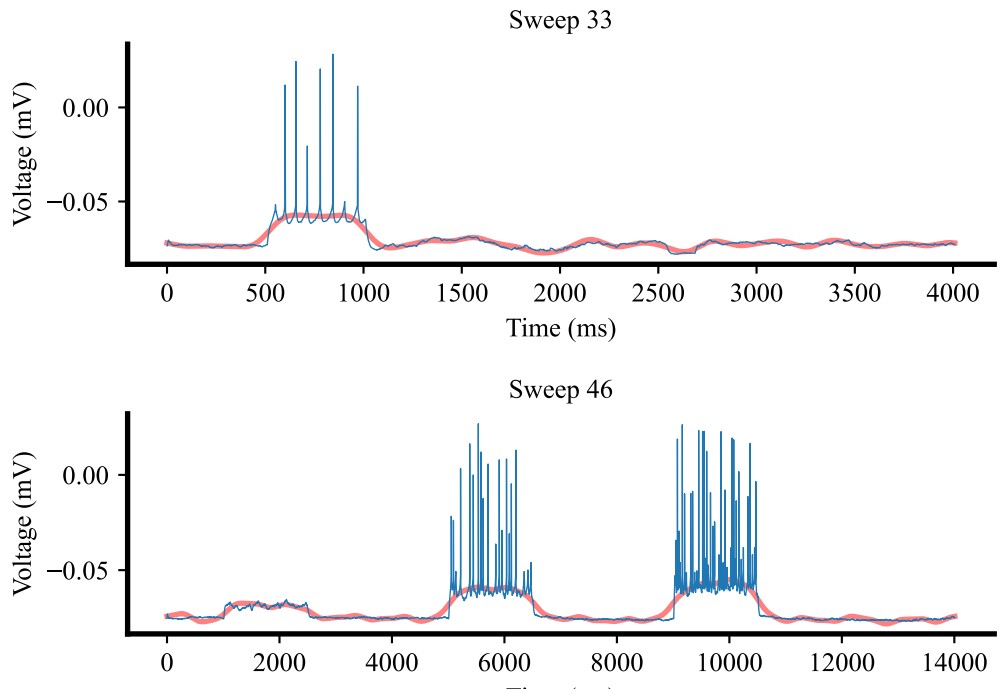

Figure 6: The "Neuron 33" and "Neuron 46" datasets, along with their reconstruction using the top 5 Fourier coefficients shown in red.

**Traffic prediction.** We download an automobile traffic prediction dataset from Kaggle (Fedesoriano, 2020). This dataset aggregates readings across four sensors with date and time annotations. We process the date and time annotation into day of year ($d$), day of week ($w$), hour ($h$), and minute ($m$) labels and transform to $(\mathbb{S}^1)^4$ analogously to the month timestamps in the global temperature data. Letting $l$ be the (numeric) label of the sensor, we apply the following transformation to our data:

$$\phi : \mathbb{R}^5 \to (\mathbb{S}^1)^5 \times \mathbb{E}^1 \tag{71}$$

$$\phi(d, w, h, m, l) = \left( \cos\left(2\pi\frac{d}{365}\right), \; \sin\left(2\pi\frac{d}{365}\right), \right.$$

$$\cos\left(2\pi\frac{w}{7}\right), \; \sin\left(2\pi\frac{w}{7}\right),$$

$$\cos\left(2\pi\frac{h}{24}\right), \; \sin\left(2\pi\frac{h}{24}\right),$$

$$\left. \cos\left(2\pi\frac{m}{60}\right), \; \sin\left(2\pi\frac{m}{60}\right), l \right) \tag{72}$$

## F ABLATIONS AND EFFECTS OF HYPERPARAMETERS

For all experiments, we sampled 100 mixtures of 32 Gaussians using the signature $\mathcal{P} = \mathbb{S}^2 \times \mathbb{E}^2 \times \mathbb{H}^2$ in an 8-class regression setting (analogous to the multi-$K$ benchmark in Tables 2 and 3, varying one parameter at a time. Results are plotted in Figure 7.

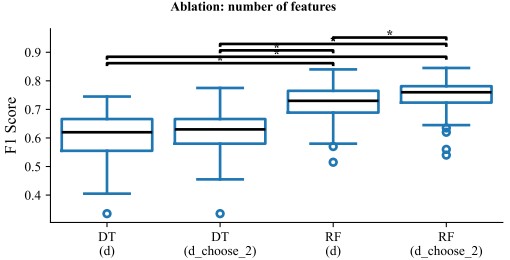

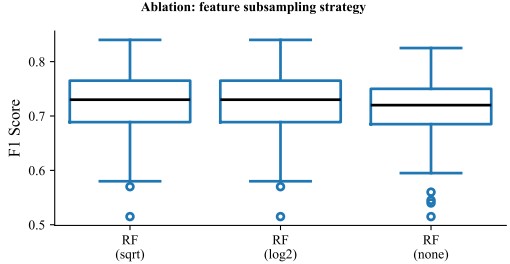

(a) Changing the number of features seen by each DT/RF from 6 to 9 by including the $(x_1, x_2)$ angle is massively beneficial.

(b) Changing feature subsampling approaches in RFs doesn't appear to do much.

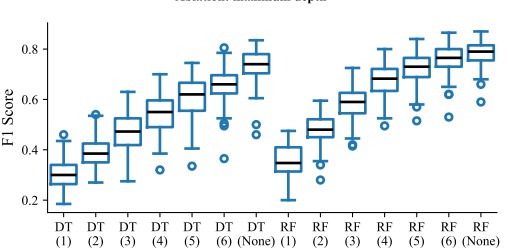

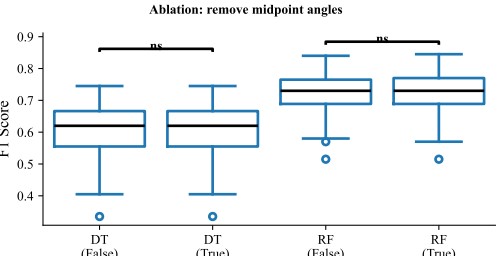

(c) Increasing the maximum depth of each DT/RF is massively beneficial, and shows no signs of overfitting even at unrestricted max depth. All within-predictor differences are significant.

(d) Replacing the midpoint-angle computations with arithmetic means has no statistically significant effect on performance for DTs or RFs, surprisingly.

Figure 7: Effects of various hyperparameters on the performance of our algorithms.

# G DETAILED RESULTS

## G.1 GLOBAL TEMPERATURE PREDICTION PLOTS

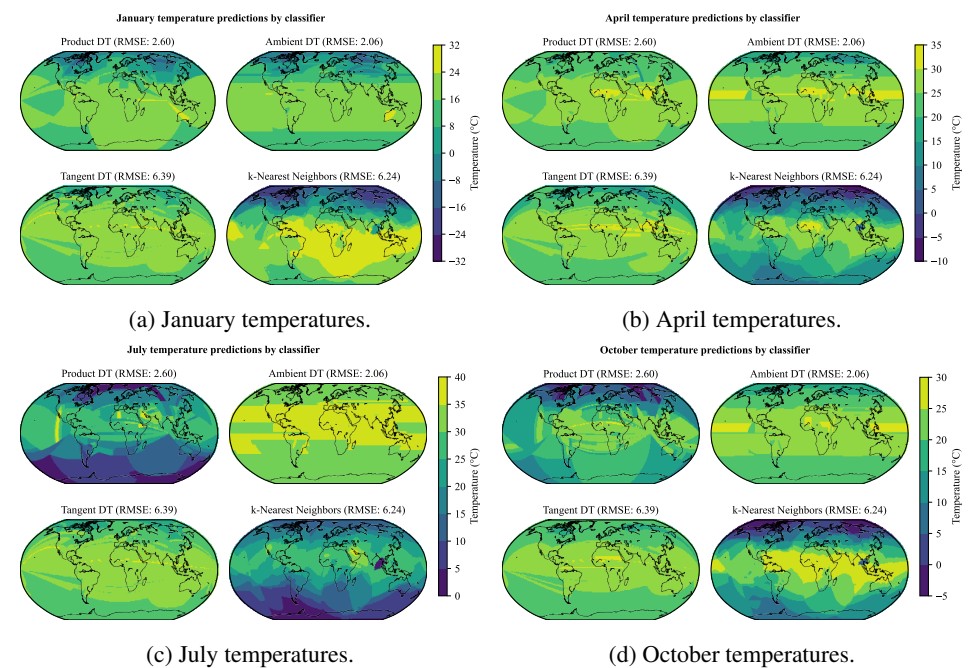

(a) January temperatures.

(b) April temperatures.

(c) July temperatures.

(d) October temperatures.

Figure 8: Decision boundaries for the temperature prediction task for the months of January, April, July, and October, colored by predicted temperature across four trained predictors.

## G.2 VAE LATENT SPACE VISUALIZATIONS

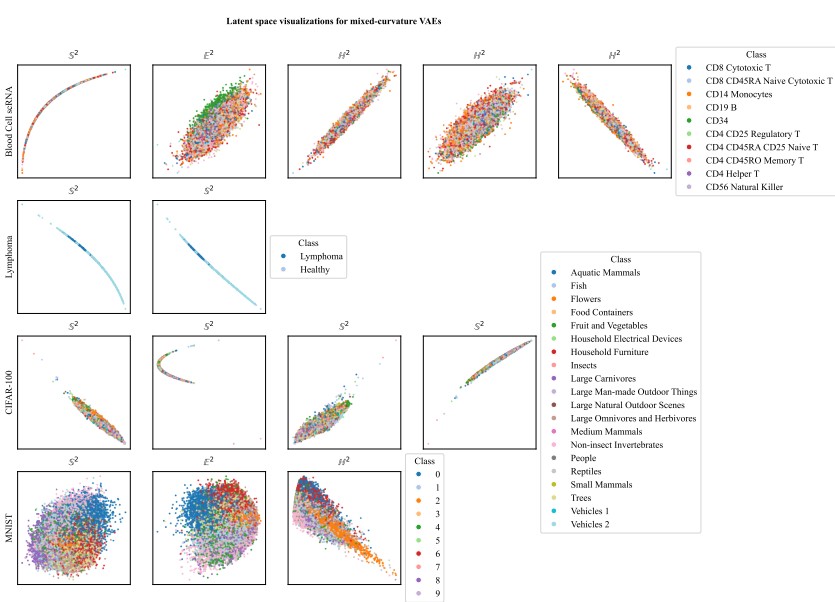

Figure 9: Visualizations of the latent space for all four of the datasets we embed using a VAE, colored by class. For visualization purposes, we show $\mathbb{S}^2$ components in 2-dimensional polar coordinates, and project $\mathbb{H}^2$ embeddings to the Poincaré disk.

## G.3 FULL RESULTS TABLE

Table 5: Full results for all benchmarks. Unlike Tables 2 and 3, this table reports full results for all classifiers. It also includes single-$K$ results and all signatures for graph embeddings. Recall the following shorthand: C=Classification ($F_1$), R=Regression (RMSE), LP=Link prediction ($F_1$); **bold** = best predictor, underline = second-best predictor; * = beating **product space methods**, † = beating **ambient space methods**, ‡ = beating **tangent plane methods**, § = beating $k$-**nearest neighbors**, ¶ = beating **product space perceptrons**.

| Dataset | Task | Signature | Perceptron | $k$-Neighbors | Euclidean DT | Euclidean RF | Tangent DT | Tangent RF | Product DT | Product RF |
|---|---|---|---|---|---|---|---|---|---|---|
| Gaussian | C | $\mathbb{E}^2$ | 19.5 ± .5 | 28.2 ± .4 | 28.4 ± .5 | 30.5 ± .5 | 28.4 ± .5 | 30.5 ± .5 | 28.5 ± .5 | **30.9 ± .5** |
| | | $\mathbb{H}^{2,0.25}$ | 18.4 ± .6 | 27.8 ± .4 | 28.5 ± .5 | 29.7 ± .5 | 28.7 ± .5 | 30.1 ± .5 | 28.7 ± .5 | **30.6 ± .5** |
| | | $\mathbb{H}^{2,0.5}$ | 21.2 ± .5 | 28.3 ± .5 | 27.8 ± .5 | 28.8 ± .5 | 27.6 ± .5 | 28.9 ± .5 | 28.3 ± .5 | **30.2 ± .4** |
| | | $\mathbb{H}^{2,1.0}$ | 21.0 ± .5 | 28.5 ± .5 | 27.9 ± .5 | 28.9 ± .5 | 26.1 ± .5 | 27.7 ± .5 | 28.2 ± .5 | **30.8 ± .5** |
| | | $\mathbb{H}^{2,2.0}$ | 18.2 ± .5 | 27.0 ± .5 | 26.2 ± .5 | 26.6 ± .5 | 24.3 ± .5 | 25.6 ± .5 | 26.4 ± .5 | **28.1 ± .5** |
| | | $\mathbb{H}^{2,4.0}$ | 17.3 ± .5 | 26.1 ± .5 | 25.3 ± .5 | 25.5 ± .5 | 22.7 ± .5 | 24.3 ± .4 | 25.2 ± .5 | **27.2 ± .5** |
| | | $\mathbb{S}^{2,-0.25}$ | 16.1 ± .5 | 29.4 ± .5 | 29.1 ± .5 | 29.5 ± .5 | 29.2 ± .4 | 30.5 ± .5 | 29.2 ± .5 | **31.2 ± .5** |
| | | $\mathbb{S}^{2,-0.5}$ | 16.3 ± .5 | 30.6 ± .5 | 29.2 ± .5 | 30.4 ± .5 | 29.6 ± .5 | 31.4 ± .5 | 30.2 ± .5 | **32.3 ± .5** |
| | | $\mathbb{S}^{2,-1.0}$ | 16.1 ± .5 | 32.6 ± .5 | 30.0 ± .5 | 30.4 ± .5 | 30.3 ± .5 | 32.1 ± .5 | 30.9 ± .5 | **33.2 ± .5** |
| | | $\mathbb{S}^{2,-2.0}$ | 15.6 ± .6 | 36.1 ± .5 | 32.3 ± .5 | 32.9 ± .6 | 33.8 ± .6 | 35.5 ± .6 | 34.1 ± .6 | **36.7 ± .5** |
| | | $\mathbb{S}^{2,-4.0}$ | 16.4 ± .6 | 41.6 ± .5 | 32.4 ± .5 | 33.0 ± .5 | 35.2 ± .5 | 37.6 ± .5 | 36.3 ± .5 | **40.1 ± .5** |
| | R | $\mathbb{E}^2$ | | .211 ± .002 | .201 ± .002 | .199 ± .002 | .201 ± .002 | .199 ± .002 | .201 ± .002 | **.198 ± .002** |
| | | $\mathbb{H}^{2,0.25}$ | | .207 ± .003 | .199 ± .002 | **.196 ± .002** | .199 ± .002 | **.196 ± .002** | .199 ± .002 | **.196 ± .002** |
| | | $\mathbb{H}^{2,0.5}$ | | .206 ± .003 | .198 ± .002 | **.194 ± .002** | .197 ± .002 | **.194 ± .002** | .197 ± .003 | **.193 ± .002** |
| | | $\mathbb{H}^{2,1.0}$ | | .214 ± .003 | .205 ± .003 | **.201 ± .003** | .206 ± .003 | .203 ± .003 | .205 ± .003 | **.201 ± .003** |
| | | $\mathbb{H}^{2,2.0}$ | | .211 ± .003 | .202 ± .003 | **.199 ± .003** | .203 ± .003 | .201 ± .003 | .202 ± .003 | **.198 ± .003** |
| | | $\mathbb{H}^{2,4.0}$ | | .215 ± .003 | .206 ± .003 | **.202 ± .003** | .207 ± .003 | .204 ± .003 | .206 ± .003 | **.202 ± .003** |
| | | $\mathbb{S}^{2,-0.25}$ | | .206 ± .003 | .198 ± .003 | **.195 ± .003** | .198 ± .003 | **.195 ± .003** | .198 ± .003 | **.194 ± .003** |
| | | $\mathbb{S}^{2,-0.5}$ | | .209 ± .003 | .203 ± .003 | **.199 ± .003** | .201 ± .003 | **.198 ± .003** | .202 ± .003 | **.197 ± .003** |
| | | $\mathbb{S}^{2,-1.0}$ | | .207 ± .003 | .203 ± .003 | **.199 ± .003** | .201 ± .003 | **.197 ± .003** | .201 ± .003 | **.197 ± .003** |
| | | $\mathbb{S}^{2,-2.0}$ | | .206 ± .003 | .205 ± .003 | .201 ± .003 | .201 ± .003 | **.198 ± .003** | .201 ± .003 | **.196 ± .003** |
| | | $\mathbb{S}^{2,-4.0}$ | | .198 ± .003 | .199 ± .003 | .196 ± .002 | .196 ± .003 | **.192 ± .003** | .195 ± .003 | **.190 ± .003** |
| Gaussian | C | $(\mathbb{S}^2)^2$ | 25.4 ± .5 | **35.7 ± .5** | 29.6 ± .5 | 32.3 ± .5 | 28.9 ± .4 | 31.5 ± .5 | 30.2 ± .5 | 33.1 ± .5 |
| | | $\mathbb{E}^4$ | 21.2 ± .5 | **34.9 ± .5** | 27.1 ± .5 | 30.0 ± .4 | 27.1 ± .5 | 30.0 ± .4 | 28.1 ± .5 | 31.3 ± .4 |
| | | $\mathbb{H}^2\mathbb{E}^2$ | 20.2 ± .5 | **40.3 ± .4** | 30.8 ± .5 | 34.7 ± .5 | 30.9 ± .5 | 34.8 ± .5 | 32.0 ± .5 | 36.1 ± .5 |
| | | $\mathbb{H}^2\mathbb{S}^2$ | 20.2 ± .5 | **38.4 ± .5** | 30.6 ± .5 | 33.3 ± .5 | 30.5 ± .5 | 33.6 ± .5 | 31.2 ± .5 | 35.2 ± .5 |
| | | $\mathbb{H}^4$ | 15.6 ± .5 | **47.5 ± .5** | 32.9 ± .5 | 35.9 ± .4 | 33.0 ± .4 | 37.9 ± .4 | 35.2 ± .5 | 40.0 ± .4 |
| | | $\mathbb{S}^2\mathbb{E}^2$ | 23.7 ± .5 | **35.8 ± .5** | 29.5 ± .4 | 32.7 ± .5 | 29.3 ± .5 | 32.4 ± .4 | 30.0 ± .5 | 33.5 ± .4 |
| | | $\mathbb{S}^4$ | 18.1 ± .6 | **33.1 ± .4** | 25.9 ± .5 | 27.6 ± .4 | 24.0 ± .4 | 26.2 ± .5 | 25.1 ± .5 | 28.0 ± .5 |
| | | $(\mathbb{H}^2)^2$ | 15.7 ± .6 | **41.5 ± .5** | 31.7 ± .5 | 34.5 ± .5 | 31.9 ± .5 | 35.4 ± .5 | 32.3 ± .5 | 37.0 ± .5 |
| | R | $(\mathbb{S}^2)^2$ | | .196 ± .002 | .196 ± .002 | **.191 ± .002** | .197 ± .002 | **.191 ± .002** | .196 ± .002 | **.191 ± .002** |
| | | $\mathbb{E}^4$ | | .194 ± .003 | .197 ± .003 | **.191 ± .002** | .197 ± .003 | **.191 ± .002** | .196 ± .003 | **.190 ± .002** |
| | | $\mathbb{H}^2\mathbb{E}^2$ | | .196 ± .003 | .199 ± .002 | .194 ± .002 | .199 ± .003 | **.193 ± .002** | .198 ± .002 | **.193 ± .002** |
| | | $\mathbb{H}^2\mathbb{S}^2$ | | .197 ± .003 | .200 ± .002 | **.194 ± .002** | .199 ± .002 | **.194 ± .002** | .199 ± .002 | **.193 ± .002** |
| | | $\mathbb{H}^4$ | | **.175 ± .003** | .189 ± .003 | .184 ± .003 | .187 ± .003 | .181 ± .003 | .185 ± .003 | **.178 ± .003** |
| | | $\mathbb{S}^2\mathbb{E}^2$ | | .199 ± .003 | .200 ± .003 | **.194 ± .002** | .201 ± .003 | .195 ± .003 | .199 ± .003 | **.194 ± .002** |
| | | $\mathbb{S}^4$ | | .194 ± .002 | .193 ± .002 | **.188 ± .002** | .193 ± .002 | .190 ± .002 | .194 ± .002 | **.189 ± .002** |
| | | $(\mathbb{H}^2)^2$ | | .193 ± .003 | .198 ± .002 | .193 ± .002 | .193 ± .002 | .192 ± .002 | .196 ± .002 | **.191 ± .002** |
| CiteSeer | C | $(\mathbb{S}^2)^2$ | 13.5 ± .6 | 25.1 ± .6 | 25.3 ± .5 | 27.0 ± .7 | 25.3 ± .6 | 26.2 ± .7 | 25.8 ± .6 | **27.1 ± .7** |
| | | $\mathbb{E}^4$ | 13.4 ± .5 | 24.1 ± .4 | 24.5 ± .7 | **25.9 ± .4** | 24.5 ± .7 | 25.0 ± .7 | 23.7 ± .4 | 25.1 ± .5 |
| | | $\mathbb{H}^2\mathbb{E}^2$ | 13.4 ± .4 | 24.7 ± .7 | 25.4 ± .5 | **26.6 ± .6** | 24.9 ± .5 | 25.6 ± .5 | 25.2 ± .5 | 26.4 ± .6 |
| | | $\mathbb{H}^2\mathbb{S}^2$ | 13.7 ± .2 | 25.9 ± .5 | 25.4 ± .4 | **26.1 ± .7** | 25.0 ± .6 | 25.9 ± .7 | 25.8 ± .6 | 25.6 ± .6 |
| | | $\mathbb{H}^4$ | 14.1 ± .4 | 26.2 ± .8 | 25.5 ± .8 | **26.9 ± .9** | 24.6 ± .7 | 26.5 ± .8 | 24.0 ± .4 | 26.0 ± .9 |
| | | $\mathbb{S}^2\mathbb{E}^2$ | 13.7 ± .3 | 25.6 ± .8 | 24.9 ± .6 | **26.3 ± .6** | 25.0 ± .7 | 25.8 ± .6 | 24.4 ± .5 | 25.9 ± .5 |
| | | $\mathbb{S}^4$ | 13.7 ± .4 | 24.7 ± .7 | 24.5 ± .9 | **25.5 ± .7** | 24.4 ± .7 | 24.9 ± .7 | 24.2 ± .7 | 24.9 ± .5 |
| | | $(\mathbb{H}^2)^2$ | 5.4 ± .2 | 24.9 ± .7 | 25.8 ± .6 | **27.3 ± .5** | 25.3 ± .8 | 26.6 ± .5 | 25.4 ± .6 | 26.8 ± .7 |
| Cora | C | $(\mathbb{S}^2)^2$ | 18.2 ± 1.2 | 21.4 ± .4 | 29.2 ± .5 | **29.9 ± .4** | 29.2 ± .4 | 29.7 ± .5 | 27.9 ± 1.2 | 29.6 ± .5 |
| | | $\mathbb{E}^4$ | 16.4 ± .5 | 21.1 ± .4 | 28.4 ± .4 | 29.3 ± .5 | 28.4 ± .4 | **29.7 ± .6** | 28.9 ± .5 | 29.4 ± .6 |
| | | $\mathbb{H}^2\mathbb{E}^2$ | 17.3 ± .5 | 20.1 ± .6 | 28.7 ± .5 | 29.2 ± .5 | 28.8 ± .4 | **29.4 ± .5** | 28.7 ± .4 | 29.2 ± .5 |
| | | $\mathbb{H}^2\mathbb{S}^2$ | 15.9 ± .2 | 20.9 ± .5 | 28.5 ± .5 | 29.8 ± .6 | 28.8 ± .4 | 29.8 ± .6 | 29.0 ± .6 | **29.9 ± .5** |
| | | $\mathbb{H}^4$ | 20.4 ± 1.8 | 20.7 ± .4 | 28.5 ± .6 | **28.9 ± .5** | 28.1 ± .6 | 28.7 ± .5 | 27.4 ± .6 | 28.9 ± .4 |
| | | $\mathbb{S}^2\mathbb{E}^2$ | 16.7 ± .2 | 21.0 ± .5 | 28.0 ± .6 | **28.6 ± .5** | 27.8 ± .6 | 28.5 ± .5 | 28.5 ± .6 | 28.5 ± .5 |
| | | $\mathbb{S}^4$ | 16.8 ± .5 | 20.7 ± .5 | 27.9 ± .8 | **28.9 ± .6** | 27.8 ± .5 | **28.9 ± .5** | 27.7 ± .7 | 28.8 ± .6 |
| | | $(\mathbb{H}^2)^2$ | 5.1 ± .2 | 20.7 ± .4 | 29.3 ± .6 | **30.3 ± .6** | 29.6 ± .6 | 30.3 ± .6 | 29.5 ± .5 | 30.3 ± .6 |
| PolBlogs | C | $(\mathbb{S}^2)^2$ | 50.4 ± .8 | **93.5 ± .4** | 93.2 ± .5 | 93.1 ± .4 | 90.9 ± .8 | 91.0 ± .7 | 89.6 ± 3.3 | 92.9 ± .4 |
| | | $\mathbb{E}^4$ | 48.4 ± .8 | **93.9 ± .4** | 92.4 ± .3 | 92.7 ± .5 | 92.5 ± .3 | 93.1 ± .3 | 93.3 ± .3 | 93.6 ± .5 |
| | | $\mathbb{H}^2\mathbb{E}^2$ | 63.6 ± 5.2 | **93.6 ± .4** | 92.9 ± .4 | 93.4 ± .3 | 92.7 ± .4 | 92.9 ± .5 | 93.1 ± .5 | 93.6 ± .2 |
| | | $\mathbb{H}^2\mathbb{S}^2$ | 62.1 ± 5.6 | **94.0 ± .5** | 92.9 ± .5 | 93.6 ± .6 | 92.2 ± .5 | 93.2 ± .6 | 92.4 ± .4 | 93.3 ± .5 |
| | | $\mathbb{H}^4$ | 48.7 ± .6 | **93.5 ± .3** | 92.1 ± .7 | 92.5 ± .5 | 91.9 ± .7 | 92.5 ± .6 | 92.7 ± .5 | 93.1 ± .4 |
| | | $\mathbb{S}^2\mathbb{E}^2$ | 49.8 ± 2.6 | **94.3 ± .2** | 93.4 ± .6 | 93.9 ± .5 | 92.5 ± 1.1 | 93.6 ± .5 | 93.8 ± .6 | 93.2 ± .6 |
| | | $\mathbb{S}^4$ | 48.0 ± 1.2 | **93.4 ± .3** | 91.5 ± .5 | 92.9 ± .5 | 91.7 ± .8 | 92.8 ± .5 | 92.8 ± .3 | 93.1 ± .4 |
| | | $(\mathbb{H}^2)^2$ | 48.0 ± .9 | **93.9 ± .4** | 92.5 ± .6 | 92.8 ± .7 | 92.7 ± .6 | 93.3 ± .5 | 92.7 ± .4 | 93.7 ± .6 |
| CS PhDs | R | $(\mathbb{S}^2)^2$ | .043 ± .004 | .058 ± .006 | .048 ± .004 | .055 ± .006 | .046 ± .004 | **.041 ± .003** | .046 ± .003 | .046 ± .005 |
| | | $\mathbb{E}^4$ | **.035 ± .003** | .056 ± .003 | .040 ± .005 | .047 ± .005 | .040 ± .005 | .048 ± .004 | .038 ± .004 | .046 ± .005 |
| | | $\mathbb{H}^2\mathbb{E}^2$ | .043 ± .004 | .057 ± .004 | .040 ± .004 | .051 ± .003 | **.039 ± .005** | .049 ± .006 | .045 ± .003 | .044 ± .004 |
| | | $\mathbb{H}^2\mathbb{S}^2$ | .045 ± .004 | .048 ± .003 | **.043 ± .004** | .054 ± .004 | .045 ± .005 | .048 ± .006 | .044 ± .004 | .055 ± .005 |
| | | $\mathbb{H}^4$ | .044 ± .002 | .053 ± .005 | .052 ± .002 | .052 ± .005 | .049 ± .005 | .060 ± .003 | **.041 ± .004** | .057 ± .003 |
| | | $\mathbb{S}^2\mathbb{E}^2$ | .042 ± .005 | .067 ± .006 | .040 ± .002 | .050 ± .004 | **.038 ± .004** | .048 ± .007 | .040 ± .004 | .053 ± .005 |
| | | $\mathbb{S}^4$ | .043 ± .004 | .065 ± .005 | **.040 ± .004** | .050 ± .004 | **.041 ± .003** | .043 ± .004 | .047 ± .004 | .048 ± .003 |
| | | $(\mathbb{H}^2)^2$ | **.001 ± .001** | .067 ± .006 | .040 ± .003 | .050 ± .004 | .043 ± .002 | .052 ± .006 | .042 ± .004 | .051 ± .004 |
| AdjNoun | LP | $\mathbb{S}^2\mathbb{E}^2\mathbb{H}^2$ | **93.7 ± 1.1** | 93.3 ± 1.1 | 93.5 ± .9 | **93.7 ± 1.1** | 93.5 ± .9 | **93.7 ± 1.1** | 93.5 ± .9 | **93.7 ± 1.1** |
| Dolphins | LP | $\mathbb{S}^2\mathbb{E}^2\mathbb{H}^2$ | 90.7 ± .7 | 92.1 ± .6 | **96.6 ± .3** | 92.3 ± .9 | 95.0 ± .8 | 90.9 ± .8 | **96.6 ± .3** | 90.7 ± .7 |
| Football | LP | $\mathbb{S}^2\mathbb{E}^2\mathbb{H}^2$ | 82.0 ± 3.3 | 79.8 ± 3.3 | **85.7 ± 3.6** | 83.1 ± 3.5 | **85.7 ± 3.6** | 83.1 ± 3.5 | **85.7 ± 3.6** | 82.0 ± 3.3 |
| Karate Club | LP | $\mathbb{S}^2\mathbb{E}^2\mathbb{H}^2$ | 65.7 ± 10.0 | 89.4 ± 1.9 | **95.1 ± 1.5** | 88.6 ± 2.2 | **95.1 ± 1.5** | 88.8 ± 1.9 | **95.1 ± 1.5** | 88.8 ± 2.6 |
| Les Mis | LP | $\mathbb{S}^2\mathbb{E}^2\mathbb{H}^2$ | 92.2 ± .9 | 93.8 ± 1.1 | **95.7 ± .7** | 92.7 ± .9 | 95.5 ± .6 | 93.7 ± 1.0 | 95.6 ± .8 | 92.2 ± .9 |
| PolBooks | LP | $\mathbb{S}^2\mathbb{E}^2\mathbb{H}^2$ | 92.2 ± .6 | 94.8 ± .3 | **95.8 ± .4** | 92.9 ± .6 | **95.8 ± .4** | 92.1 ± .6 | **95.8 ± .4** | 92.2 ± .6 |
| Blood | C | $\mathbb{S}^2\mathbb{E}^2(\mathbb{H}^2)^3$ | 2.8 ± .0 | 17.4 ± .5 | 18.9 ± .4 | 19.3 ± .5 | 18.7 ± .4 | 19.6 ± .4 | 17.6 ± 1.0 | **20.1 ± .5** |
| CIFAR-100 | C | $(\mathbb{S}^2)^4$ | 5.7 ± .4 | 8.6 ± .4 | 10.0 ± .4 | 11.5 ± .5 | 10.1 ± .4 | 11.5 ± .5 | 10.8 ± .3 | **12.0 ± .3** |
| Lymphoma | C | $(\mathbb{S}^2)^2$ | 78.1 ± .3 | 77.8 ± 1.4 | 81.7 ± 1.2 | 81.6 ± 1.3 | 81.2 ± 1.4 | 81.4 ± 1.3 | **83.7 ± 1.2** | 83.1 ± 1.2 |
| MNIST | C | $\mathbb{S}^2\mathbb{E}^2\mathbb{H}^2$ | 16.9 ± 3.0 | **41.9 ± 3.7** | 28.9 ± 1.3 | 35.7 ± 2.8 | 28.6 ± 1.0 | 36.5 ± 2.9 | 30.9 ± 1.6 | 39.4 ± 2.3 |
| Landmasses | C | $\mathbb{S}^2$ | 70.6 ± .4 | **91.4 ± .2** | 81.2 ± .4 | 83.5 ± .3 | 79.7 ± .3 | 81.8 ± .3 | 83.5 ± .3 | 84.2 ± .3 |
| Neuron 33 | C | $(\mathbb{S}^1)^5$ | 53.9 ± .3 | 50.5 ± .5 | 76.2 ± .4 | 76.0 ± .5 | 76.2 ± .4 | 75.8 ± .5 | 76.0 ± .5 | **77.0 ± .4** |
| Neuron 46 | C | $(\mathbb{S}^1)^5$ | 51.5 ± .1 | 50.2 ± .2 | 60.7 ± .3 | 61.1 ± .3 | 59.3 ± .3 | 59.9 ± .3 | 60.8 ± .3 | **61.2 ± .3** |
| Temperature | R | $\mathbb{S}^2\mathbb{S}^1$ | | 7.198 ± .212 | 5.290 ± .246 | **4.531 ± .187** | 7.823 ± .196 | 6.261 ± .132 | 7.574 ± .249 | 7.130 ± .123 |
| Traffic | R | $\mathbb{E}^1(\mathbb{S}^1)^4$ | | .510 ± .003 | .521 ± .003 | **.505 ± .003** | .526 ± .003 | .515 ± .003 | .534 ± .003 | .577 ± .005 |

Row-group labels (left margin): Synthetic (single-$K$); Synthetic (multi-$K$); Graph embeddings; VAE; Other.

# H  DATASETS AVAILABILITY

Table 6: All of the datasets used in this paper, with download links and citations. CC-BY-SA is short for the Creative Commons Attribution-ShareAlike license. Allen TOU is the Allen Institute terms of use, found at `https://alleninstitute.org/terms-of-use/`.

| Dataset | Link | License | Citation |
|---|---|---|---|
| CiteSeer | Network Repository: CiteSeer | CC-BY-SA | Giles et al. (1998) |
| Cora | Network Repository: CORA | CC-BY-SA | Sen et al. (2008) |
| Polblogs | Network Repository: Polblogs | CC-BY-SA | Adamic & Glance (2005) |
| CS PhDs | Pajek datasets: PhD students in CS | CC-BY-SA | Johnson (1984) |
| Adjnoun | Network Repository: Adjnoun | CC-BY-SA | Newman (2006) |
| Dolphins | Network Repository: Dolphins | CC-BY-SA | Lusseau et al. (2003) |
| Football | Network Repository: Football | CC-BY-SA | Girvan & Newman (2002) |
| Karate Club | Network Repository: Karate | CC-BY-SA | Zachary (1977) |
| Les Mis | Network Repository: Les Mis | CC-BY-SA | Knuth (1993) |
| Polbooks | Network Repository: Polblooks | CC-BY-SA | Krebs (2004) |
| Blood | 10x Genomics: CD8+ Cytotoxic T-cells | CC-BY-SA | Zheng et al. (2017) |
| Blood | CD8+/CD45RA+ Naive Cytotoxic T Cells | CC-BY-SA | Zheng et al. (2017) |
| Blood | 10x Genomics: CD56+ Natural Killer Cells | CC-BY-SA | Zheng et al. (2017) |
| Blood | 10x Genomics: CD4+ Helper T Cells | CC-BY-SA | Zheng et al. (2017) |
| Blood | 10x Genomics: CD4+/CD45RO+ Memory T Cells | CC-BY-SA | Zheng et al. (2017) |
| Blood | 10x Genomics: CD4+/CD45RA+/CD25- Naive T cells | CC-BY-SA | Zheng et al. (2017) |
| Blood | CD4+/CD25+ Regulatory T Cells | CC-BY-SA | Zheng et al. (2017) |
| Blood | 10x Genomics: CD34+ Cells | CC-BY-SA | Zheng et al. (2017) |
| Blood | CD19+ B Cells | CC-BY-SA | Zheng et al. (2017) |
| Blood | 10x Genomics: CD14+ Monocytes | CC-BY-SA | Zheng et al. (2017) |
| Lymphoma | Hodgkin's Lymphoma, Dissociated Tumor: Targeted, Immunology Panel | CC-BY-SA | 10x Genomics (2020a) |
| Lymphoma | PBMCs from a Healthy Donor: Targeted-Compare, Immunology Panel | CC-BY-SA | 10x Genomics (2020b) |
| MNIST | HuggingFace: MNIST | MIT | Lecun et al. (1998) |
| CIFAR-100 | HuggingFace: CIFAR-100 | None | Krizhevsky (2009) |
| Landmasses | Basemap 1.4.1: `is_land` | None | None |
| Neurons | Allen Brain Atlas | Allen TOU | Jones et al. (2009) |
| Temperature | Wikipedia: List of cities by average temperature | CC-BY-SA | Wikipedia (2024) |
| Traffic | Kaggle: Traffic Prediction Dataset | None | Fedesoriano (2020) |

# I COMPARISON TO NEURAL BASELINES

## I.1 MODELS

Neural networks, especially graph neural networks, are a popular choice for representing and working with mixed-curvature representations (Sun et al., 2021; Cho et al., 2023; Bachmann et al., 2020; McNeela et al., 2024). Following the typical node classification approach described in the literature, we compared our method to both deep neural networks (ML) and graph neural networks (GNN). We find that our methods are generally competitive with neural baselines, especially when less data is available.

We trained each model for 200 epochs using Adam Kingma & Ba (2017) with a learning rate of .01, $\beta_1 = 0.9$, and $\beta_2 = -0.999$. Each model used one hidden dimension equal to its ambient dimension. For classification, we used an output dimension equal to the number of classes and cross-entropy loss; for regression, we used a single output dimension and mean squared error loss.

We additionally train tangent plane variants of the MLP and GNN models, in which the data is preprocessed using a logarithmic map and subsequently treated as Euclidean. For datasets where the graph topology is not provided (e.g. Gaussian mixtures), we take the pairwise distances in the manifold geometry and transform them into dense weighted edges using the Gaussian kernel:

$$\mathbf{D}_{i,j} = \delta_{\mathcal{P}}(\mathbf{x_i}, \mathbf{x_j}) \tag{73}$$
$$\mathbf{A} = \exp(-\mathbf{D}) \tag{74}$$

For graph datasets with known topology, we instead used the true adjacency matrix and an ablation in which the adjacency matrix is replaced by the dense Gaussian kernel estimate. Interestingly, for several of the graph datasets, substituting the true adjacencies with a Gaussian kernel on embedding distances did not substantially hinder performance; however, it rarely helped.

## I.2 DATASETS

Due to time considerations, we ran our benchmarks on a representative sample of the datasets. For adjacency-free datasets, we chose Gaussian mixtures in single-component signatures (as in Figures 3 and 4) and multiple-component signatures (as in Tables 2 and 3). We also ran each graph dataset on the lowest $D_{avg}$ signature (i.e. the signatures reported in Tables 2 and 3).

## I.3 RESULTS

We tabulate our results as follows: Tables 7 and 9 contain classification and regression benchmarks for single-$K$ manifolds; Tables 8 and 10 contain classification and regression benchmarks for product spaces; and Table 11 contains graph dataset benchmarks. For all benchmarks except graph data, we beat both MLPs and GNNs; for datasets with informative graph topologies, GNNs are a sensible alternative.

Table 7: Comparison to neural networks on the constant-curvature classification task.

| $K$ | Product DT | Product RF | $T_{\mu_0}\mathcal{P}$ MLP | MLP | $T_{\mu_0}\mathcal{P}$ GNN | GNN |
|---|---|---|---|---|---|---|
| -4 | $\underline{0.34 \pm 0.10}$ | $\mathbf{0.36 \pm 0.07}$ | $0.27 \pm 0.08$ | $0.17 \pm 0.10$ | $0.23 \pm 0.18$ | $0.18 \pm 0.15$ |
| -2 | $\underline{0.36 \pm 0.10}$ | $\mathbf{0.37 \pm 0.11}$ | $0.29 \pm 0.12$ | $0.21 \pm 0.16$ | $0.29 \pm 0.10$ | $0.21 \pm 0.11$ |
| -1 | $\underline{0.32 \pm 0.09}$ | $\mathbf{0.34 \pm 0.08}$ | $0.26 \pm 0.14$ | $0.23 \pm 0.09$ | $0.27 \pm 0.11$ | $0.22 \pm 0.15$ |
| -0.5 | $\underline{0.32 \pm 0.09}$ | $\mathbf{0.33 \pm 0.10}$ | $0.27 \pm 0.10$ | $0.29 \pm 0.08$ | $0.28 \pm 0.07$ | $0.23 \pm 0.12$ |
| -0.25 | $\underline{0.30 \pm 0.11}$ | $\mathbf{0.32 \pm 0.11}$ | $0.26 \pm 0.11$ | $0.25 \pm 0.08$ | $0.26 \pm 0.09$ | $0.21 \pm 0.08$ |
| 0 | $\underline{0.29 \pm 0.11}$ | $\mathbf{0.31 \pm 0.08}$ | $0.23 \pm 0.11$ | $0.26 \pm 0.12$ | $0.24 \pm 0.05$ | $0.24 \pm 0.05$ |
| 0.25 | $\underline{0.28 \pm 0.09}$ | $\mathbf{0.30 \pm 0.11}$ | $0.26 \pm 0.11$ | $\underline{0.28 \pm 0.09}$ | $0.27 \pm 0.09$ | $0.21 \pm 0.14$ |
| 0.5 | $0.25 \pm 0.10$ | $\mathbf{0.29 \pm 0.11}$ | $\underline{0.26 \pm 0.11}$ | $0.25 \pm 0.13$ | $0.25 \pm 0.12$ | $0.22 \pm 0.14$ |
| 1 | $\underline{0.27 \pm 0.07}$ | $\mathbf{0.29 \pm 0.05}$ | $0.21 \pm 0.07$ | $0.22 \pm 0.07$ | $0.22 \pm 0.07$ | $0.21 \pm 0.07$ |
| 2 | $0.26 \pm 0.07$ | $\mathbf{0.29 \pm 0.07}$ | $0.23 \pm 0.09$ | $\underline{0.26 \pm 0.09}$ | $0.23 \pm 0.12$ | $0.23 \pm 0.12$ |
| 4 | $\underline{0.25 \pm 0.11}$ | $\mathbf{0.26 \pm 0.08}$ | $0.22 \pm 0.06$ | $0.21 \pm 0.07$ | $0.21 \pm 0.09$ | $0.22 \pm 0.07$ |

Table 8: Comparison to neural networks on the mixed-curvature classification task.

| $\mathcal{P}$ | Product DT | Product RF | $T_{\mu_0}\mathcal{P}$ MLP | MLP | $T_{\mu_0}\mathcal{P}$ GNN | GNN |
|---|---|---|---|---|---|---|
| $\mathbb{E}^4$ | $\underline{0.31 \pm 0.05}$ | $\mathbf{0.35 \pm 0.07}$ | $0.30 \pm 0.11$ | $\underline{0.31 \pm 0.11}$ | $0.27 \pm 0.12$ | $0.27 \pm 0.12$ |
| $\mathbb{H}^4$ | $\underline{0.39 \pm 0.08}$ | $\mathbf{0.43 \pm 0.10}$ | $0.38 \pm 0.11$ | $0.31 \pm 0.09$ | $0.32 \pm 0.13$ | $0.22 \pm 0.09$ |
| $\mathbb{H}^2\mathbb{E}^2$ | $0.34 \pm 0.11$ | $\mathbf{0.38 \pm 0.11}$ | $\underline{0.35 \pm 0.13}$ | $0.34 \pm 0.14$ | $0.30 \pm 0.13$ | $0.23 \pm 0.11$ |
| $(\mathbb{H}^2)^2$ | $\underline{0.35 \pm 0.09}$ | $\mathbf{0.36 \pm 0.08}$ | $\underline{0.35 \pm 0.06}$ | $0.34 \pm 0.06$ | $0.28 \pm 0.07$ | $0.23 \pm 0.11$ |
| $\mathbb{H}^2\mathbb{S}^2$ | $0.31 \pm 0.09$ | $\mathbf{0.36 \pm 0.10}$ | $0.33 \pm 0.10$ | $\underline{0.35 \pm 0.09}$ | $0.27 \pm 0.09$ | $0.22 \pm 0.06$ |
| $\mathbb{S}^4$ | $\underline{0.25 \pm 0.06}$ | $\mathbf{0.30 \pm 0.05}$ | $\underline{0.25 \pm 0.07}$ | $0.23 \pm 0.08$ | $0.20 \pm 0.08$ | $0.20 \pm 0.10$ |
| $\mathbb{S}^2\mathbb{E}^2$ | $0.29 \pm 0.09$ | $\mathbf{0.32 \pm 0.08}$ | $0.30 \pm 0.09$ | $\underline{0.31 \pm 0.11}$ | $0.25 \pm 0.06$ | $0.24 \pm 0.08$ |
| $(\mathbb{S}^2)^2$ | $0.30 \pm 0.08$ | $\mathbf{0.34 \pm 0.10}$ | $0.33 \pm 0.08$ | $\mathbf{0.34 \pm 0.10}$ | $0.23 \pm 0.09$ | $0.18 \pm 0.11$ |

Table 9: Comparison to neural networks on the single- and mixed-curvature regression tasks.

| $K$ | Product DT | Product RF | $T_{\mu_0}\mathcal{P}$ MLP | MLP | $T_{\mu_0}\mathcal{P}$ GNN | GNN |
|---|---|---|---|---|---|---|
| -4 | $\underline{0.20 \pm 0.04}$ | $\mathbf{0.19 \pm 0.04}$ | $0.55 \pm 0.13$ | $0.55 \pm 0.13$ | $0.55 \pm 0.13$ | $0.55 \pm 0.13$ |
| -2 | $\underline{0.21 \pm 0.04}$ | $\mathbf{0.20 \pm 0.04}$ | $0.55 \pm 0.08$ | $0.55 \pm 0.08$ | $0.55 \pm 0.08$ | $0.55 \pm 0.08$ |
| -1 | $\mathbf{0.19 \pm 0.03}$ | $\mathbf{0.19 \pm 0.03}$ | $0.54 \pm 0.09$ | $0.54 \pm 0.09$ | $0.54 \pm 0.09$ | $0.54 \pm 0.09$ |
| -0.5 | $\underline{0.19 \pm 0.05}$ | $\mathbf{0.18 \pm 0.05}$ | $0.53 \pm 0.12$ | $0.53 \pm 0.12$ | $0.53 \pm 0.12$ | $0.53 \pm 0.12$ |
| -0.25 | $\underline{0.20 \pm 0.02}$ | $\mathbf{0.19 \pm 0.02}$ | $0.53 \pm 0.10$ | $0.53 \pm 0.10$ | $0.53 \pm 0.10$ | $0.53 \pm 0.10$ |
| 0 | $\underline{0.21 \pm 0.03}$ | $\mathbf{0.20 \pm 0.03}$ | $0.55 \pm 0.11$ | $0.55 \pm 0.11$ | $0.55 \pm 0.11$ | $0.55 \pm 0.11$ |
| 0.25 | $\underline{0.21 \pm 0.03}$ | $\mathbf{0.20 \pm 0.03}$ | $0.54 \pm 0.08$ | $0.54 \pm 0.08$ | $0.54 \pm 0.08$ | $0.54 \pm 0.08$ |
| 0.5 | $\underline{0.20 \pm 0.06}$ | $\mathbf{0.19 \pm 0.05}$ | $0.53 \pm 0.09$ | $0.53 \pm 0.09$ | $0.53 \pm 0.09$ | $0.53 \pm 0.09$ |
| 1 | $\mathbf{0.20 \pm 0.04}$ | $\mathbf{0.20 \pm 0.04}$ | $0.56 \pm 0.09$ | $0.56 \pm 0.09$ | $0.56 \pm 0.09$ | $0.56 \pm 0.09$ |
| 2 | $\mathbf{0.21 \pm 0.06}$ | $\mathbf{0.21 \pm 0.06}$ | $0.55 \pm 0.11$ | $0.55 \pm 0.11$ | $0.55 \pm 0.11$ | $0.55 \pm 0.11$ |
| 4 | $\mathbf{0.20 \pm 0.03}$ | $\mathbf{0.20 \pm 0.03}$ | $0.50 \pm 0.10$ | $0.50 \pm 0.10$ | $0.50 \pm 0.10$ | $0.50 \pm 0.10$ |

Table 10: Comparison to neural networks on the multi-curvature regression task.

| $\mathcal{P}$ | Product DT | Product RF | $T_{\mu_0}\mathcal{P}$ MLP | MLP | $T_{\mu_0}\mathcal{P}$ GNN | GNN |
|---|---|---|---|---|---|---|
| $\mathbb{E}^4$ | $\underline{0.27 \pm 0.04}$ | $\mathbf{0.20 \pm 0.05}$ | $0.56 \pm 0.09$ | $0.56 \pm 0.09$ | $0.56 \pm 0.09$ | $0.56 \pm 0.09$ |
| $\mathbb{H}^4$ | $\underline{0.24 \pm 0.06}$ | $\mathbf{0.18 \pm 0.04}$ | $0.54 \pm 0.10$ | $0.54 \pm 0.10$ | $0.54 \pm 0.10$ | $0.54 \pm 0.10$ |
| $\mathbb{H}^2\mathbb{E}^2$ | $\underline{0.26 \pm 0.05}$ | $\mathbf{0.20 \pm 0.03}$ | $0.54 \pm 0.12$ | $0.54 \pm 0.12$ | $0.54 \pm 0.12$ | $0.54 \pm 0.12$ |
| $(\mathbb{H}^2)^2$ | $\underline{0.24 \pm 0.06}$ | $\mathbf{0.18 \pm 0.04}$ | $0.52 \pm 0.11$ | $0.52 \pm 0.11$ | $0.52 \pm 0.11$ | $0.52 \pm 0.11$ |
| $\mathbb{H}^2\mathbb{S}^2$ | $\underline{0.26 \pm 0.06}$ | $\mathbf{0.20 \pm 0.04}$ | $0.52 \pm 0.11$ | $0.52 \pm 0.11$ | $0.52 \pm 0.11$ | $0.52 \pm 0.11$ |
| $\mathbb{S}^4$ | $\underline{0.25 \pm 0.02}$ | $\mathbf{0.19 \pm 0.03}$ | $0.53 \pm 0.10$ | $0.53 \pm 0.10$ | $0.53 \pm 0.10$ | $0.53 \pm 0.10$ |
| $\mathbb{S}^2\mathbb{E}^2$ | $\underline{0.26 \pm 0.08}$ | $\mathbf{0.19 \pm 0.06}$ | $0.51 \pm 0.10$ | $0.51 \pm 0.10$ | $0.51 \pm 0.10$ | $0.51 \pm 0.10$ |
| $(\mathbb{S}^2)^2$ | $\underline{0.28 \pm 0.06}$ | $\mathbf{0.20 \pm 0.04}$ | $0.54 \pm 0.11$ | $0.54 \pm 0.11$ | $0.54 \pm 0.11$ | $0.54 \pm 0.11$ |

Table 11: Comparison to neural networks on graph node classification/regression tasks (Citeseer, Cora, Polblogs are classification tasks; CS PhDs is a regression task).

| Dataset | Product DT | Product RF | $T_{\mu_0}\mathcal{P}$ MLP | MLP | $T_{\mu_0}\mathcal{P}$ GNN | GNN |
|---|---|---|---|---|---|---|
| Citeseer | $0.23 \pm 0.04$ | $\mathbf{0.24 \pm 0.04}$ | $0.23 \pm 0.03$ | $\mathbf{0.24 \pm 0.02}$ | $\mathbf{0.24 \pm 0.03}$ | $\mathbf{0.24 \pm 0.03}$ |
| Cora | $0.19 \pm 0.03$ | $0.22 \pm 0.04$ | $\mathbf{0.29 \pm 0.03}$ | $\mathbf{0.29 \pm 0.03}$ | $\mathbf{0.29 \pm 0.03}$ | $\mathbf{0.29 \pm 0.03}$ |
| Polblogs | $0.89 \pm 0.19$ | $\underline{0.93 \pm 0.02}$ | $0.85 \pm 0.25$ | $\mathbf{0.94 \pm 0.03}$ | $0.88 \pm 0.17$ | $0.82 \pm 0.38$ |

Table 12: Complexity comparison of machine learning models where: $n$: number of samples, $d$: number of features, $h$: neurons per layer, $L$: number of layers, $D$: maximum tree depth, $s$: number of support vectors. We include the complexity of computing pairwise distance, which are necessary for operating models like $k$-nearest neighbors and GNNs without topologies, as well.

| Model | Phase | Time | | Space | |
| --- | --- | --- | --- | --- | --- |
| | | Worst | Avg | Worst | Avg |
| Dists | | $n^2d$ | $n^2d$ | $n^2$ | $n^2$ |
| MLP | Train | $ndh + Lnh^2$ | $ndh + Lnh^2$ | $nd + dh + L(h^2 + nh)$ | $h^2L$ |
| | Test | $h^2L$ | $h^2L$ | $h^2L$ | $h^2L$ |
| Perceptron | Train | $nd$ | $nd$ | $d$ | $d$ |
| | Test | $d$ | $d$ | $d$ | $d$ |
| SVM | Train | $n^3d$ | $n^2d$ | $n^2$ | $n^2$ |
| | Test | $sd$ | $sd$ | $sd$ | $sd$ |
| GNN | Train | $n^2d$ | $n^2d$ | $n^2$ | $n^2$ |
| | Test | $n^2$ | $n^2$ | $n^2$ | $n^2$ |
| k-NN | Train | $1$ | $1$ | $nd$ | $nd$ |
| | Test | $nd + n\log n$ | $\log n$ | $nd$ | $nd$ |
| Decision Tree | Train | $Dnd$ | $Dnd$ | $2^D$ | $2^D$ |
| | Test | $D$ | $D$ | $1$ | $1$ |
| ProductDT (vanilla) | Train | $Dnd$ | $Dnd$ | $2^D$ | $2^D$ |
| | Test | $d + D$ | $d + D$ | $d$ | $d$ |
| ProductDT ($\binom{d}{2}$ splits) | Train | $Dnd^2$ | $Dnd^2$ | $2^D$ | $2^D$ |
| | Test | $d^2 + D$ | $d^2 + D$ | $d^2$ | $d^2$ |

## J  RUNTIMES AND COMPLEXITY

We summarize complexities for models used in this paper, as well as the pairwise distance preprocessing necessary for operations such as computing nearest neighbors and creating reasonable graph edges for GNNs, in Table 12. Complexity estimates are adapted from Virgolin (2021).

To see that the training time complexity of ProductDT is $O(Dnd)$, observe that we must first preprocess the data into angles, which takes $O(nd)$ operations. From there, the angular comparison is a constant-time modification to the decision tree algorithm, so the complexity of ProductDT is $O(nd + Dnd) = O(nd)$. For inference, asymptotic performance is slightly slower than decision trees because preprocessing an input requires $O(d)$ operations.

If using all $\binom{d}{2}$ 2-D projections, training time complexities are all multiplied by $d$, and the $O(d^2)$ preprocessing step is added to test time complexities.

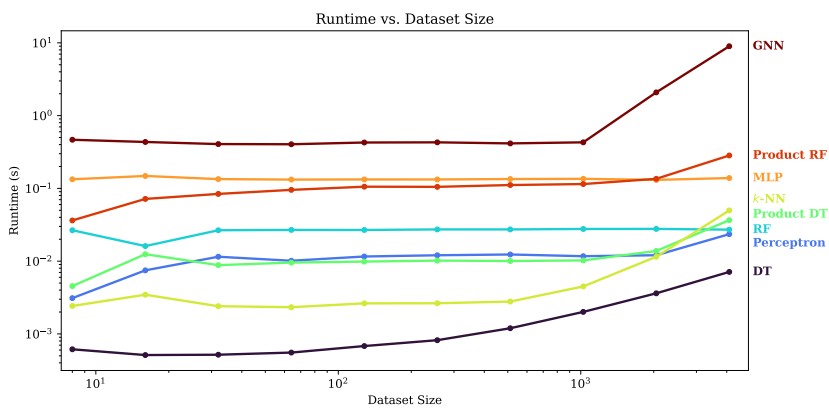

Figure 10: Runtime comparison for all of our methods

# K INTERPRETABILITY AND VISUALIZATION

Alongside their demonstrated accuracy and efficiency, decision tree algorithms are attractive for their tractability and interpretability. In particular, given a trained decision tree $\mathcal{T}$, it is possible to:

1. Predict its behavior on the entire space of possible inputs (equivalently: $\mathcal{T}$ partitions $\mathcal{P}$ in a tractable way).

2. Determine the importance of features (for classic decision trees) or feature pairs/components (for ProductDT) by observing how often and how early a feature(/pair/component) is used in the decision tree procedure. Heuristically, early-splitting features are more important.

3. Visualize every node using a 2-dimensional projection of the input data and angle

## K.1 SUBMANIFOLD-LEVEL ATTRIBUTION EXPERIMENT

To determine whether our method could accurately distinguish between relevant and irrelevant submanifolds, we drew independent samples from Gaussian mixtures in $\mathbb{H}^2$, $\mathbb{E}^2$, and $\mathbb{S}^2$, and yielding datasets $(\mathbf{X}_{\mathbb{H}}, \mathbf{y}_{\mathbb{H}}), (\mathbf{X}_{\mathbb{E}}, \mathbf{y}_{\mathbb{E}}), (\mathbf{X}_{\mathbb{S}}, \mathbf{y}_{\mathbb{S}})$. We then concatenated these embeddings together:

$$\mathbf{X}_{\mathcal{P}} = \mathbf{X}_{\mathbb{H}} \oplus \mathbf{X}_{\mathbb{E}} \oplus \mathbf{X}_{\mathbb{S}}. \tag{75}$$

We trained three separate decision tree models on $\mathbf{X}_{\mathcal{P}}$, using $\mathbf{y}_{\mathbb{H}}, \mathbf{y}_{\mathbb{E}}$, and $\mathbf{y}_{\mathbb{S}}$ as labels. Because the labels and embeddings were drawn independently, it should be the case that only the component from the same manifold as the labels contains any relevant information, and the other two components are simply noise. Therefore, measuring the fraction of splits that fall in the "correct" manifold is a useful proxy for understanding tree models' ability to pick out signal that happens in individual component manifolds.

Our results are summarized in Table 13. We found that both product space and ambient decision trees perform well at this task, which is to be expected.

We note that this analysis is unique to tree methods, where the split dimensions are part of the architecture; other methods, such as perceptrons, $k$-nearest neighbors, or neural networks are harder to query for feature(/component) importances. Therefore, we consider this simple experiment a useful demonstration of how decision tree learning can reveal aspects of structure in mixed-curvature datasets that other learning algorithms cannot reveal.

Table 13: Intepretability outcomes for Gaussian mixture. Percentages reflect the proportion of splits in the trained decision tree which fell in the non-spurious component manifold.

| Model | $\mathbb{H}^2$ | $\mathbb{E}^2$ | $\mathbb{S}^2$ |
|---|---|---|---|
| Product DT | 100% | 83% | 86% |
| Ambient DT | 100% | 83% | 67% |

## K.2 VISUALIZATION

A trained tree gives us all of the information we need to visualize the data and how it is split at every node, since each node looks at a 2-dimensional projection. We display three levels of a decision tree with a max depth of 3 in Figure 11. Note that, in this case, the decision tree also gives us relevant information about which 2-dimensional projections are worth looking at on the basis of their feature importances.

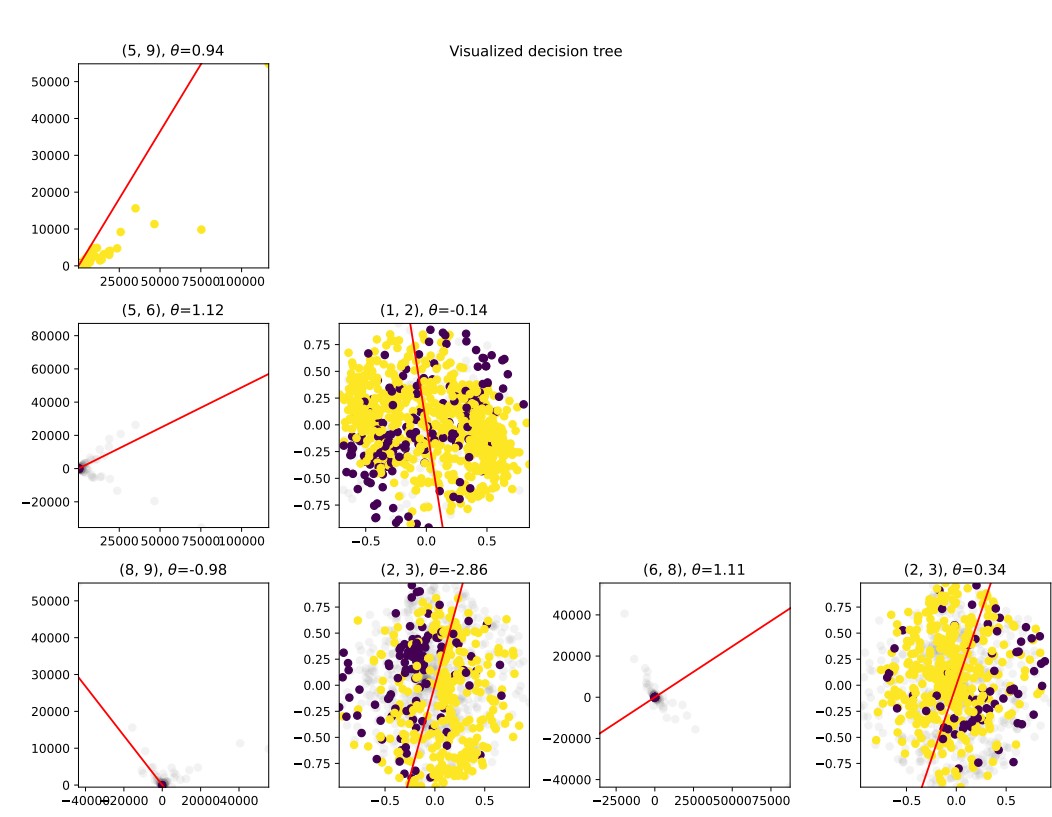

Figure 11: An example of a visualized decision tree for a Gaussian mixture in $\mathcal{P} = \mathbb{S}^4 \mathbb{H}^4$. Greyed-out points are discarded "earlier" in the tree.

