# OpenReview forum: "Mixed-curvature decision trees and random forests"
_ICLR.cc/2025/Conference — Submitted to ICLR 2025_

### Official Review · Reviewer_ZS6K · 2024-10-17

**Soundness:** 2
**Presentation:** 3
**Contribution:** 3
**Rating:** 5
**Confidence:** 4

**Summary:**

The authors propose an extension of decision trees and random forests, which typically assume Euclidean space, to mixed-curvature representations using product manifolds based on constant curvature spaces: Euclidean space, hyperspherical space, and hyperbolic space (focusing on the hyperboloid model).

**Strengths:**

- The paper generalizes decision trees and random forests to mixed-curvature representations which can lead to more expressive/flexible classifiers.
- They leverage ideas from previous papers in deep learning which use mixed-curvature, to improve random forests and decision trees.
- The proposed model may have applications in biology: which I believe is true and the authors mention in the introduction, but they do not really test it in the paper.

**Weaknesses:**

- It seems that the method suffers from similar issues to those discussed in: Neural Latent Geometry Search: Product Manifold Inference via Gromov-Hausdorff-Informed Bayesian Optimization. How does one select the product manifold signature? This should be discussed as part of the limitations. The paper mentioned above suggests using Bayesian optimization to select the signature, for instance.
- In general section 4 is poorly written, there are a lot of baselines, but it is unclear how DTs and RFs are used in different tasks. It is not clearly explained. Maybe additional explanations about the different tasks in the appendix would be helpful: that is, describing the problem setup rather than mainly focusing on reporting hyperaparameters.
- Table 2 has too many symbols. It is not clearly explained what is meant by Max Ambient and Max Product. Could you please define them in the main text or the caption.
- In table 2, for Cora and Citeseer if you are doing node classification, the performance is surprisingly low. What would be the results in terms of accuracy? In the case of Cora and Citeseer is more typical to see the results in accuracy than F1, so it would be helpful to know what they are to compare for instance to the performance of a GCN.
- In Section 4.4. the authors mention they use variational autoencoders for classification. Do you mean autoencoders? Variational autoencoders are generally used for generative modeling. Or do you mean you use the KL loss to regularize the latent space but then use it for classification. It is unclear how classification is performed: in the regularized latent space of an autoencoder? The performance seems very low too for MNIST, etc. KL regularization may be hurting classification performance, if you remove it the autoencoder has more freedom to encode the samples in the latent space without having to stay Gaussian-like and they are probably easier to classify. Could you provide a step-by-step description of the experimental procedure used please.
- The conclusion section lacks a discussion on limitations. As previously mentioned: it is not obvious how to choose the signature. Also, why use RF when there are other better performing methods for most of the tasks discussed in the experiment section? One could argue that RFs may be easier to implement for less experienced practitioners, but introducing the mixed-curvature representations makes the method more unaccessible to a general audience. Potential points to discuss: 1) Challenges in selecting appropriate signatures, 2) Trade-offs between RFs with mixed-curvature representations versus other high-performing methods, 3) Potential barriers to adoption for practitioners less familiar with non-Euclidean geometries

**Questions:**

Suggestions:

- It should be made clear in the text that product manifolds based on constant curvature manifolds cannot generate any arbitrary manifold, as part of limitations.

- Relevant works are missing:

Historical context on manifold learning and machine learning: Algorithms for manifold learning by Cayton, Representation Learning: A Review and New Perspectives by Bengio

Hyperbolic spaces and machine learning: Before product manifolds, there were many works on using hyperbolic spaces which served as predecessor which are worth mentioning, for instance, Poincare embeddings for learning hierarchical representations, Neural embeddings of graphs in hyperbolic space, Hyperbolic entailment cones for learning hierarchical embeddings

Product manifolds and product manifold derived features: Heterogeneous manifolds for curvature-aware graph
 embedding, Neural Spacetimes for DAG Representation Learning

Computationally tractable manifolds beyong constant curvature manifolds: One of the motivations for using product manifolds of Euclidean, hyperbolic, and hyperspherical spaces, is that they allow machine learning algorithms to model data on more general manifolds than Euclidean space, while still providing a closed-form solution for geodesic distances, exponential maps, etc. However, product manifolds of this kind are also limited: it is not possible to model any arbitrary manifold by just using this construction. Other computationally tractable alternatives also exist: Neural Snowflakes: Universal Latent Graph Inference via Trainable Latent Geometries, which is based on fractals, and Computationally tractable riemannian manifolds for graph embeddings, which is based on matrix manifolds.

- Curvature K is not defined in Section 2.1.2 for hyperspherical space.

- In equation 10 the wrong symbol is used for Cartesian product
- In equation 12 the wrong symbol is used for Concatenation

---

> ### Author Response · Authors · 2024-11-23
>
> We appreciate the reviewer’s careful reading of our paper, recognition of our contributions, and detailed edits. In particular, we appreciate the reviewer’s excellent suggestions for relevant literature on mixed-curvature representations, which we have incorporated into our resubmission and which greatly improve our Related Work section.
>
> In the interest of starting a dialogue early, we are posting our response even as we improve our paper, per the reviewer’s comments. We will further post here upon submitting updates to the paper.
>
> We will respond to the remaining comments inline:
> > The proposed model may have applications in biology: which I believe is true and the authors mention in the introduction, but they do not really test it in the paper.
>
> While we are excited about the applications of mixed-curvature embeddings (and, hopefully, our decision trees) to biological data, we think a full investigation into this application is out of scope for this paper. In our view, this is a methods paper proposing and evaluating a novel method for classification/regression on product manifold-valued data, and therefore we restrict ourselves to well-known datasets embedded using established methods from the literature. We believe this approach is more appropriate to ICLR as a venue, and well-suited for a paper introducing a novel method. An exploration of biological methods is best left for future work, in venues catering to a more biologically-inclined readership.
> However, we would like to emphasize that we intentionally included several biological datasets across various domains and data modalities in our benchmarks:
> * Blood cell type classification (VAE experiments in Table 2)
> * Lymphoma case versus control transcriptome classification (VAE experiments in Table 2)
> * Allen Brain Atlas neuron spiking prediction (neurons 33 and 46) (Empirical experiments in Table 2)
>
> > It seems that the method suffers from similar issues to those discussed in: Neural Latent Geometry Search: Product Manifold Inference via Gromov-Hausdorff-Informed Bayesian Optimization. How does one select the product manifold signature? This should be discussed as part of the limitations. The paper mentioned above suggests using Bayesian optimization to select the signature, for instance.
>
> We thank the reader for this comment, and would like to mention that we are also very interested in the question of signature estimation. We also thank the reader for pointing us to this very interesting paper, which we were not previously aware of. We agree that this work is relevant to the overall project of learning and using mixed-curvature embeddings, and we believe our work is complementary to it insofar as it can help practitioners make the most of high-quality embeddings in downstream tasks.
>
> In our initial submission, we sought to deemphasize the question of signature selection on the grounds that we consider the problem of regression/classification/link prediction to be “downstream” of embedding generation. Since, we additionally have no reason to believe a priori that signature selection and overall embedding quality would systematically bias our evaluations in favor of any particular classifier, we use standard/basic methods from the literature to generate embeddings.
>
> The reviewer is correct to point out that these points are treated implicitly in our paper. We will add a limitations section that makes our position explicit and refers interested readers to the signature estimation literature, and in particular to the reviewer’s suggested paper.

---

> > ### Author Response · Authors · 2024-11-23
> >
> > > In general section 4 is poorly written, there are a lot of baselines, but it is unclear how DTs and RFs are used in different tasks. It is not clearly explained. Maybe additional explanations about the different tasks in the appendix would be helpful: that is, describing the problem setup rather than mainly focusing on reporting hyperaparameters.
> >
> > We thank the reviewer for this suggestion. We believe the following improvements will address these concerns about presentation:
> > * At the very beginning of Section 4, describe the experimental setting:
> >   * Results tables are grouped according to the method used to generate embeddings (gaussian mixtures, graph embeddings, VAE latent space encoders, direct transformation to hyperspherical coordinates)
> >   * All data is split in the same fashion
> >   * Evaluate F1/RMSE using 8 predictors (k-NN, product space perceptron, tangent plane DT, tangent plane RF, euclidean DT, euclidean RF, product space DT, product space RF).
> > * We rearrange subsections to reflect the nontrivial data generation before predictor training and evaluation: we will move the “Baselines and parameters” subsection after the dataset descriptions, which we will aggregate into a single “Datasets” subsection.
> >
> > That is, the final structure of section 4 will be:
> >
> > Benchmarks
> >   * Brief overview of the problem setup
> >   * Brief overview of the figures/tables
> >   * Datasets (with links to corresponding Appendix sections)
> >      Synthetic data
> >     * Graph embeddings
> >     * Mixed-curvature VAE latent space
> >     * Other (direct transformations)
> >   * Baselines
> >   * Results - with very brief commentary
> >
> > We will also rearrange the Appendix to match the order of results by merging Appendix A and Appendix E, and moving the baselines/predictor hyperparameters section after the embedding details.
> >
> > Table 2 has too many symbols. It is not clearly explained what is meant by Max Ambient and Max Product. Could you please define them in the main text or the caption.
> >
> > > Thanks to the reviewer’s comment, we expanded the unclear description of Max Ambient and Max Product and rewritten it as:
> >
> > “F1 scores for all classification and link prediction (LP) benchmarks. Best predictors are shown in bold, while second-best predictors are underlined. Superscripts indicate statistical significance (see footnote). For brevity, we omit columns for low-performing methods and merge DT and RF columns (e.g. “Ambient” means max(mean(Ambient DT), mean(Ambient RF)).)”
> >
> > Consistent with superscript notation, we have made footnotes for statistical significance symbols and put Tables 2 and 3 on the same page. To improve the readability of the main body of the paper, we have followed the reviewer’s suggestion and removed the statistical significance annotations from Tables 2 and 3, and have left them in the complete results table in the Appendix.
> >
> > > In table 2, for Cora and Citeseer if you are doing node classification, the performance is surprisingly low. What would be the results in terms of accuracy? In the case of Cora and Citeseer is more typical to see the results in accuracy than F1, so it would be helpful to know what they are to compare for instance to the performance of a GCN.
> >
> > We acknowledge that many methods perform better than ours on graph benchmarks. We do not expect to achieve globally competitive performance on these tasks because 1) our embedding spaces are very small, 2) we use classic, simple techniques for generating graph embeddings, and 3) our predictors do not have access to the graph topology.
> >
> > The goal of our graph embedding pipeline is simply to generate a low-dimensional, mixed-curvature embedding associated with a challenging classification task. Because we have no reason to believe that benchmarking in this setting introduces bias in favor of any particular model, we believe that these results are robustly favorable for our model.
> >
> > Nevertheless, we see how such additional results could be illuminating to readers with a stronger interest in graph benchmarks. We are rerunning our graph benchmarks and will include accuracy scores and comparisons to GCNs along with a section in the Appendix reporting our results.

---

> > > ### Author Response · Authors · 2024-11-23
> > >
> > > > In Section 4.4. the authors mention they use variational autoencoders for classification. Do you mean autoencoders? Variational autoencoders are generally used for generative modeling. Or do you mean you use the KL loss to regularize the latent space but then use it for classification. It is unclear how classification is performed: in the regularized latent space of an autoencoder? The performance seems very low too for MNIST, etc. KL regularization may be hurting classification performance, if you remove it the autoencoder has more freedom to encode the samples in the latent space without having to stay Gaussian-like and they are probably easier to classify. Could you provide a step-by-step description of the experimental procedure used please.
> > >
> > > The VAE is used to generate training data, not to perform classification. In Section E.3 of the Appendix, we detail our method, which follows the approach shown in both Tabaghi et al (2023) and Skopek et al (2019):
> > > 1. Train a VAE with a mixed-curvature latent space on the training set
> > > 2. Once the VAE is trained, encode both the training and test set using the learned encoder. The mixed-curvature embeddings of the data are the means returned by the encoder. Do not sample.
> > > 3. The rest of the evaluation is standard: train each model on the encoded training set and evaluate on the encoded test set.
> > >
> > > The reviewer is correct to point out that scores are low for all classifiers on VAE-encoded data: we were unable to reproduce the product space perceptron accuracies reported in Tabaghi et al (2023) for the Blood, CIFAR-100, and Lymphoma datasets (though we beat their perceptron algorithm in our own benchmarks). We note that we use the same architecture and hyperparameters for training the VAE.
> > >
> > > As with the graph benchmarks, we would like to reiterate that we expect any issues with embeddings to affect all classifiers similarly, thus we believe that evaluating our classifiers in more challenging settings arising from small latent spaces and noisy embeddings provides a sound comparison of their performance.
> > >
> > > > The conclusion section lacks a discussion on limitations. As previously mentioned: it is not obvious how to choose the signature. Also, why use RF when there are other better performing methods for most of the tasks discussed in the experiment section? One could argue that RFs may be easier to implement for less experienced practitioners, but introducing the mixed-curvature representations makes the method more unaccessible to a general audience. Potential points to discuss: 1) Challenges in selecting appropriate signatures, 2) Trade-offs between RFs with mixed-curvature representations versus other high-performing methods, 3) Potential barriers to adoption for practitioners less familiar with non-Euclidean geometries
> > >
> > > We thank the reviewer for the detailed suggestion for a limitations section, which we will include in our revised draft. We believe all three points mentioned are relevant, and also add a fourth point noting that the rotation-invariance of many non-Euclidean embeddings deprives the basis dimensions of the strong semantics seen in tabular data and other domains for which decision trees are typically well-suited.
> > >
> > > We also appreciate the point about the inaccessibility of non-Euclidean methods for many practitioners. We faced a fractured landscape of incompatible software when we started benchmarking our methods, and have taken pains to correct this for our paper. Although we do not emphasize this in the text of the original submission, we have included source code with our submission that unifies embeddings generation, manifold methods, and mixed-curvature classifiers in a single Torch implementation. Once the paper is deanonymized, we will update it with links to our repository. We hope this repository will help practitioners easily access methods for generating and working with non-Euclidean embeddings, and we can continue to expand it as this field continues to grow.

---

> > > > ### Author Response · Authors · 2024-11-23
> > > >
> > > > > It should be made clear in the text that product manifolds based on constant curvature manifolds cannot generate any arbitrary manifold, as part of limitations.
> > > >
> > > > We have added this to our limitations. We have chosen Neural Snowflakes to cite for the claim that product spaces cannot isometrically embed an arbitrary graph topology.
> > > >
> > > >
> > > > > Relevant works are missing: [...]
> > > >
> > > > We appreciate this thoughtful overview. We initially wanted to restrict our related work discussion more strictly to product spaces; however, this suggestion has changed our minds. Accordingly, we have added these works, and several others, to the related work section. We believe that this has greatly improved the related work, and our paper is now much more coherently placed in the context of existing non-Euclidean representation learning work.
> > > >
> > > > > Curvature K is not defined in Section 2.1.2 for hyperspherical space.
> > > >
> > > > We have added a definition for curvature in Section 2.1.2
> > > >
> > > > > In equation 10 the wrong symbol is used for Cartesian product
> > > >
> > > > We have rewritten the Cartesian product explicitly.
> > > >
> > > > > In equation 12 the wrong symbol is used for Concatenation
> > > >
> > > > We have rewritten this equation using the direct product ($\bigoplus$) following the convention in Tabaghi et al (2023).

---

> > > > > ### Comment · Reviewer_ZS6K · 2024-11-23
> > > > > **Updated version**
> > > > >
> > > > > Dear authors,
> > > > >
> > > > > Thank you for your response. Could you upload the updated paper with changes highlighted in red so I can give feedback and make a decision.

---

> > > > > > ### Comment · Reviewer_ZS6K · 2024-11-26
> > > > > >
> > > > > > Given that the authors have not provided a revised version of the paper I will maintain my rating.

---

> > > > > > > ### Author Response · Authors · 2024-11-26
> > > > > > >
> > > > > > > We apologize for the delayed submission of our revised manuscript. We have uploaded a new version with changes highlighted in red, per the reviewer's request. We draw the reviewer's attention to the following changes, which respond specifically to points raised in their review:
> > > > > > > * The updated Related work section
> > > > > > > * The updated Conclusion (with limitations and future work)
> > > > > > > * The rewritten Benchmarks section
> > > > > > > * The revised Tables 2 and 3, which remove the statistical significance symbols
> > > > > > > * Appendix Section A.I, Comparison to Neural Baselines, which includes both MLP/GNN benchmarks and accuracies for our methods, as requested.
> > > > > > >
> > > > > > > We hope that this revised manuscript addresses the reviewer's concerns, and will happily continue this discussion throughout the extended discussion period. We appreciate the reviewer's patience and continued consideration of our work.

---

> > > > > > > > ### Comment · Reviewer_ZS6K · 2024-11-28
> > > > > > > >
> > > > > > > > Thank you for the clarifications.
> > > > > > > >
> > > > > > > > I agree that applications in biology are not critical and you already consider relevant datasets, so I do not wish to be insistent on that. The clarification regarding the VAE experiments in the updated text is also useful, I did not understand it before.
> > > > > > > >
> > > > > > > > Overall I think the manuscript has improved substantially during the review period and I will raise my score to a 5.
> > > > > > > >
> > > > > > > > The reason I did not raise my score further is that I remain concerned about the experiments, and I believe other reviewers share similar concerns. I attempted to examine the supplementary code, but it lacks a README or clear guidelines for reproducing the experiments. Additionally, I noticed some files, such as signature_estimation.py, appear to be incomplete, containing only a raise NotImplementedError statement for the curvature selection methods.
> > > > > > > >
> > > > > > > > I acknowledge the extensive experiments you have conducted. However, moving forward, I suggest emphasizing the main experiments most relevant to your method in the main paper, while reserving other ablations and setups for the appendix. Ensuring the reproducibility of your code is equally important. For instance, in the context of graph embeddings, if your method is not particularly suited for such tasks, it might be unnecessary to include it in the main text.
> > > > > > > >
> > > > > > > > My main question is: What is this method specifically designed for? What types of tasks should practitioners aim to use it for? I have not been able to extract particular conclusions to these questions as part of the experimental section.

---

> > > > > > > > > ### Author Response · Authors · 2024-12-02
> > > > > > > > > **Further response to ZS6K**
> > > > > > > > >
> > > > > > > > > We thank the reviewer for their continued attention to our work, and for improving their score in response to our rebuttal. We hope that our additional benchmarks and the rest of this response will clear up remaining concerns around experiments, results focus, and reproducibility.
> > > > > > > > >
> > > > > > > > > We have posted a General Comment titled "Updated and expanded neural net benchmarks" with updated tables (in Markdown format, as we can no longer submit revised drafts of our paper on OpenReview). These tables include a full suite of benchmarks and reflect a more competitive set of neural network architectures/hyperparameters; the overall content of the tables remains favorable to our method. In particular, we were the best on 38 out of 57 benchmarks, and top-2 on 50 out of 57 benchmarks.
> > > > > > > > >
> > > > > > > > > We apologize for the missing reproducibility information/README files in our code submission. In general, we attempted to write `dataloaders.py` and `benchmarking.py` in such a way as to minimize the amount of code needed to run specific benchmarks, but we have omitted the notebooks we actually used to run them. The full Github repository, which we will include in the camera-ready if accepted, will contain README files as well as the Jupyter notebooks we used to run our benchmarks.
> > > > > > > > >
> > > > > > > > > We also apologize for including in the placeholder file `signature_estimation.py` in our submission. We hope this is evidence both for our genuine interest in this question of signature estimation for future work, and for our intent to keep the work in this manuscript independent of the question of signature selection. Since the supplementary files submission, we have implemented a number of extra functions to estimate curvature/signature:
> > > > > > > > > * Sectional curvature, as described in Gu et al, 2019 [1] (this is the placeholder `sectional_curvature_method` in `signature_estimation.py`).
> > > > > > > > > * Gromov delta-hyperbolicity, as described in Khrulkov et al, 2020 [2].
> > > > > > > > > * Scalar curvature intrinsic estimator, as described in Hickok and Blumberg, 2023 [3].
> > > > > > > > >
> > > > > > > > > In the future, we are also interested in testing more sophisticated signature estimation methods, including:
> > > > > > > > > * The greedy signature estimator used in Tabaghi et al, 2022 [4] (this is the placeholder `greedy_curvature_method` in `signature_estimation.py`).
> > > > > > > > > * The Bayesian optimizer described in Borde et al, 2023 [5].
> > > > > > > > >
> > > > > > > > > Regarding experimental focus, we view the core purpose of this work as proposing a new method for classification and regression that works with mixed-curvature embeddings. We sought to benchmark on a diverse a sample of mixed-curvature data sources, including prominent techniques from the literature (Gu et al [1] method and mixed-curvature VAE [6]), synthetic data, and data that naturally embeds in product manifolds such as geospatial data. This was our reasoning for including graph benchmarks in the main body of the paper; as we find that our results are fairly competitive with neural methods even for graph benchmarks, we are inclined to leave them in so that users can see our performance across a variety of data types at a glance.
> > > > > > > > >
> > > > > > > > > Our hope is that this technique will be useful for probing mixed-curvature representations on the fly with minimal hyperparameter tuning and computational overhead. The interpretability experiments that we describe in the new Appendix K furnish further examples of how trained probes can be used to understand the relationship between classification/regression tasks and embedding geometry in a way that is unique to decision tree algorithms.
> > > > > > > > >
> > > > > > > > > **References:**
> > > > > > > > >
> > > > > > > > > [1] Albert Gu, Frederic Sala, Beliz Gunel, Christopher Ré. Learning Mixed-Curvature Representations in Product Spaces. ICLR 2019.
> > > > > > > > >
> > > > > > > > > [2] Valentin Khrulkov, Leyla Mirvakhabova, Evgeniya Ustinova, Ivan Oseledets, Victor Lempitsky. Hyperbolic Image Embeddings. arXiv 2020.
> > > > > > > > >
> > > > > > > > > [3] Abigail Hickok, Andrew J. Blumberg. An Intrinsic Approach to Scalar-Curvature Estimation for Point Clouds. arXiv 2023.
> > > > > > > > >
> > > > > > > > > [4] Puoya Tabaghi, Chao Pan, Eli Chien, Jianhao Peng, Olgica Milenkovic. Linear Classifiers in Product Space Forms. arXiv 2021.
> > > > > > > > >
> > > > > > > > > [5] Haitz Sáez de Ocáriz Borde, Alvaro Arroyo, Ismael Morales, Ingmar Posner, Xiaowen Dong. Neural Latent Geometry Search: Product Manifold Inference via Gromov-Hausdorff-Informed Bayesian Optimization. NeurIPS 2023.
> > > > > > > > >
> > > > > > > > > [6] Ondrej Skopek, Octavian-Eugen Ganea, Gary Bécigneul. Mixed-curvature Variational Autoencoders. arXiv 2019.

---

### Official Review · Reviewer_iVqG · 2024-10-23

**Soundness:** 3
**Presentation:** 4
**Contribution:** 2
**Rating:** 5
**Confidence:** 3

**Summary:**

This paper proposes a method for training decision trees that takes into account a product manifold structure, formed by combining manifolds with constant curvature. A notable technical contribution is the angular reformulation of tree splits. The effectiveness of the proposed method is demonstrated through experiments on both synthetic and real-world datasets.

**Strengths:**

The paper provides a thorough background knowledge, making it accessible even to readers unfamiliar with geometry and decision trees. Visualization quality is also good. Additionally, the experimental settings described in the Appendix are detailed, and the paper overall does not have unclear sections.

**Weaknesses:**

While the paper makes a solid contribution, in my understanding, it falls short in terms of significance and novelty. For instance, Section 3, which serves as the core of the proposed method, raises some concerns. Section 3.1 discusses Euclidean decision trees, which is straightforward. Section 3.2 builds on existing research. Although Section 3.3 may introduce a new perspective, the authors themselves describe it as “quite simple”. The choice of does not seem to involve a particularly non-trivial approach. Furthermore, Section 3.4 primarily combines elements from Sections 3.1 through 3.3. While the paper mentions the advantages of redefining decision tree learning through angular comparisons, it is difficult to claim that these benefits are especially significant.

Additionally, although the experiments are well-executed, the impact of the proposed method could be better highlighted by including a more diverse set of baselines for comparison. For example, it would be beneficial to compare the method against non-tree models or conventional decision trees applied to real-world data, such as traffic data, without considering manifold structures. Even if the technical novelty is limited, demonstrating significant experimental results would enhance the paper’s relevance to the machine learning community. However, the current experimental results do not sufficiently showcase the advantages of the proposed method.

**Questions:**

- I believe it would be helpful to clearly identify the technical differences between your proposed approach and previous research. As I reviewed your method, I struggled to find any significant technical novelty, as indicated in the Weaknesses section. If I have misunderstood something, please kindly let me know.

- Is it possible to consider oblique decision trees? Specifically, can the method be extended to split not only within the same type of space but also across different spaces with oblique boundaries?

- Regarding the third contribution listed—“We extend techniques for sampling distributions in non-Euclidean manifolds to describe mixtures of Gaussians in product manifolds”—am I correct in understanding that this pertains to the creation of datasets, as described in Section 4.1 and Appendix A, rather than being directly related to learning decision trees on product manifolds?

- In the Introduction, the intention to use decision trees and random forests is mentioned in the fourth paragraph. What is the motivation behind developing these tree-based models? If the motivation lies in accuracy, I believe a comparison with other non-tree-based models would be necessary. On the other hand, if the choice is driven by factors like interpretability, then it would be beneficial to provide experimental results or analysis supporting this aspect.

---

> ### Author Response · Authors · 2024-11-23
>
> We thank the reviewer for their very positive feedback on the quality of our writing, visualization, and experiments, as well as acknowledging that our research constitutes a meaningful contribution to the literature. We respond to specific comments inline:
>
> > While the paper makes a solid contribution, in my understanding, it falls short in terms of significance and novelty. For instance, Section 3, which serves as the core of the proposed method, raises some concerns. Section 3.1 discusses Euclidean decision trees, which is straightforward. Section 3.2 builds on existing research. Although Section 3.3 may introduce a new perspective, the authors themselves describe it as “quite simple”. The choice of does not seem to involve a particularly non-trivial approach. Furthermore, Section 3.4 primarily combines elements from Sections 3.1 through 3.3. While the paper mentions the advantages of redefining decision tree learning through angular comparisons, it is difficult to claim that these benefits are especially significant.
>
> We deliberately pursue algorithmic and mathematical simplicity in our paper, which proves essential for practical applications in non-Euclidean spaces. The proliferation of Riemannian manifold operations in conventional methods, and even the reliance on repeated sparse dot products in tools like HyperDT, are prohibitive for high-dimensional data.
>
> The contributions extend beyond mere simplification. We introduce the first mixed-curvature decision tree, the first mixed-curvature regressor of any kind, and the first hyperspherical decision tree. This perspective is particularly relevant given the prevalence of hyperspherical data in modern machine learning (e.g. any L2-normalized datasets).
>
> The work advances the field of mixed-curvature geometry in several concrete ways:
> * It translates a well-known concept (decision trees) into the complicated domain of mixed-curvature embeddings without adding substantial complexity; in doing so, it expands the toolkit available for mixed-curvature data analysis
> * It provides an angular formulation of decision tree splits that unifies decision trees of all curvatures
> * It establishes a framework for pairwise tasks such as link prediction
> * It provides reproducible benchmarks that establish performance baselines for future research
>
> These advances, while building on existing foundations, represent meaningful progress in making non-Euclidean methods more accessible and practical for real-world applications.
>
> > Additionally, although the experiments are well-executed, the impact of the proposed method could be better highlighted by including a more diverse set of baselines for comparison. For example, it would be beneficial to compare the method against non-tree models or conventional decision trees applied to real-world data, such as traffic data, without considering manifold structures. Even if the technical novelty is limited, demonstrating significant experimental results would enhance the paper’s relevance to the machine learning community. However, the current experimental results do not sufficiently showcase the advantages of the proposed method.
>
> We apologize for not clarifying that we already compare to conventional decision trees. We do this in two ways: using ambient coordinates (called “Ambient” in our benchmarks), and using tangent plane coordinates (“Tangent”). In the case of Empirical datasets, we sometimes have the option of also training decision trees on pre-transformation data, though the original feature space tends to be very small, as our embeddings are generated by processing datetime features according to cycles of varying periods. We did not include these benchmarks in our paper.
>
> For transparency, we provide an example of such a benchmark here. On the traffic dataset, training a decision tree on the DateTime column without transformations yields an RMSE of 0.80, and breaking the DateTime column into day of week, hour, day of year, and minute columns without transforming yields an RMSE of 0.78. For reference, RMSEs for all regressors on the transformed data are in the 0.45-0.60 range.
>
> > I believe it would be helpful to clearly identify the technical differences between your proposed approach and previous research. As I reviewed your method, I struggled to find any significant technical novelty, as indicated in the Weaknesses section. If I have misunderstood something, please kindly let me know.
>
> The technical novelty of our paper consists in presenting a novel perspective on decision trees, yielding a reformulation of the CART algorithm that covers all constant curvatures, recovers the known Euclidean and hyperbolic algorithm, and composes to product manifolds. In our benchmarks, we demonstrate the effectiveness of our approach.

---

> > ### Author Response · Authors · 2024-11-23
> >
> > > Is it possible to consider oblique decision trees? Specifically, can the method be extended to split not only within the same type of space but also across different spaces with oblique boundaries?
> >
> > Absolutely! Mixed-curvature decision trees can even be thought of as a kind of oblique decision trees (the angle-under-projection formulation can be thought of as a linear combination of input features), but we retain the overall search structure of CART by choosing the 2-D subspaces in which our decision hyperplanes can fall in advance. We choose CART because it is well-known, not because it is especially well-suited to the problem at hand; we believe this makes it clearest to readers what the essential modifications needed to adapt decision trees to a mixed-curvature are. However, considering oblique decision trees is an excellent suggestion for future work, and we expect it would be very effective.
> >
> > Per Tabaghi et al (2023) [1], for any weight matrix, $\mathbf{wx} = 0$ is a linear classification boundary in mixed curvature spaces, where $\mathbf{w}$ is a weight vector with as many dimensions as there are ambient dimensions in the entire product manifold. Notice that this more general formulation allows $\mathbf{w}$ can be nonzero in multiple component manifolds at once. Once again, this is a specific case of an oblique decision tree where there is no intercept.
> >
> > Well-designed oblique variants of mixed-curvature decision trees could even outperform our current CART-esque approach, as the current inductive bias of preprocessing data points as angles under projections to pairs of basis dimensions privileges the basis dimensions of the original representation. Privileged bases are a cornerstone of classical applications of decision trees on tabular data, but in representation learning contexts where the objective is rotation- (and often translation-) invariant, they are more arbitrary.
> >
> > In the revised draft of our paper, we have added a section to the conclusion that discusses the question of privileged bases in the context of future work, and proposes an extension to oblique decision trees.
> >
> > > Regarding the third contribution listed—“We extend techniques for sampling distributions in non-Euclidean manifolds to describe mixtures of Gaussians in product manifolds”—am I correct in understanding that this pertains to the creation of datasets, as described in Section 4.1 and Appendix A, rather than being directly related to learning decision trees on product manifolds?
> >
> > This is correct. As we mention above, such a method for generating synthetic data (especially classification and regression benchmarks) had not been specified for mixed-curvature manifolds.
> >
> > > In the Introduction, the intention to use decision trees and random forests is mentioned in the fourth paragraph. What is the motivation behind developing these tree-based models? If the motivation lies in accuracy, I believe a comparison with other non-tree-based models would be necessary. On the other hand, if the choice is driven by factors like interpretability, then it would be beneficial to provide experimental results or analysis supporting this aspect.
> >
> > We thank the reviewer for this suggestion, and have rewritten the introduction and conclusions to argue more explicitly for decision tree models in this context. We summarize our arguments as follows:
> > * Since decision tree-based algorithms are among the most widely-used machine learning models, it is natural to include them in a mixed-curvature data analysis toolkit
> > * Decision trees are predictable: their behavior on the entire manifold from which input data comes can be determined purely from the splits. This is not true of neural networks.
> > * Mixed-curvature decision trees have feature importances. Since each split is labeled with a feature (in our case, this corresponds to a 2-D subspace of one of the input manifolds), it is possible to ask which features/component manifolds contribute to a prediction directly from the trained model parameters, without resorting to methods like SHAP.
> >
> > With respect to existing benchmarks, we draw the reader’s attention to the mixed-curvature perceptron and $k$-nearest neighbors benchmarks in the results, which are the two existing non-decision tree, non-neural net methods for working with data on mixed-curvature manifold. We are evaluating neural network-based models and running interpretability experiments, which so far remain favorable to our model. We will post an update when these are complete.
> >
> > **References:**
> > [1] Puoya Tabaghi, Chao Pan, Eli Chien, Jianhao Peng, Olgica Milenkovic. Linear Classifiers in Product Space Forms.

---

> > > ### Comment · Reviewer_iVqG · 2024-11-26
> > >
> > > Thank you for your response. I had intended to reply after the revised paper was submitted; however, given the approaching deadline, I am providing my feedback now. While I appreciate the rebuttal, my concerns remain unresolved. Unless the revised paper significantly addresses these points, I will maintain my current score.

---

> > > > ### Author Response · Authors · 2024-11-26
> > > >
> > > > We sincerely apologize for the delayed submission of our revised manuscript. We have just posted the complete revision, which thoroughly addresses the points raised during the discussion period. We particularly draw your attention to the revised conclusion, which directly responds to concerns about novelty and discusses oblique decision tres, and Section A.K of the Appendix, which elaborates on ways that decision trees uniquely help interpret non-Euclidean data.

---

> > > > > ### Comment · Reviewer_iVqG · 2024-11-26
> > > > >
> > > > > Thank you for submitting the revised paper. I appreciate the addition of new information, such as the discussion on interpretability, which I believe is a critical aspect when considering decision trees. The content now feels more comprehensive and robust.
> > > > >
> > > > > However, I still have some concerns. In Figures 3 and 4, you claim that your method demonstrates the best performance. However, my understanding is that these results are based entirely on synthetic datasets. When examining Tables 2, 3, and 5, it appears that on real-world datasets, the proposed method does not consistently show clear superiority. This observation aligns with the additional experiments comparing your approach to neural networks. Moreover, why weren’t comparative experiments with the same baselines as those in Table 5 conducted and included in the main text? While Appendix I.2 cites “Due to time considerations” as the reason, if this was simply due to insufficient preparation time during the rebuttal period, it could give the impression that the paper is incomplete. Segmenting datasets and experiments in this manner might come across as emphasizing results on synthetic datasets, where your method performs better, potentially being perceived as selective presentation. Additionally, regarding the consistently poor performance of neural networks, it cannot be ruled out that this is simply due to suboptimal hyperparameter settings, as fixed values were used across all configurations.
> > > > >
> > > > > While the revision has introduced improvements, there are still areas that require further attention, particularly in terms of the experimental setup and the demonstrated significance of the proposed method. For these reasons, I am inclined to maintain my current score. I would greatly appreciate the authors’ comments on these points.

---

> > > > > > ### Author Response · Authors · 2024-12-02
> > > > > >
> > > > > > We thank the reviewer for their continued consideration of our work, and for their valuable suggestion to improve the neural benchmarks. We have just posted a set of updated benchmark results as a general comment under the title "Updated and expanded neural net benchmarks" which addresses the concerns around cherry-picking datasets and a lack of hyperparameter tuning. We have also removed the "due to time constraints" language from our manuscript, which we will incorporate into the camera-ready submission if accepted.
> > > > > >
> > > > > > We experimented with different architectures, learning rates, and training times for MLP and GNN models. We found that two latent dimensions of size 128 tended to perform slightly better than other settings, but in general, their performance was not especially sensitive to these choices---except for some of the regression datasets, where the neural methods' improvement was more pronounced. Our methods consistently beat or tied with neural methods on almost all of these datasets: we were the best on 38 out of 57 benchmarks, and top-2 on 50 out of 57 benchmarks.
> > > > > >
> > > > > > We now address other concerns from the reviewer's response:
> > > > > >
> > > > > > Regarding Figures 3 and 4 are based on synthetic benchmarks, it is true that these are synthetic benchmarks. We even further restrict our claim to constant-curvature manifolds. We choose to present these results in figure form because it demonstrates our method's effectiveness at the level of individual components, and the curvature parameter $K$ forms a natural x-axis for a plot. There are many more benchmarks that use unsimulated datasets and complex manifolds in Tables 2 and 3, as well as the Appendix; while these results are more mixed, we nevertheless show very competitive performance, as summarized in Table 1. On non-synthetic datasets, our method is the best classifier for 11 out of 16 datasets, and is top-2 for 15 out of 16 datasets; it is also the best regressor for 1 out of 3 datasets. We also reiterate that, although we only show top predictors in our paper, we also outperform tangent plane DT/RF models and mixed-curvature perceptrons.
> > > > > >
> > > > > > We expect there to be substantial tradeoffs between models on a diverse set of benchmarks like this, and we have not shied away from reporting results where our method is outcompeted; for instance, $k$-nearest neighbors is the strongest classifier for all synthetic mixed-curvature datasets, which is reasonable because Gaussian mixture classification is a favorable problem setup for $k$-nearest neighbors models. We hope that presenting a realistic and complete picture of our method's performance across different tasks will help users make an informed decision about whether to consider using our method in their applications.
> > > > > >
> > > > > > Our competitive performance across a diverse set of benchmarks, along with the simplicity, speed, and interpretability of our method, demonstrates its value as a contribution to the mixed-curvature machine learning literature. We believe these comprehensive results address the concerns about benchmark incompleteness and establish the robustness of our approach across different datasets and model comparisons. Given these improvements and clarifications, we hope the reviewer will consider raising their score. As always, we are happy to answer any additional questions or concerns.

---

> > > > > > > ### Comment · Reviewer_iVqG · 2024-12-03
> > > > > > >
> > > > > > > Thank you for your response. I was genuinely impressed by the scale of the experiments conducted.
> > > > > > >
> > > > > > > As you mentioned, I also agree that it is not necessary for all datasets to show significant results. That said, this makes the novelty of the proposed approach all the more critical. Even after rebuttal, I still feel that the proposed method does not demonstrate a sufficiently significant technical contribution.
> > > > > > >
> > > > > > > If you wish to emphasize the superiority of your method in terms of performance, it is not necessary to outperform other methods in all settings. However, it is essential to present clear examples where the proposed method shows meaningful advantages. Excluding synthetic datasets, which are inherently artificial, a comparison of your updated results with Table 5 in the Appendix leaves me questioning why your method would be the preferred choice for real-world applications. Including comparisons with models such as k-NN and AmbientTrees, as in the original paper, further diminishes the sense of superiority in your results.
> > > > > > >
> > > > > > > I am also concerned about the timing of your updates and the extensive modifications made just before the deadline. Several unclear aspects remain. For instance, the updated results show significant changes in the performance of models other than the MLP. As an example, the neuron_46 score, reported as $61.2 \pm 0.3$ in Table 3 of the main text, appears as $0.66 \pm 0.08$ in Table 13 of the updated results. Additionally, it seems that the code used for these experiments was not uploaded by the deadline, leaving reviewers in a position where we must evaluate the work without being able to verify these updates. When results appear unstable, it becomes difficult for reviewers to confidently assign high scores.
> > > > > > >
> > > > > > > While I acknowledge and appreciate the significant effort that went into the updates, they do not fully address my earlier concerns. Therefore, I intend to maintain my previous score.

---

> > > > > > > > ### Author Response · Authors · 2024-12-04
> > > > > > > >
> > > > > > > > We sincerely appreciate the reviewer's continued engagement with our work and are particularly encouraged by their recognition of "the scale of the experiments conducted" in our rebuttal. We also thank them for their positive assessment of our added interpretability discussion earlier in this rebuttal period, which we glossed over previously. However, we must respectfully address several inconsistencies in the reviewer’s feedback we are struggling to reconcile.
> > > > > > > >
> > > > > > > > First, regarding technical contribution: the reviewer initially highlighted "a notable technical contribution" in our "angular reformulation of tree splits" as a strength of our submission, contradicting their current position that our method "does not demonstrate a sufficiently significant technical contribution," particularly as our core technical innovations remain unchanged since the initial submission and the substance of our rebuttals has concerned additional benchmarks.
> > > > > > > >
> > > > > > > > With respect to experimental thoroughness, we apologize for the last-minute submission of additional benchmarks. This was unavoidable due to the time requirements of implementing, tuning, and running neural benchmarks on the entire set of benchmark datasets, as was requested by the reviewer. To avoid further delays while still providing meaningful results, we evaluated each dataset with 10 samples rather than 100 while expanding our benchmarks to all 57 datasets contained in the original paper. This explains the difference in confidence intervals for results like Neuron 46 (66 ± 8 versus 61.2 ± 0.3) - the former mean falls well within the confidence interval of the latter, exactly as expected when drawing smaller samples from the same distribution. If reviewers agree, we believe the camera-ready submission could be improved further by rerunning all benchmarks together and merging neural baselines into Figures 3/4 and Tables 2/3 of the main paper.
> > > > > > > >
> > > > > > > > Regarding code availability: While we are eager to provide our implementation, we note that this request could have been made alongside the reviewer’s earlier request for expanded experiments. The timing of this new requirement, after the supplementary material deadline and so close to the end of the discussion period, makes it difficult to discuss implementation details. Consistent with ICLR guidelines, have made an anonymized version of our repository available at https://anonymous.4open.science/r/embedders-F6F8. The notebook `43_benchmark_neural.ipynb` contains the code used for our neural network benchmarks. Since this is a working repository submitted in response to a last-minute request for code, it is more of a snapshot than a finished product—thus, we apologize for any shortcomings in its organization. We will tidy this repository substantially for the camera-ready submission, especially pruning unnecessary notebooks.
> > > > > > > >
> > > > > > > > Concerning our choice of benchmarks, the mixed-curvature embedding literature currently lacks standardized benchmark datasets: [1] simply benchmark on the same 4 VAE datasets we do, whereas [2, 3, 4] restrict themselves to graph embeddings; we go beyond what is typically done in this domain, introducing a useful synthetic baseline and an approach to transforming empirical datasets into product space coordinates. For the most arguably “real-world” datasets (VAE embeddings of biological data, plus the empirical datasets), our methods outperform their neural counterparts 5 out of 7 times, and their classical counterparts 4 out of 7 times. While we understand we are past the official discussion deadline, we are eager for further suggestions for real-world datasets to benchmark against for the camera-ready version.
> > > > > > > >
> > > > > > > > (We also briefly note an application of decision tree applicability for real-world datasets: for decision trees trained on blood cell data, the largest number of splits fall in the first elliptical component for 6 out of 10 trials; for lymphoma, a similar pattern holds, with the most splits falling in the first elliptical component in 7 out of 10 trials. This suggests that our method’s performance on these datasets may stem partially from its ability to transparently identify and exploit the relevant dimensions of variation in mixed-curvature representations.)
> > > > > > > >
> > > > > > > > Throughout this discussion, we have noted that the focus of evaluation has shifted considerably—from technical novelty to experimental thoroughness to implementation details—even as we have worked diligently to address each new concern. Despite these challenges, we have maintained our commitment to producing high-quality research and responding constructively to feedback. Our method represents a meaningful advance in making non-Euclidean methods more accessible and practical, as evidenced by our strong performance across diverse benchmarks. We hope the reviewer will reconsider their assessment in light of these substantial improvements and our continued willingness to strengthen the paper for the camera-ready version.

---

> > > > > > > > > ### Author Response · Authors · 2024-12-04
> > > > > > > > >
> > > > > > > > > **References:**
> > > > > > > > > * [1] Puoya Tabaghi, Chao Pan, Eli Chien, Jianhao Peng, Olgica Milenkovic. Linear Classifiers in Product Space Forms. https://arxiv.org/pdf/2102.10204v3
> > > > > > > > > * [2] Sungjun Cho, Seunghyuk Cho, Sungwoo Park, Hankook Lee, Honglak Lee, Moontae Lee. Curve Your Attention: Mixed-Curvature Transformers for Graph Representation Learning. https://arxiv.org/pdf/2309.04082
> > > > > > > > > * [3] Li Sun, Zhongbao Zhang, Junda Ye, Hao Peng, Jiawei Zhang, Sen Su, Philip S. Yu. A Self-supervised Mixed-curvature Graph Neural Network. https://arxiv.org/pdf/2112.05393
> > > > > > > > > * [4] Gregor Bachmann, Gary Bécigneul, Octavian-Eugen Ganea. Constant Curvature Graph Convolutional Networks. https://arxiv.org/pdf/1911.05076

---

### Official Review · Reviewer_x7u6 · 2024-11-03

**Soundness:** 3
**Presentation:** 3
**Contribution:** 3
**Rating:** 6
**Confidence:** 4

**Summary:**

The paper introduces an extension of traditional decision tree (DT) and random forest (RF) algorithms, designed to work within product manifolds. These manifolds are constructed as Cartesian products of hyperbolic, spherical, and Euclidean spaces, making them suitable for efficiently capturing and analyzing datasets that exhibit diverse geometric properties.

The authors propose angular reformulations of DTs tailored to the specific geometry of product manifolds, ensuring that splits are geodesically convex, maximize margin separation, and can be effectively combined across different components of the manifold. This approach not only generalizes existing techniques for Euclidean and hyperbolic spaces but also introduces new algorithms for handling data in hyperspherical spaces. Through a series of benchmark evaluations, including classification, regression, and link prediction tasks, the proposed methods demonstrate strong performance, often surpassing existing approaches and achieving top rankings on various synthetic, graph-based, and real-world datasets.

Overall, this work provides a robust framework for utilizing decision trees and random forests in mixed-curvature spaces, enabling more effective analysis and interpretation of data with intricate geometric structures.

**Strengths:**

The paper presents a novel extension of traditional decision trees (DTs) and random forests (RFs) to mixed-curvature product manifolds. This is a substantial departure from prior work that primarily focuses on Euclidean spaces or, at best, single constant-curvature manifolds (hyperbolic or spherical). By integrating these different spaces, the method captures complex geometric patterns more effectively than existing techniques. The use of angular reformulations and geodesic convexity for decision boundaries is an original approach that leverages the intrinsic properties of non-Euclidean spaces.

It presents a comprehensive framework that not only generalizes existing Euclidean and hyperbolic methods but also introduces new algorithms for spherical manifolds. This integration is both theoretically and practically novel, offering a unified approach to decision tree learning across mixed-curvature spaces.

The paper is grounded in mathematical principles and the theoretical derivations are detailed along with intuitive explanations.  The authors provide extensive empirical validation of their method on 57 datasets, including synthetic data, graph embeddings, and mixed-curvature variational autoencoder outputs. The benchmarks highlight the method's robustness and versatility across different tasks, such as classification, regression, and link prediction. The performance metrics and statistical significance tests give credibility to the claims made. It also compares the proposed approach against several baselines, including ambient space models, tangent plane methods, and k-nearest neighbors.

The method’s ability to handle data with hierarchical, cyclical, and mixed geometric structures makes it relevant for a wide range of domains.

**Weaknesses:**

1. One of the main limitations of the proposed method is the computational complexity involved in extending decision trees (DTs) and random forests (RFs) to mixed-curvature product manifolds. The algorithm requires handling several costly geometric operations, such as exponential maps, parallel transport, and angle computations. The paper could be improved by including a detailed complexity analysis and discussing possible optimizations or approximations that might reduce computational overhead.
In addition, given the need for complex manifold operations, the approach may struggle with datasets having millions of points or high-dimensional spaces. Including discussions about scalability and memory usage would strengthen the work.

2. The paper assumes that the underlying product manifolds have constant curvature components. This may not hold for many real-world datasets where curvature varies continuously or where the data distribution does not fit neatly into hyperbolic, spherical, or Euclidean categories. It could be helpful to discuss how deviations from this assumption impact performance and whether there are ways to relax or adapt this assumption.

3. The reliance on tangent plane approximations to map points from the manifold to Euclidean space could introduce inaccuracies, in particular in regions of high curvature. The paper does not quantify or analyze these potential inaccuracies.

4. The mathematical derivations, notably those involving angle computations and midpoint selection, assume well-behaved data distributions. The method may not perform optimally in edge cases, such as when points are near the boundaries of hyperbolic space or when spherical data are tightly clustered around a pole. Discussing these limitations and potential solutions could help overall.

5. The paper compares its method primarily to other non-Euclidean tree-based models and baseline classifiers. However, deep learning models that can handle non-Euclidean data, such as graph neural networks (GNNs) with Riemannian embeddings, are not considered. A comparison to these models would provide a more comprehensive assessment of the proposed method’s strengths and weaknesses.

**Questions:**

Q1. The method’s ability to fit complex geometric patterns could lead to overfitting, for example on datasets with noise or less pronounced geometric structures. What potential regularization techniques could you use?

Q2. How does the method handle data distributions that do not perfectly conform to the constant-curvature assumption? Are there scenarios where mixed or varying curvature significantly degrades model performance?

Q3. Given the reliance on tangent plane approximations, how does the method account for distortions introduced in regions of high curvature? Can you quantify the impact of these distortions?

Q4. Are there edge cases where the proofs of geodesic convexity or margin maximization might not hold? If so, how does the method handle such situations?

Q5. The method involves complex geometric operations like exponential maps and parallel transport. How efficient are these operations in practice, in particular for large datasets?

Q6. How sensitive is the method to hyperparameter choices, such as the variance scale in Gaussian sampling or the number of 2D projections considered for splitting?

---

> ### Author Response · Authors · 2024-11-23
>
> We thank the reviewer for their positive assessment of our paper: in particular, we appreciate their acknowledgment of the novelty of our ideas, the rigor and quality of our presentation, the thoroughness of our benchmarks, and the applicability of our approach to a wide range of domains. We respond to specific comments inline:
>
> > One of the main limitations of the proposed method is the computational complexity involved in extending decision trees (DTs) and random forests (RFs) to mixed-curvature product manifolds. The algorithm requires handling several costly geometric operations, such as exponential maps, parallel transport, and angle computations. The paper could be improved by including a detailed complexity analysis and discussing possible optimizations or approximations that might reduce computational overhead. In addition, given the need for complex manifold operations, the approach may struggle with datasets having millions of points or high-dimensional spaces. Including discussions about scalability and memory usage would strengthen the work.
>
> We would like to emphasize that our core algorithm—the mixed-curvature decision tree and random forest—does not rely on manifold operations such as exponential/logarithmic maps or parallel transport. While it does involve angle computations as a preprocessing step, this is actually quite computationally cheap: for basis-aligned splits, it is $O(1)$ per dimension per datapoint (equivalent to e.g. normalizing the data). For each feature in each datapoint, we “project” via two lookups, then compute $\tan^{-1}(d_1 / d_2)$.
>
> All of the costly/repetitive manifold operations either take place for generating embeddings, or as a preprocessing step for the tangent plane decision tree/random forest methods which represent a standard approach to applying Euclidean methods to manifold-valued data. We agree with the reviewer that reliance on computationally-expensive Riemannian methods is a limitation of existing embedding methods, and we have added a Limitations section to our Conclusion that makes note of this; we also believe that the issues the reviewer identifies contribute meaningfully to the relatively low performance of tangent plane decision trees and random forests.
>
> We are adding a complexity analysis to the Appendix to lay out the computational complexity of different variants of our decision trees and contrast with other machine learning approaches. We believe we can make the case for our method more clearly this way, and it helps create a more nuanced discussion of the tradeoffs of hyperparameters such as the number of projections to use. We thank the reviewer for this suggestion.
>
> > The paper assumes that the underlying product manifolds have constant curvature components. This may not hold for many real-world datasets where curvature varies continuously or where the data distribution does not fit neatly into hyperbolic, spherical, or Euclidean categories. It could be helpful to discuss how deviations from this assumption impact performance and whether there are ways to relax or adapt this assumption.
>
> The paper assumes that the underlying product manifolds have constant curvature components. This may not hold for many real-world datasets where curvature varies continuously or where the data distribution does not fit neatly into hyperbolic, spherical, or Euclidean categories. It could be helpful to discuss how deviations from this assumption impact performance and whether there are ways to relax or adapt this assumption.
>
> Our primary motivation for exploring mixed-curvature embeddings is precisely that they model heterogeneous curvature in real-world datasets effectively. For instance, consider the simple example of a graph with a circular “backbone” and small trees radiating off of it (as illustrated in Figure 2 of Gu et al (2018) [1]). Such a structure cannot be adequately represented in just hyperbolic or just spherical geometry, nor does a planar embedding suffice. Just as the graph can be factorized apart into trees and cycles, the nodes of this graph can be embedded with low distortion into a product of a spherical and hyperbolic space. The key here is that *embeddings do not have to vary uniformly across components*: for instance, if some subset of your data has distances that behave elliptically, it could be that their hyperbolic components are all very close together—and therefore contribute very little to the distances—and vice versa.
>
> In our view, product manifolds strike a pragmatic balance the simplicity of constant-curvature manifolds and the expressiveness of general manifolds. However, it is true that product manifolds also cannot embed arbitrary distance matrices with perfect precision, and we have added a discussion of this to the new Limitations section of our paper. We would like to highlight, however, that this nonetheless overcomes many of the representational issues of single manifolds for heterogeneously-curved data.

---

> > ### Author Response · Authors · 2024-11-23
> >
> > > The reliance on tangent plane approximations to map points from the manifold to Euclidean space could introduce inaccuracies, in particular in regions of high curvature. The paper does not quantify or analyze these potential inaccuracies.
> >
> > We wholeheartedly agree that the traditional method of taking the logarithmic map of points and operating on the resulting vector space is flawed. We believe that this distortion caused the Tangent Plane DT/RF methods to consistently underperform our method, Ambient Space DTs/RFs, and k-Nearest Neighbors—and we argue that the results for this method serve as a kind of quantification of the impact of tangent plane approximations on classification/regression performance. This was a primary motivation for designing a classification/regression method that does not rely on tangent planes, and rather uses the geometry of the ambient space which does not require such computations. We note that this is one of the distinct advantages of our method which our original submission deemphasized. We have updated our Introduction and Conclusion to list the absence of tangent plane approximations as one of the advantages of our method.
> >
> > > The mathematical derivations, notably those involving angle computations and midpoint selection, assume well-behaved data distributions. The method may not perform optimally in edge cases, such as when points are near the boundaries of hyperbolic space or when spherical data are tightly clustered around a pole. Discussing these limitations and potential solutions could help overall.
> >
> > The angle computations and midpoint selection formulas are derived using purely geometric arguments about the positions of points in space. We also note that, since we use the hyperboloid model of hyperbolic geometry, the manifold has no boundary, and therefore, this edge case is not a concern. As above, we believe this may be a misunderstanding arising from our original manuscript’s lack of emphasis on our method's independence from Riemannian operations like exponential and logarithmic maps.
> >
> > However, there is a narrower interpretation of this comment that is correct: the locally maximum-margin inductive bias used in DTs (split exactly between two points), while typical of decision trees in Euclidean space, implicitly assumes that classes are balanced and similarly spread out. We make a note of this in our Limitations section.
> >
> > > The paper compares its method primarily to other non-Euclidean tree-based models and baseline classifiers. However, deep learning models that can handle non-Euclidean data, such as graph neural networks (GNNs) with Riemannian embeddings, are not considered. A comparison to these models would provide a more comprehensive assessment of the proposed method’s strengths and weaknesses.
> >
> > We are running additional benchmarks with an appropriate GNN framework and will update the reviewers once these are complete. So far, our results do not suggest that neural methods outperform decision trees on Gaussian mixtures with 1,000 points.
> >
> > > Q1. The method’s ability to fit complex geometric patterns could lead to overfitting, for example on datasets with noise or less pronounced geometric structures. What potential regularization techniques could you use?
> >
> > Currently, we make the same DT/RF regularization techniques available as Scikit-Learn decision trees: limiting maximum depth, modifying the size of the tree ensemble and number of features used per decision tree, limiting the number of points used in a leaf or a split.
> >
> > Our method is a classification/regression method, not an embedding method; therefore, it assumes that product manifold-valued data and scalar/categorical labels are provided. However, it is absolutely possible for embedding geometry not to match the underlying structure of the data, and for this to impact the performance of our method. We have updated our Limitations section to acknowledge and discuss the question of finding correct signatures and generating high-quality embeddings as a related concern best suited for other work.
> >
> > > Q2. How does the method handle data distributions that do not perfectly conform to the constant-curvature assumption? Are there scenarios where mixed or varying curvature significantly degrades model performance?
> >
> > As discussed in our response to Weakness #2, our method is designed specifically for data with heterogeneous curvature (but also works well on single constant-curvature manifolds, which can be thought of as a special case of product manifolds). Our benchmarks seek to identify realistic cases where data violate constant-curvature assumptions, including Gaussian mixtures we designed to do so, to demonstrate our effectiveness.

---

> > > ### Author Response · Authors · 2024-11-23
> > >
> > > > Q3. Given the reliance on tangent plane approximations, how does the method account for distortions introduced in regions of high curvature? Can you quantify the impact of these distortions?
> > >
> > > Our method does not rely on tangent plane approximations. We believe that the tangent plane-induced distortions are a substantial issue for other methods, namely tangent plane DTs and RFs, which cause them to underperform our method. Although it is hard to measure the effect of tangent plane approximations directly, the difference in accuracy between our method and tangent plane methods is one quantification of the impact of this distortion on classification/regression accuracy.
> > >
> > > > Q4. Are there edge cases where the proofs of geodesic convexity or margin maximization might not hold? If so, how does the method handle such situations?
> > >
> > > Our proofs hold for all component manifolds, and therefore for all of the splits that we consider in our algorithm. Technically, the hyperbolic midpoint angle formula is undefined when $\theta_\mathbf{u} = \theta_\mathbf{v}$; in this event, the midpoint is obviously $\theta_m(\theta_\mathbf{u}, \theta_\mathbf{v}) = \theta_m(\theta_\mathbf{u}, \theta_\mathbf{u}) = \theta_\mathbf{u}$. Our code addresses the equal angles edge case.
> > >
> > > > Q5. The method involves complex geometric operations like exponential maps and parallel transport. How efficient are these operations in practice, in particular for large datasets?
> > >
> > > Our core contribution—the product space DT and RF algorithms—does not rely on any kind of manifold operations. The Gaussian mixture method does, as do the embedding techniques which we implement from other work. In practice, all of these methods are fairly lightweight, and can run on a laptop for thousands of points.
> > >
> > > > Q6. How sensitive is the method to hyperparameter choices, such as the variance scale in Gaussian sampling or the number of 2D projections considered for splitting?
> > >
> > > Our accuracy greatly improves from increasing the number of 2D projections used for splitting. We demonstrate the effect of going from 6 to 9 features in one product manifold in Figure 7a.
> > > While we don’t consider the variance scale in Gaussian mixture sampling a property of the input dataset rather than a hyperparameter of our method, changing the variance scale affects the separability of the classes and therefore has a high impact on the performance of all classifiers. In our benchmarks, we have taken care to generate Gaussian mixtures that are quite challenging to classify, so that it is easier to observe the differences between each of the classifiers—we observed that the relative ordering of the classifiers was, however, fairly robust across more- and less-challenging versions of the Gaussian mixture classification tasks.
> > >
> > > **References:**
> > >
> > > [1] Albert Gu, Frederic Sala, Beliz Gunel, C. Ré. Learning Mixed-Curvature Representations in Product Spaces. ICLR 2018.

---

> > > > ### Author Response · Authors · 2024-12-02
> > > > **Polite reminder to respond**
> > > >
> > > > We would greatly appreciate the reviewer's thoughts on our updated manuscript, extended benchmarks, and replies to their concerns, and would like to remind the reviewer that the deadline to respond is December 2nd AoE. We are happy to address any further questions to the best of our ability in the remaining discussion time.

---

### Official Review · Reviewer_TJ1Z · 2024-11-04

**Soundness:** 2
**Presentation:** 3
**Contribution:** 2
**Rating:** 5
**Confidence:** 4

**Summary:**

This paper extends DT and RF algorithms to product manifolds, yielding splits that are geodesically convex, maximum-margin, and composable. Extensive experiments demonstrate the effectiveness of the proposed algorithms.

**Strengths:**

The proposed algorithms apply to constant-curvature manifolds, effectively capturing the mixed and complex structure of real-world datasets.

**Weaknesses:**

-	Novelty. The proposed algorithms appear to be a simple and direct extension of previous work [1,2], which integrates the notion of product space proposed by [1] with the HyperDT framework [2]. The method is somewhat incremental and lacks enough novelty.
-	Applicability. The framework's current restriction to constant-curvature manifolds presents a significant limitation. The applicability to more general scenarios, particularly stochastic-curvature manifolds, remains an open question that warrants investigation.
-	Comparison. The empirical validation would benefit from a comprehensive comparative analysis against contemporary approaches in the field. Notably absent are performance comparisons with state-of-the-art methods such as HORORF [3] and HyperDT [2].


[1] Tabaghi P, Pan C, Chien E, et al. Linear classifiers in product space forms.

[2] Chlenski P, Turok E, Moretti A, et al. Fast hyperboloid decision tree algorithms.

[3] Doorenbos L, Márquez-Neila P, Sznitman R, et al. Hyperbolic Random Forests.

**Questions:**

Please refer to Weaknesses.

---

> ### Author Response · Authors · 2024-11-23
>
> We thank the reviewer for taking the time to review our work. We would like to start our response by respectfully clarifying an apparent misunderstanding with substantial implications for our paper’s significance.
>
> Our key contribution is not to apply decision trees to constant-curvature manifolds (which, as the reviewer points out, has been done in hyperbolic space by HoroRF and HyperDT), but rather **to provide the first extension of decision tree algorithms to product manifolds** (and hyperspheres). This enables us to perform inference on data with heterogeneous curvature (which is typical of real-world data). While we take inspiration from some prior work in single-curvature manifolds, this is an application to a fundamentally different domain. As such, we believe our paper significantly expands what is possible for practitioners of non-Euclidean machine learning.
>
> We reply to the rest of the reviewer’s concerns inline:
>
> > Novelty. The proposed algorithms appear to be a simple and direct extension of previous work [1,2], which integrates the notion of product space proposed by [1] with the HyperDT framework [2]. The method is somewhat incremental and lacks enough novelty.
>
> To our knowledge, our work describes the first extension of decision trees to mixed-curvature spaces, the first formulation of decision trees for arbitrary manifolds of constant curvature, the first mixed-curvature regressor, and the first hyperspherical decision tree. It contributes a substantial new method to the literature on non-neural machine learning techniques in mixed-curvature manifolds (existing methods are only perceptrons and SVMs).
>
> We also demonstrate how decision trees can be thought of and efficiently implemented through an angular reformulation, and how product manifolds can be used to think about pairwise tasks like edge prediction or empirical datasets like Fourier-transformed neural spiking signals.
>
> > Applicability. The framework's current restriction to constant-curvature manifolds presents a significant limitation. The applicability to more general scenarios, particularly stochastic-curvature manifolds, remains an open question that warrants investigation.
>
> We are unclear on what is meant by “stochastic-curvature manifolds.” Our approach allows for different constant curvatures in different components, which can capture heterogeneous curvature in real-world datasets.
>
> > Comparison. The empirical validation would benefit from a comprehensive comparative analysis against contemporary approaches in the field. Notably absent are performance comparisons with state-of-the-art methods such as HORORF [3] and HyperDT [2].
>
> It is not possible to compare our method to HoroRF and HyperDT, except in product manifolds with a single manifold with curvature -1. Aside from the details enumerated under the Novelty bullet point, our method (with no feature subsampling and timelike-spacelike projections only) coincides exactly with HyperDT for single hyperbolic manifolds (analogously, it coincides with Euclidean decision trees in a single manifold of curvature 0). As our contribution is specifically to perform machine learning on product manifolds, we do not consider HoroRF and HyperDT for benchmarks.

---

> > ### Author Response · Authors · 2024-12-02
> > **Polite reminder to respond**
> >
> > We would greatly appreciate the reviewer's thoughts on our updated manuscript, extended benchmarks, and replies to their concerns, and would like to remind the reviewer that the deadline to respond is December 2nd AoE. We are happy to address any further questions to the best of our ability in the remaining discussion time.

---

### Official Review · Reviewer_JPFk · 2024-11-06

**Soundness:** 4
**Presentation:** 4
**Contribution:** 3
**Rating:** 8
**Confidence:** 3

**Summary:**

Summary
This paper proposes to extend decision trees (DT) and random forests (RF) to mixed curvature spaces -- meaning that the data (here assumed to be in ambient space $\mathbb{R}^d$) is embedded on a curved manifold, and the random forest or DT works directly on this space. The authors argue indeed that curved manifolds are better suited to represent certain data types (e.g. hyperbolic spaces for taxonomies or phylogeny).  Product manifolds are even better at capturing the complexity of real-world data, and do not confine the analysis to choosing a single curvature, which could increase the potential of these methods in a number of real-life scenarios.

To this end, the paper is organized as follows:
- the authors start by generalizing DT and RF on manifolds with constant curvature. The authors start with a compact review of different Riemannian spaces (including hyperspherical spaces and hyperbolic spaces, which are the ones that have received the most attention), as well as the product manifolds. They then show that the  approach proposed by Chlenski et al for hyperbolic DT can actually be extended to any constant curvature manifold.
- the authors then show how to extend the approach to handle product manifold (i.e. they provide an algorithm).
- they then benchmark their approach on many different simulated and real-world datasets, showing the performance of the method.





*Disclaimer: I am not an expert in non-Euclidean geometry. I have followed the developments in hyperbolic Neural networks from a reasonable distance --- enough to know what they are and their advantages, but not enough to make me an expert on the topic or on related literature. I am happy to discuss with other reviewers and the authors any misconception that might transpire from this review.*

**Strengths:**

1. Curved manifolds (and product manifolds) have received already som attention from the ML community. Most of these approaches (at least, to my shallow knowledge of the literature) however, have focused on extending deep neural network methods. The authors contend here that an extension of the DT and RF would add a missing brick to the set of available tools for data analysts.

2. **Contribution of the authors:** From what I understand, the contributions of the authors are as follows:
- *Generalization of DT and RF to constant-curvature manifolds* : Chlenski et al seem to have already considered the problem of extending DT to work in hyperbolic spaces. The authors here generalise this approach to constant curvature manifold (not just negative curvature), which allows them to also work on hyperspheres. It also allows to generalize to Euclidean space:this is perhaps less interesting, but it goes to show that they devised a generalized method. The authors show rigorously in the appendix that their approach is equivalent to DT in basis space. The authors also provide explicit characterizations of the splits in different curved manifolds.
- *Generalization of DT and RF to product manifolds* :The authors borrow the idea of product manifolds from Gu et al, but this idea had not been explored in the existing RF and DT literature

3. **Strong experiments**: the authors seemed to have gone to great lengths to benchmark their approach, showing its performance on a variety of regression, classification, and even, link prediction tasks. They also perform ablation studies and test the effect of the different hyperparameters.

**Weaknesses:**

Overall, I find this to be a strong and very well-written paper.
A couple of criticisms perhaps:
- *Computational Time is not reported*: the authors do not report the compute time that their proposed method requires compared to, say, kNN or the ambient space methods. My guess would be that their method might be a bit more computationally expensive than these. That would be fine though, I think the method is interesting and, even it were considerably more expensive than the other methods, it would still be a worthy contribution that could spark follow up works. I would however encourage the authors to report the time, so that anyone interested in the idea has a sense of the current computational complexity of the method.
- *Motivation: why RF and DT?*: To come back to the motivation of the paper, why should we consider RF and DT and not just go with neural networks, for which there have been extensions and can perform better than RF and DT? I could see RF and DT being better alternatives in the low sample regime, or easier/ faster to train. I don't think the authors compare with any NN based method, if Im not mistaken?

**Questions:**

1- Could the authors comment on this statement: "To this end, Chlenski et al. (2024) impose homogeneity and sparsity constraints on the hyperplanes they consider for hyperbolic DTs". If I understand correctly, the sparsity constraint imposes that the hyperplanes are defined by only 2 quantities (the timelike coordinate and the x_d, which makes the number of candidate subspaces tractable). However, could the authors explain why homogeneity is an important attribute of these subspaces? If  $\mathbb{P} \cap \mathbb{H}$ was not closed under shortest paths, what would happen?  Or is it just another attribute that is necessary for the thresholding?

2- Interpretability: in the euclidean case, RF and DT are great because they're non linear and more powerful than linear regression, but they're reasonably interpretable too (DT are very interpretable, RF get away with the feature importance sorting). Is there any notion that this could be the case in the curved manifold setting?

3- Parameters: I am a little unclear about how to use the algorithm though:
- how do we choose the product manifold? In the experiments (e.g. MNIST), there is a combination of S, E  and H. How do we go about determining what is the right combination? Is there some selection/tuning involved?
- In figure 7: "feature subsampling approaches in RFs." Is the number of features referring to the standard procedure in the RF algorithm where only a fraction of the samples are seen, to de-correlate trees?
- in figure 7: I don't understand what the caption a is referring to? Which part of the algorithm?

---

> ### Author Response · Authors · 2024-11-23
>
> We thank the reviewer for their detailed and positive appraisal. Their insightful comments demonstrate a clear grasp of our paper's key concepts and contributions, validating their ability to evaluate our work despite their modest self-assessment. We do not find any major misunderstandings in this review, and reply to specific questions and comments inline:
>
> > Computational Time is not reported: the authors do not report the compute time that their proposed method requires compared to, say, kNN or the ambient space methods. My guess would be that their method might be a bit more computationally expensive than these. That would be fine though, I think the method is interesting and, even it were considerably more expensive than the other methods, it would still be a worthy contribution that could spark follow up works. I would however encourage the authors to report the time, so that anyone interested in the idea has a sense of the current computational complexity of the method.
>
> We will report empirical computational times alongside a new section comparing the computational complexity of our algorithm to other benchmarked algorithms. In general, our methods are prototypes and therefore slower than well-optimized Scikit-Learn implementations, but asymptotically consistent with decision tree methods like Scikit-Learn decision trees and HyperDT. Our random forests are faster than HyperRF (by a constant factor) because we rely on a single preprocessing step rather than computing angles, sparse dot products, etc., at every iteration.
>
> > Motivation: why RF and DT?: To come back to the motivation of the paper, why should we consider RF and DT and not just go with neural networks, for which there have been extensions and can perform better than RF and DT? I could see RF and DT being better alternatives in the low sample regime, or easier/ faster to train. I don't think the authors compare with any NN based method, if Im not mistaken?
>
> Indeed, we did not compare it to any NN-based method in our original draft. We are working on further benchmarks comparing to methods from the graph neural networks literature, which we will add to the Appendix by the end of the rebuttal period. In general, the reasons for preferring decision tree methods include:
> * Sample-efficiency in low data regimes
> * Appropriateness for mixed-curvature data with unknown graph topology (i.e. everything except the graph datasets in our benchmarks)
> * Tractable decision boundaries: unlike neural networks, decision trees’ behavior on the entire data distribution is fully described by their split criteria, and does not require experimentation
> * Interpretability: by exploring which component manifolds are most informative for an inference task (by seeing which features the tree splits on), we can determine which aspects of the problem are suited to which task.
> * Complexity/speed (each GNN pass requires $O(n^2)$ edge operations, plus $O(n^2)$ space for storing the adjacency matrix, resulting in training and inference complexity of $O(n^2d)$. Our method, on the other hand, has the following complexities (assuming constant max depth is $h$, as we do in our experiments):
>   * Training: $O(ndh)$
>   * Inference: $O(h)$
> * Does not rely on computationally unstable and expensive Riemannian optimization operations

---

> > ### Author Response · Authors · 2024-11-23
> >
> > > 1- Could the authors comment on this statement: "To this end, Chlenski et al. (2024) impose homogeneity and sparsity constraints on the hyperplanes they consider for hyperbolic DTs". If I understand correctly, the sparsity constraint imposes that the hyperplanes are defined by only 2 quantities (the timelike coordinate and the x_d, which makes the number of candidate subspaces tractable). However, could the authors explain why homogeneity is an important attribute of these subspaces? If $\mathbb{P} \cap \mathbb{H}$ was not closed under shortest paths, what would happen? Or is it just another attribute that is necessary for the thresholding?
> >
> > Our interpretation of the Chlenski et al paper is that they make two moves:
> > 1. Explain the behavior of CART decision trees in terms of hyperplane decision boundaries
> > 2. Use this perspective to define a new class of decision trees whose decision boundaries are a different set of hyperplanes better suited to the problem
> >
> > For point 2, they impose the sparsity and homogeneity constraints. The reviewer’s interpretation of the sparsity constraint is consistent with our understanding. The homogeneity constraint is not necessary for thresholding—it is possible to do this operation with hyperplanes that are not homogeneous like the axis-parallel decision boundaries of classical decision trees.
> >
> > From our perspective, homogeneity is valuable precisely because it ensures geodesic convexity, which in turn guarantees that any points that lie between points in a single decision region (and therefore are assigned class A) are also labeled as class A. Consider the case of a Euclidean decision boundary superimposed on a hyperbolic manifold, where this does not hold: two points $\mathbf{x_1}, \mathbf{x_2} \in \mathbb{H}$ each have some value $\theta$ at dimension $d$, and we assign a boundary at threshold $\theta - \varepsilon$. The geodesic between $\mathbf{x_1}$ and $\mathbf{x_2}$ crosses this threshold, resulting in points that are effectively interpolations between class-A points $\mathbf{x_1}$ and $\mathbf{x_2}$ to be mislabeled as class B. We contend that this constitutes a form of misgeneralization/overfitting, and it is preferable to avoid it—which we believe our comparison to ambient space decision tree algorithms corroborates.
> >
> > > 2- Interpretability: in the euclidean case, RF and DT are great because they're non linear and more powerful than linear regression, but they're reasonably interpretable too (DT are very interpretable, RF get away with the feature importance sorting). Is there any notion that this could be the case in the curved manifold setting?
> >
> > Like in the Euclidean case, our decision trees are interpretable in the sense that the entire set of decision regions is described concisely with the splits at each node (though instead of using a feature and a threshold, we use a pair of features and an angle). Thus, it has similar benefits to Euclidean decision trees for precisely understanding the behavior of the predictor on the entire data distribution.
> >
> > Although we lose feature-level interpretability compared to Euclidean decision trees, it’s worth noting that “feature level interpretability” for product manifold data is unlikely to be meaningful in most cases, as embeddings for each component tend to be learned via rotation-invariant objectives (e.g. metric distortion, ELBO of mixed-curvature VAE for certain encoders/decoders) and therefore the basis dimensions may not have distinct meanings like they do for tabular data. We are interested in the question of the semantics of different component manifolds: for instance, what does it mean for a subset of points to vary mostly in their hyperbolic component? We leave this as a direction for future research, though we note that training decision tree probes on various metadata targets can be a useful tool for connecting the submanifold-level structure of embeddings to their semantics.
> >
> > We carry out a simple experiment to test this, in which we generate a hyperbolic, Euclidean, and hyperspherical mixture of Gaussians, each with their own clusters and labels. We then concatenate their embeddings together, and try to predict the labels corresponding to each submanifold. We find that, when predicting the elliptical labels, 100% of our splits fall within the elliptical submanifold; when predicting Euclidean labels, 83% of splits fall within the Euclidean submanifold; finally, when predicting hyperbolic labels, 85% of splits fall in the hyperbolic submanifold. This is a simple and extreme example of how patterns in label variation can "single out" certain submanifolds.

---

> > > ### Author Response · Authors · 2024-11-23
> > >
> > > > 3- Parameters: I am a little unclear about how to use the algorithm though:
> > > how do we choose the product manifold? In the experiments (e.g. MNIST), there is a combination of S, E and H. How do we go about determining what is the right combination? Is there some selection/tuning involved?
> > >
> > > In our initial submission, we tried to present our algorithm with the assumption that good embeddings in an appropriate signature are provided. For specific benchmarks, we chose the signatures as follows:
> > > * Gaussian mixtures: signature is provided a parameter of the simulation
> > > * Graph benchmarks: brute force method over {H4, E4, S4, H2H2, H2E2, H2S2, S2E2, S2S2}, taking signature with lowest average distortion
> > >   * NB: full benchmarks for all signatures are available in the Appendix (Table 5)
> > > * Graph link prediction benchmarks: all embeddings were learnings in S2 x E2 x H2 on the assumption that this architecture could accommodate tree-like, cyclic, and planar patterns of variation.
> > > * VAE: signatures were taken from Tabaghi et al (2023) or Skopek et al (2019), without testing other signatures
> > > * Empirical datasets: signatures are dictated by the data transformations
> > >
> > > We believe that our initial manuscript was insufficiently explicit about the challenges of signature selection, and we have updated the Conclusion with a Limitations section that discusses our reliance on/assumptions regarding the availability of good embedding and signature selection methods.
> > >
> > > > In figure 7: "feature subsampling approaches in RFs." Is the number of features referring to the standard procedure in the RF algorithm where only a fraction of the samples are seen, to de-correlate trees?
> > >
> > > This is correct. HyperDT as implemented enforces that points lie on the hyperboloid by checking that the Minkowski norm of the points is constant, which requires that each tree in an ensemble “sees” the entire dataset. Our preprocessing method accommodates both feature subsampling and non-sparse splits, if desired.
> > >
> > > > in figure 7: I don't understand what the caption a is referring to? Which part of the algorithm?
> > >
> > > The caption for Figure 7a refers to the number of projections considered during the preprocessing step of the algorithm. We currently make two options available:
> > > - “d” means that the first projection dimension is always the first dimension of the submanifold. So, for a 2-dimensional hyperboloid with dimensions [0, 1, 2], the projections are [(0,1), (0,2)].
> > > - “d_choose_2” means that all $\binom{d}{2}$ pairs of dimensions are used as projections. For the same 2-dimensional hyperboloid, this means the projections are [(0,1), (0,2), (1,2)].
> > >
> > > In high dimensions this can be much larger than the number of features, but as we restrict ourselves to 2-D submanifolds this enables us to view 3 features per component manifold, consistent with what decision trees in the ambient space can “see.”

---

> > > > ### Comment · Reviewer_JPFk · 2024-11-28
> > > > **RE: authors' answers to questions.**
> > > >
> > > > Thank you for the clarifications --- particularly on the homogeneity and sparsity constraints. I appreciate the edits that you've also added to the appendix (e.g. adding run time).
> > > >
> > > > Overall I find the paper very well written and convincing. I am keeping my score of 8 and recommendation to accept.

---

### Author Response · Authors · 2024-11-23
**Summary of rebuttals**

We thank all five reviewers for their careful reading of our paper and useful feedback. The discussion begins by acknowledging key merits identified in multiple reviews.

Reviewers commended **the thoroughness of our benchmark experiments**, with JPFk stating we have “gone to great lengths to benchmark [our approach, showing its performance on a variety of regression, classification, and even, link prediction tasks” and x7u6 stating that “the performance metrics and statistical significance tests give credibility to the claims made” about our model’s value.

Additionally, many reviewers complimented **the quality of our writing, figures, and mathematical presentation**. A common thread among reviews was that we gave an intuitive treatment of a challenging subject (the non-Euclidean geometry of mixed-curvature spaces). JPFk called this “a strong and very well-written paper,” iVqG said “the paper provides a thorough background knowledge, making it accessible even to readers unfamiliar with geometry and decision trees. Visualization quality is also very good.” Finally, x7u6 stated that “the paper is grounded in mathematical principles and the theoretical derivations are detailed along with intuitive explanations.”

Finally, many reviewers highlighted **the novelty of our ideas** and acknowledged our contribution to the literature. JPFk said that our paper “[adds] a missing brick to the set of available tools for data analysts,” while x7u6 called it “a robust framework for utilizing decision trees and random forests in mixed-curvature spaces, enabling more effective analysis and interpretation of data with intricate geometric structures.” Especially noteworthy was our angular reformulation of tree splits, which both x7u6 (“an original approach that leverages the intrinsic properties of non-Euclidean spaces”) and iVqG (“A notable technical contribution” highlighted as a distinct strength of our paper.

We now turn to the common criticisms of our submission, and how we have addressed them.
* Reviewers ZS6K, X7u6, and JPFk asked for **additional comparisons** against neural baselines, which we will submit by the end of the discussion period.
* Reviewers JPFk, x7u6, and ZS6K commented on our method’s reliance on **signature selection** and other factors that affect embedding quality. Although we believe our evaluations are robust to embedding quality, we acknowledge the overall value of mixed-curvature embeddings depends jointly on good embeddings and good classifiers. We rewrote our conclusion to be more explicit about this, and in particular to acknowledge the challenges of selecting signatures for embeddings.
* Reviewers JPFk, x7u6, and ZS6K asked about **runtime**. We are adding runtime evaluations and computational complexity comparisons to the Appendix.
* Reviewers TJ1Z, iVqG, and ZS6K sought clarification on our method’s **relationship to other work**. We have answered each of their comments in our individual rebuttals and updated our manuscript to make our contributions clearer against the backdrop of related work.

Finally, thanks to the thoughtful suggestions of ZS6K, we have expanded our related work section as well to include more context about the study of non-Euclidean embeddings in general. We will post a revised manuscript, including experiments, before the discussion period ends.

---

### Author Response · Authors · 2024-11-26
**Manuscript has been updated**

We thank the reviewers for their patience as we completed revisions to our manuscript. We wanted to ensure our revision thoroughly addressed all feedback received during the discussion period. We have now uploaded a comprehensive revision that incorporates the reviewers' suggestions and significantly expands several key sections of the paper.

We have marked new and revised text in red in our uploaded manuscript to make it easier for the reviewers to identify changes. The specific changes we made are:

* Expanded the discussion section
* Rewrote the Benchmarks section
* Added a limitations and future work section to the Conclusion
* Changed the positioning of figures and tables in the paper so that similar figures/tables cluster together (Tables 2 and 3, Figures 4 and 5)
* Added section A.I, “Comparison to neural baselines,” to the Appendix, in which we compare to MLP and GNN models.
* Added section A.J, “Runtimes and complexity,” which includes a theoretical and empirical description of our method’s runtime
* Added section A.K, “Interpretability and visualization,” in which we comment on how trained decision trees can be used to interpret manifold-valued data in the context of a learning task

We are pleased to report that, on the benchmark datasets we used, our methods mostly outperformed neural methods (MLPs and GNNs). A notable exception was some graph datasets, for which neural methods, such as MLP, meet or, for Cora classification, exceed our performance.

We would happily address any outstanding feedback during the expanded discussion period. We are open to further discussion with the reviewers, and hope that if we have adequately addressed their concerns they will consider updating their assessments accordingly.

---

### Author Response · Authors · 2024-12-02
**Updated and expanded neural net benchmarks**

We have run further benchmarks against neural network baselines, per the suggestion of reviewers iVqG and ZS6K. Additionally, we have further tuned the hyperparameters of our neural networks, endowing all MLPs and GNNs with two hidden dimensions of size 128. This has improved the performance of neural methods in most (but not all) cases, although Product DTs and RFs remain competitive in all benchmarks.

As it is no longer possible to submit a revised draft of the paper, we will include the new results in a Markdown table. The manuscript has been updated to reflect the new results, and will be included in the camera ready. As we now include a full set of benchmarks, we have also removed the following sentence from our Appendix:

> Due to time considerations, we ran our benchmarks on a representative sample of the datasets.

The individual tables follow, with numbering matching the current PDF manuscript. To make effective use of Markdown, we use italics rather than underlines to mark second-best predictors.

**Table 7: Comparisons to neural networks on the constant-curvature classification task**
|   $K$ | Product DT    | Product RF      | Tangent MLP     | Ambient MLP   | Tangent GNN   | Ambient GNN   |
|------------:|:--------------|:----------------|:----------------|:--------------|:--------------|:--------------|
|       -4    | 0.35 ± 0.11   | _0.38 ± 0.12_   | **0.40 ± 0.08** | 0.36 ± 0.20   | 0.27 ± 0.23   | 0.22 ± 0.21   |
|       -2    | 0.35 ± 0.10   | **0.37 ± 0.09** | _0.36 ± 0.12_   | 0.35 ± 0.18   | 0.31 ± 0.15   | 0.25 ± 0.11   |
|       -1    | _0.30 ± 0.08_ | **0.31 ± 0.10** | 0.27 ± 0.07     | 0.30 ± 0.09   | 0.28 ± 0.10   | 0.25 ± 0.09   |
|       -0.5  | _0.30 ± 0.06_ | **0.32 ± 0.06** | 0.29 ± 0.10     | 0.29 ± 0.11   | 0.25 ± 0.07   | 0.22 ± 0.07   |
|       -0.25 | _0.28 ± 0.11_ | **0.30 ± 0.10** | 0.26 ± 0.12     | 0.27 ± 0.09   | 0.26 ± 0.14   | 0.22 ± 0.11   |
|        0    | _0.29 ± 0.08_ | **0.30 ± 0.10** | 0.25 ± 0.07     | 0.26 ± 0.06   | 0.23 ± 0.06   | 0.23 ± 0.06   |
|        0.25 | _0.27 ± 0.09_ | **0.30 ± 0.09** | 0.23 ± 0.08     | 0.24 ± 0.05   | 0.24 ± 0.12   | 0.21 ± 0.11   |
|        0.5  | 0.25 ± 0.11   | **0.30 ± 0.11** | 0.24 ± 0.08     | _0.28 ± 0.04_ | 0.18 ± 0.13   | 0.18 ± 0.09   |
|        1    | 0.25 ± 0.08   | **0.29 ± 0.09** | 0.19 ± 0.05     | _0.27 ± 0.06_ | 0.17 ± 0.08   | 0.17 ± 0.09   |
|        2    | _0.27 ± 0.07_ | **0.30 ± 0.07** | 0.18 ± 0.07     | 0.26 ± 0.08   | 0.20 ± 0.13   | 0.18 ± 0.14   |
|        4    | _0.24 ± 0.08_ | **0.26 ± 0.06** | 0.20 ± 0.08     | 0.24 ± 0.07   | 0.16 ± 0.08   | 0.17 ± 0.11   |

**Table 8: Comparison to neural networks on the mixed-curvature classification task**
| $\mathcal{P}$   | Product DT   | Product RF      | Tangent MLP     | Ambient MLP     | Tangent GNN   | Ambient GNN   |
|:------------|:-------------|:----------------|:----------------|:----------------|:--------------|:--------------|
| E           | 0.30 ± 0.11  | _0.33 ± 0.11_   | 0.31 ± 0.07     | **0.35 ± 0.06** | 0.26 ± 0.13   | 0.26 ± 0.13   |
| H           | 0.34 ± 0.13  | 0.39 ± 0.11     | **0.44 ± 0.08** | **0.44 ± 0.08** | 0.27 ± 0.14   | 0.28 ± 0.14   |
| HE          | 0.35 ± 0.13  | _0.37 ± 0.10_   | 0.35 ± 0.14     | **0.39 ± 0.13** | 0.20 ± 0.17   | 0.20 ± 0.10   |
| HH          | 0.32 ± 0.10  | _0.36 ± 0.06_   | 0.35 ± 0.07     | **0.38 ± 0.08** | 0.18 ± 0.09   | 0.18 ± 0.10   |
| HS          | 0.29 ± 0.09  | _0.35 ± 0.07_   | 0.33 ± 0.07     | **0.37 ± 0.12** | 0.22 ± 0.15   | 0.23 ± 0.07   |
| S           | 0.23 ± 0.06  | _0.25 ± 0.07_   | 0.23 ± 0.05     | **0.31 ± 0.06** | 0.13 ± 0.08   | 0.13 ± 0.08   |
| SE          | 0.31 ± 0.08  | **0.34 ± 0.08** | 0.30 ± 0.04     | _0.33 ± 0.06_   | 0.17 ± 0.11   | 0.18 ± 0.09   |
| SS          | 0.32 ± 0.06  | _0.35 ± 0.06_   | 0.28 ± 0.06     | **0.36 ± 0.08** | 0.18 ± 0.14   | 0.20 ± 0.13   |

---

> ### Author Response · Authors · 2024-12-02
> **Updated and expanded neural net benchmarks (continued)**
>
> **Table 9: Comparison to neural networks on the single-curvature regression task**
> |   $K$ | Product DT      | Product RF      | Tangent MLP   | Ambient MLP     | Tangent GNN     | Ambient GNN         |
> |------------:|:----------------|:----------------|:--------------|:----------------|:----------------|:--------------------|
> |       -4    | **0.19 ± 0.03** | **0.19 ± 0.03** | 0.34 ± 0.06   | 15.53 ± 36.13   | 0.50 ± 0.69     | 33387.07 ± 73833.56 |
> |       -2    | **0.21 ± 0.05** | **0.21 ± 0.05** | 0.35 ± 0.05   | 0.31 ± 0.24     | 0.43 ± 0.32     | 132.89 ± 334.04     |
> |       -1    | **0.19 ± 0.04** | **0.19 ± 0.04** | 0.32 ± 0.04   | 0.27 ± 0.23     | 0.51 ± 0.55     | 33.60 ± 97.79       |
> |       -0.5  | **0.19 ± 0.04** | **0.19 ± 0.04** | 0.32 ± 0.05   | **0.19 ± 0.05** | 0.31 ± 0.23     | 0.95 ± 0.91         |
> |       -0.25 | **0.19 ± 0.03** | **0.19 ± 0.03** | 0.33 ± 0.06   | 0.20 ± 0.04     | 0.46 ± 0.72     | 1.03 ± 2.20         |
> |        0    | _0.20 ± 0.03_   | **0.19 ± 0.03** | 0.33 ± 0.09   | 0.20 ± 0.03     | 0.27 ± 0.15     | 0.27 ± 0.15         |
> |        0.25 | **0.21 ± 0.04** | **0.21 ± 0.04** | 0.33 ± 0.06   | **0.21 ± 0.04** | 0.23 ± 0.07     | 0.27 ± 0.11         |
> |        0.5  | _0.20 ± 0.05_   | **0.19 ± 0.05** | 0.30 ± 0.05   | 0.20 ± 0.05     | 0.22 ± 0.06     | 0.22 ± 0.05         |
> |        1    | _0.20 ± 0.04_   | **0.19 ± 0.04** | 0.30 ± 0.05   | 0.20 ± 0.05     | 0.20 ± 0.04     | 0.23 ± 0.05         |
> |        2    | **0.21 ± 0.06** | **0.21 ± 0.07** | 0.27 ± 0.08   | **0.21 ± 0.07** | 0.23 ± 0.08     | 0.22 ± 0.07         |
> |        4    | **0.21 ± 0.04** | **0.21 ± 0.03** | 0.27 ± 0.05   | **0.21 ± 0.03** | **0.21 ± 0.03** | 0.26 ± 0.07         |
>
> **Table 10: comparison to neural networks on the multi-curvature regression task**
> | signature   | Product DT      | Product RF      | Tangent MLP   | Ambient MLP     | Tangent GNN   | Ambient GNN   |
> |:------------|:----------------|:----------------|:--------------|:----------------|:--------------|:--------------|
> | E           | _0.20 ± 0.04_   | **0.19 ± 0.04** | 0.36 ± 0.08   | 0.20 ± 0.04     | 0.32 ± 0.19   | 0.32 ± 0.19   |
> | H           | **0.19 ± 0.03** | **0.19 ± 0.02** | 0.32 ± 0.08   | 0.23 ± 0.07     | 0.74 ± 0.82   | 30.07 ± 48.71 |
> | HE          | **0.20 ± 0.05** | **0.20 ± 0.04** | 0.33 ± 0.07   | **0.21 ± 0.04** | 0.42 ± 0.48   | 2.39 ± 2.05   |
> | HH          | _0.19 ± 0.04_   | **0.18 ± 0.03** | 0.32 ± 0.09   | 0.20 ± 0.02     | 0.33 ± 0.36   | 4.53 ± 10.15  |
> | HS          | **0.20 ± 0.04** | **0.20 ± 0.04** | 0.33 ± 0.06   | **0.21 ± 0.04** | 0.26 ± 0.10   | 2.80 ± 2.79   |
> | S           | **0.18 ± 0.04** | **0.18 ± 0.04** | 0.25 ± 0.07   | **0.18 ± 0.04** | 0.20 ± 0.04   | 0.22 ± 0.04   |
> | SE          | **0.19 ± 0.06** | **0.19 ± 0.05** | 0.31 ± 0.06   | **0.19 ± 0.06** | 0.34 ± 0.66   | 0.51 ± 1.06   |
> | SS          | **0.20 ± 0.04** | **0.20 ± 0.03** | 0.31 ± 0.07   | **0.20 ± 0.04** | 0.24 ± 0.10   | 0.48 ± 0.24   |
>
> **Table 11: Comparison to neural baselines on graph node classification, regression, and link prediction tasks**
> | embedding   | Product DT      | Product RF      | Tangent MLP   | Ambient MLP     | Tangent GNN     | Ambient GNN     |
> |:------------|:----------------|:----------------|:--------------|:----------------|:----------------|:----------------|
> | citeseer (C)    | _0.26 ± 0.06_   | **0.27 ± 0.03** | 0.26 ± 0.03   | 0.26 ± 0.03     | 0.26 ± 0.03     | 0.26 ± 0.03     |
> | cora (C)        | **0.28 ± 0.02** | **0.29 ± 0.04** | 0.23 ± 0.04   | 0.23 ± 0.06     | **0.29 ± 0.03** | **0.29 ± 0.03** |
> | polblogs (C)    | **0.94 ± 0.03** | **0.94 ± 0.03** | 0.91 ± 0.05   | **0.94 ± 0.03** | 0.53 ± 0.06     | 0.50 ± 0.08     |
> | cs_phds (R)     | _17.79 ± 4.67_ | **16.02 ± 4.58** | 492.33 ± 89.38 | 30.81 ± 9.08  | 479.86 ± 588.41 | 506.83 ± 884.59 |
> | adjnoun (LP)    | 0.92 ± 0.05     | **0.94 ± 0.05** | 0.90 ± 0.07   | 0.91 ± 0.08   | **0.94 ± 0.05** | **0.94 ± 0.05** |
> | dolphins (LP)    | **0.95 ± 0.06** | 0.92 ± 0.05     | _0.93 ± 0.05_ | 0.92 ± 0.06   | 0.92 ± 0.05     | 0.92 ± 0.05     |
> | football (LP)   | **0.87 ± 0.15** | _0.86 ± 0.16_   | 0.84 ± 0.13   | 0.83 ± 0.19   | 0.86 ± 0.16     | 0.86 ± 0.16     |
> | karate_club (LP) | **0.93 ± 0.08** | _0.88 ± 0.15_   | 0.87 ± 0.11   | 0.86 ± 0.12   | 0.76 ± 0.57     | 0.83 ± 0.40     |
> | lesmis (LP)     | **0.95 ± 0.02** | _0.92 ± 0.06_   | 0.91 ± 0.09   | 0.91 ± 0.10   | 0.92 ± 0.06     | 0.92 ± 0.06     |
> | polbooks (LP)   | **0.94 ± 0.04** | **0.93 ± 0.04** | 0.90 ± 0.05   | 0.91 ± 0.05   | **0.93 ± 0.04** | **0.93 ± 0.04** |

---

> > ### Author Response · Authors · 2024-12-02
> > **Updated and expanded neural net benchmarks (continued)**
> >
> > **Table 12. VAE embedding classification benchmarks.**
> >
> > | embedding        | Product DT      | Product RF      | Tangent MLP     | Ambient MLP     | Tangent GNN   | Ambient GNN   |
> > |:-----------------|:----------------|:----------------|:----------------|:----------------|:--------------|:--------------|
> > | blood_cell_scrna | 0.17 ± 0.07     | 0.18 ± 0.05     | **0.20 ± 0.07** | _0.19 ± 0.05_   | 0.12 ± 0.02   | 0.12 ± 0.03   |
> > | cifar_100        | 0.09 ± 0.02     | 0.10 ± 0.03     | **0.11 ± 0.03** | **0.12 ± 0.05** | 0.05 ± 0.01   | 0.05 ± 0.02   |
> > | lymphoma         | **0.83 ± 0.08** | **0.83 ± 0.08** | 0.80 ± 0.08     | 0.81 ± 0.08     | 0.78 ± 0.04   | 0.78 ± 0.04   |
> > | mnist            | 0.30 ± 0.08     | _0.38 ± 0.15_   | **0.46 ± 0.18** | 0.38 ± 0.27     | 0.16 ± 0.06   | 0.11 ± 0.02   |
> >
> > **Table 13. Empirical dataset classification and regression benchmarks.**
> >
> > | dataset    | Product DT      | Product RF      | Tangent MLP   | Ambient MLP     | Tangent GNN   | Ambient GNN   |
> > |:-----------|:----------------|:----------------|:--------------|:----------------|:--------------|:--------------|
> > | landmasses (C) | _0.84 ± 0.02_   | 0.84 ± 0.03     | 0.79 ± 0.07   | **0.91 ± 0.05** | 0.70 ± 0.18   | 0.72 ± 0.12   |
> > | neuron_33 (C) | _0.75 ± 0.09_   | **0.77 ± 0.09** | 0.52 ± 0.11   | 0.43 ± 0.10     | 0.47 ± 0.06   | 0.47 ± 0.08   |
> > | neuron_46 (C) | **0.66 ± 0.08** | _0.64 ± 0.05_   | 0.49 ± 0.07   | 0.48 ± 0.05     | 0.49 ± 0.05   | 0.49 ± 0.06   |
> > | temperature (R) | _8.50 ± 3.06_   | **8.03 ± 2.64** | 13.25 ± 2.65  | 11.53 ± 3.08  | 11.33 ± 2.98  | 74330.68 ± 11838.19 |
> > | traffic (R)    | **0.54 ± 0.07** | _0.57 ± 0.08_   | 1.16 ± 0.09   | 0.86 ± 0.07   | 0.95 ± 0.20   | 0.99 ± 0.41         |

---

### Author Response · Authors · 2024-12-04
**Anonymized repo available**

In the interest of transparency and reproducibility, we have made an anonymized version of the complete Github repository for our paper available at https://anonymous.4open.science/r/embedders-F6F8/. Most experiments and benchmarks were carried out in the `notebooks` subdirectory, and in particular our new neural benchmarks can be found in `notebooks/43_benchmark_neural.ipynb`. We will add README files, remove unnecessary notebooks, and include a deanonymized link with our final camera-ready submission.

---

### Meta-Review · Area_Chair_Ganw · 2024-12-23

**Metareview:**

The authors propose to extend decision trees (DTs) and random forest (RF) into the product manifolds of hyperbolic, hyperspherical, or Euclidean components with a general angular split. The authors raised concerns about its novelty given results on product manifolds, and DTs/RF in the hyperbolic space. Although the authors evaluate the proposed methods on a large set of datasets, the Authors raised concerns on the sample size, baselines, time consumption. It is better to evaluate the proposed method on a systematic way to better convey the messages on advantages of the proposed method, and describe the experimental setting clearly. It is also good to specify how to determine the product manifold for a given dataset to apply the proposed approach. The Reviewers have mixed evaluations on the submission. Given aforementioned concerns, most of the Reviewers have not had enthusiastic supports on the submission yet. The authors may consider the Reviewers' comments to improve the submission.

**Additional Comments On Reviewer Discussion:**

The authors have not well-addressed concerns of the Reviewers on empirical evidences, e.g., baselines, sample sizes, experimental setup. The Reviewers also concern on the novelty of the proposed approach given results on the literature on product manifold, and corresponding results of DTs/RF on manifold, e.g., hyperbolic space.

---

### Decision · Program_Chairs · 2025-01-22

Reject